# Motor innervation directs the correct development of the mouse sympathetic nervous system

Alek G. Erickson [1,10], Alessia Motta [2,10], Maria Eleni Kastriti [1,3], Steven Edwards[4], Fanny Coulpier[5], Emy Théoulle[6], Aliia Murtazina [7], Irina Poverennaya [3], Daniel Wies[1], Jeremy Ganofsky [6], Giovanni Canu [8], Francois Lallemend [7], Piotr Topilko[5], Saida Hadjab [7], Kaj Fried [7], Christiana Ruhrberg [8], Quenten Schwarz [9], Valerie Castellani [6], Dario Bonanomi [2] ✉ & Igor Adameyko [1,3] ✉

The sympathetic nervous system controls bodily functions including vascular tone, cardiac rhythm, and the "fight-or-flight response". Sympathetic chain ganglia develop in parallel with preganglionic motor nerves extending from the neural tube, raising the question of whether axon targeting contributes to sympathetic chain formation. Using nerve-selective genetic ablations and lineage tracing in mouse, we reveal that motor nerve-associated Schwann cell precursors (SCPs) contribute sympathetic neurons and satellite glia after the initial seeding of sympathetic ganglia by neural crest. Motor nerve ablation causes mispositioning of SCP-derived sympathoblasts as well as sympathetic chain hypoplasia and fragmentation. Sympathetic neurons in motor-ablated embryos project precociously and abnormally towards dorsal root ganglia, eventually resulting in fusion of sympathetic and sensory ganglia. Cell interaction analysis identifies semaphorins as potential motor nerve-derived signaling molecules regulating sympathoblast positioning and outgrowth. Overall, central innervation functions both as infrastructure and regulatory niche to ensure the integrity of peripheral ganglia morphogenesis.

The autonomic nervous system (ANS), a crucial component of the peripheral nervous system (PNS), orchestrates a variety of physiological functions including heart rate, gut motility, and stress responses[1]. The sympathetic branch of the ANS controls the "fight or flight" response and is typically counteracted by the parasympathetic branch that mediates the "rest and digest" response[2]. The coordinated development of these two systems is vital for mammalian survival and

homeostasis, yet the mechanisms underlying their patterning remain enigmatic.

The sympathetic system consists of paired ganglia arranged along the spinal axis, adjacent to the dorsal aorta, which receive inputs from pre-ganglionic fibers extending from cholinergic visceral motor neurons within the ventral neural tube[3]. As most of the neuronal and all glial elements of the PNS, sympathetic ganglia originate from neural

[1]Department of Physiology and Pharmacology, Karolinska Institutet, Stockholm, Sweden. [2]Division of Neuroscience, IRCCS San Raffaele Scientific Institute, Milano, Italy. [3]Center for Brain Research, Department of Neuroimmunology, Medical University Vienna, Vienna, Austria. [4]Department of Applied Physics, KTH Royal Institute of Technology, Stockholm, Sweden. [5]Mondor Institute for Biomedical Research (IMRB), INSERM, Créteil, France. [6]University of Claude Bernard Lyon 1, MeLiS, CNRS, INSERM, NeuroMyoGene Institute, Lyon, France. [7]Department of Neuroscience, Biomedicum, Karolinska Institute, Stockholm, Sweden. [8]University College London, Department of Ophthalmology London, London, UK. [9]Center for Cancer Biology, University of South Australia, Adelaide, SA, Australia. [10]These authors contributed equally: Alek G. Erickson, Alessia Motta. ✉e-mail: bonanomi.dario@hsr.it; igor.adameyko@ki.se

crest cells (NCC), a transient migratory population that leaves the dorsal neural tube around embryonic day (E)9 in mice[4,5]. Early waves move ventrally towards the dorsal aorta where they differentiate into sympathetic neurons and glia[6,7]. Ventrolateral waves give rise to sensory cells of the dorsal root ganglia (DRGs) and the boundary cap cells (BCCs) lining the dorsal and ventral roots, while late dorsolateral waves generate the pigment-producing melanocytes. Some NCCs eventually become Schwann cell precursors (SCPs), which characteristically migrate along developing nerves[8]. SCPs contribute to various structures including parasympathetic ganglia[9,10], chromaffin cells of the adrenal medulla[11,12], and parts of the enteric nervous system[13], suggesting that peripheral nerves play a crucial role in ANS development.

Various aspects of PNS development depend on the correct positioning and growth of pre-existing nerve tracts[14–16]. A general wiring strategy, where an initial outgrowth of pioneer neurons establishes a template to guide subsequent axon growth, is evident in the pathfinding of some sympathetic and sensory fibers that depend on pre-extended motor axons[17]. Similarly, SCPs and multipotent glial progenitors of the boundary cap utilize the infrastructure provided by peripheral nerves to reach target organs. However, innervation is thought to be dispensable for the formation of sympathetic ganglia, as their initial assembly relies on early free-migrating NCC coalescing near the dorsal aorta[18]. Notwithstanding, it has been recently discovered that some sympathetic neurons in the paraganglia, as well as intra-adrenal sympathetic neurons, are derived from SCPs[12,19], pointing to the possibility that the sympathetic chain might also receive a contribution from SCPs to boost later growth. However, the extent to which motor nerves and motor nerve-associated SCPs contribute to the formation of the sympathetic chain remains undetermined.

In this study, we took advantage of genetic motor nerve ablation, in vivo tracing of boundary cap and glial lineages, and single-cell transcriptomics, to investigate the role of motor nerves in sympathetic chain formation. We discovered that genetic ablation of motor nerves impairs the development of sympathetic ganglia, as motor nerve-guided SCPs are unavailable. Simultaneously, aberrant clusters of sympathetic neurons are generated from sensory nerve-associated SCPs and become ectopically neurogenic in the absence of motor fibers. Finally, motor nerves control the navigation of sympathetic fibers, preventing them from innervating inappropriate targets, such as sensory ganglia. Thus, by controlling progenitor cell placement and axonal outgrowth, motor neurons act as "insulators" that maintain the separation between distinct elements of the PNS.

## Results

### Early-recruited SCPs are primed toward sympathetic fate while arriving on motor nerves

Sympathetic chain development begins with ventrally migrating waves of SOX10+ NCCs that coalesce in the vicinity of the dorsal aorta from where they receive inductive signals, such as Bone Morphogenetic Protein (BMP)[7]. These precursor cells, known as "sympathoblasts", express early autonomic markers (*Phox2b*) and later differentiate into bona fide sympathetic neurons (labeled by Tyrosine Hydroxylase, *Th*) (Fig. 1a)[2]. Soon after the initial assembly of sympathetic chain ganglia, preganglionic motor nerves start to arrive from the developing neural tube. To track the dynamics of motor axons during neural crest migration and subsequent steps of sympathetic ganglia development, we utilized *Hb9-GFP* transgenic mice in which motor neurons are labeled with green fluorescent protein (GFP) (Fig. 1b). We observed that outgrowing motor axons partly interrupted the neural crest wave, and freely migrating NCCs became associated with the nerve as SCPs. This transition is evidenced by a gradual shift in the migratory stream angle (Fig. 1c), widening gap between SCPs and free-migrating NCCs (Fig. 1d), and consolidation of the SOX10+ stream as the nerve grows towards the sympathetic anlagen, showing that SCPs do not leave the

nerve once attached (Fig. 1e). These relationships suggest that during early axon growth, motor nerves influence migratory patterns of SCPs.

While SCPs were previously distinguished from NCCs and ganglia-residing satellite glia solely on the basis of their association with nerves, a SCP-specific gene signature referred to as a "hub state" marked by *Itga4* expression has been recently described[20]. Indeed, at E10.5 ITGA4 levels were greater in SOX10+ cells that were associated to neurofilament-positive (2H3+) peripheral nerves of the ventral root, compared to NCCs that were still freely migrating in the caudal region (Fig. 1f), suggesting that nerve association coincided with the adoption of the SCP hub state. The conversion of NCCs into SCPs proceeded according to the axial maturation gradient (Fig. 1b and Suppl Fig. 1a), with anterior structures developing earlier than their posterior counterparts. This was also reflected in the gradients of motor innervation and induction/compaction of the PHOX2B+ sympathetic anlagen (Suppl Fig. 1b). Consequently, TH+ sympathetic neurons appeared first in the brachial and cervical regions at E10.5, while nearly all migrating SOX10+ cells in the trunk were already associated with growing peripheral nerves (Suppl Fig. 1c). During the early recruitment of SCPs, between E10.5 and E11.5, the ventral root is dominated by motor axons, as most TUJ1+ nerve fibers were also *Hb9-GFP*+ (Fig. 1g, h). By E12.5, the outgrowing spinal nerves carried both motor (efferent) and sensory (afferent) fibers that could be distinguished by *Hb9-GFP* and TRKA (NTRK1) labeling, respectively (Fig. 1i). Notably, the majority of the *white ramus communicans*, connecting to sympathetic ganglia, was composed of *Hb9-GFP*+ preganglionic visceral motor axons branching off the common nerve bundle, while TRKA+ viscerosensory axons extending from the DRGs were present in small numbers (Fig. 1i). Therefore, SCPs predominantly migrate along developing motor axons to reach the sympathetic anlagen (Fig. 1j).

Next, we reconstructed the development of the white ramus in *Hb9-GFP* embryos, seeking to identify SCPs transitioning towards the autonomic neuroblast fate, as indicated by the co-occurrence of SOX10 and PHOX2B expression. From E10.5 onwards, most NCCs that had migrated to the sympathetic chain expressed the autonomic marker PHOX2B (Fig. 2a). At this stage, some SCPs expressed PHOX2B at the extending tips of visceral motor nerves near the coalescing sympathetic ganglia (Fig. 2a, arrowheads), suggesting these cells were primed for the autonomic fate. At all later stages, this autonomic primed subpopulation of SCPs was detected exclusively along the white ramus or its fine branches (Fig. 2b, c). None of the observed SCP cell populations expressed TH along the ventral root, indicating that terminal sympathetic neuron maturation occurs only within the sympathetic ganglia (Fig. 2d). Together, these data are consistent with a model in which motor nerves recruit nearby freely migrating NCCs into SCPs, and that some of these motor nerve-associated SCPs become primed to an autonomic neurogenic fate en route as they approach the sympathetic ganglia, while others remain gliogenic (Fig. 2e).

### SCPs give rise to sympathetic chain neurons

The induction of autonomic markers observed in visceral motor nerve-associated SCPs in the dorsal aorta region suggests that they may give rise to a portion of the sympathetic chain neurons. Since NCCs in all rostral segments of the trunk have already become nerve-associated SCPs by E10.5 (Suppl Fig. 1)[21], we conducted SCP-specific lineage tracing with *Plp1-CreERT2; R26R-YFP*, inducing the reporter by tamoxifen injection at E10.5. In such an experimental setup, recombination occurs around E11, as SCPs presumably begin to contribute to the sympathetic fate. Embryos were collected at E13.5, and traced sympathetic neurons were quantified across the body axis (Fig. 3a). Approximately 10% of sympathetic neurons were labeled at the cervical level, while this proportion increased to around 40% in the posterior thoracolumbar region (Fig. 3b, c). The actual contribution of SCPs to sympathoblasts is likely higher since we have previously estimated the recombination efficiency of *Plp1-CreERT2* in SOX10+ cells to be

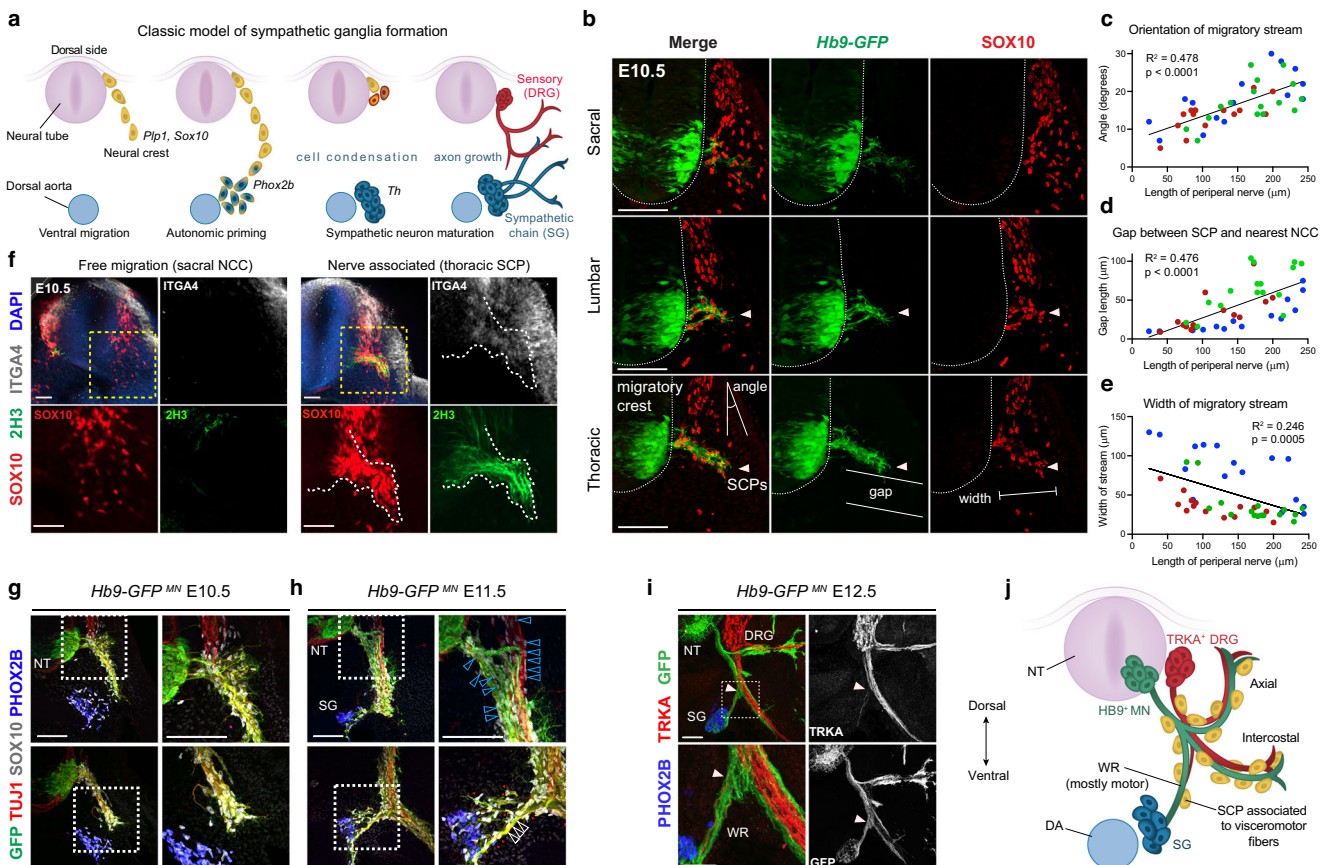

**Fig. 1 | SCP recruitment on motor nerves. a** Current model of sympathetic ganglia development from early stream of free neural crest migration. Free-NCCs (yellow) become primed toward autonomic fate (yellow-blue hybrid cells) as they approach the dorsal aorta region, where they differentiate into sympathetic neurons (blue) and coalesce to form the sympathetic ganglia chain. **b** Transverse sections of the same E10.5 *Hb9-GFP* embryo (representative of 3 embryos) from sacral to thoracic levels reveal the developmental progression of motor axons (*Hb9-GFP* labeling) exiting the ventral neural tube (outlined) and intersecting the stream of free NCCs (SOX10⁺). NCCs are recruited on motor nerves as SCPs (arrowheads) leaving a "gap" with freely migrating cells. Scatterplots of measurement of (**c**) the angle created by intersecting the line bisecting the NCC migratory stream, with the dorsoventral axis bisecting the neural tube (**d**) gap between nerve-associated SCP and the nearest free NCC, (**e**) mediolateral thickness of the NCC/SCP streams (distance between most medial and lateral SOX10⁺ cells just ventrolateral to the neural tube), perpendicular to (**c**). The red, blue, and green colors in (**c**–**e**) represent measurements from individual E10.5 embryos (n = 3). Linear regression assessed correlation coefficients and p-values. **f** E10.5 transverse optical sections (representative of 3 embryos) show ITGA4⁺/SOX10⁺ SCP associated with axons (2H3, Neurofilament) at thoracic level, and free migrating ITGA4⁻/SOX10⁺ NCCs at sacral level. Single channels are magnified from the boxed regions in the merge image. The nerve bundle is outlined. **g, h** Transverse sections of *Hb9-GFP* embryos. At E10.5 (**g**), SCPs (SOX10⁺, gray) migrate along GFP⁺/TUJ1⁺ motor axons that form the ventral root before sensory axons (GFP⁻/TUJ1⁺) have begun to extend from the DRG (representative of 3 embryos). At E11.5 (**h**), SCPs are associated with both motor and sensory fascicles in the main nerve bundle (blue arrowheads). Along the white ramus connecting to sympathetic ganglia (PHOX2B⁺, blue), SCPs are largely associated with motor axons (red arrows), which form the bulk of the nerve (white arrowheads). The boxed regions are magnified in the right panels. Representative of 6 embryos. **i** Transverse sections representative of n = 5 E12.5 *Hb9-GFP* embryos with TRKA staining of sensory projections (red) co-extending with motor axons (GFP⁺, green). Individual channels are shown in the right panels. The boxed area, magnified in the bottom panel, shows the white ramus (arrowhead) formed predominantly by preganglionic motor axons, with minimal contribution from viscerosensory fibers. PHOX2B (blue) marks sympathetic ganglia. **j** Schematic showing equal distribution of SCPs (yellow) on motor (green) and sensory (red) fibers in axial and intercostal nerves, while SCPs migrating toward sympathetic chain ganglia (blue) are almost exclusively recruited on motor axons of the white ramus. DA dorsal aorta, DRG dorsal root ganglia, MN motor neurons, NCCs neural crest cells, NT neural tube, SCPs Schwann cell precursors, SG sympathetic chain ganglia, WR white ramus communicans. Scale bars: **b**: 100 μm; **f**: 50 μm; **g**–**i**: 100 μm.

around 80%[11]. This data is consistent with a model in which a significant portion of sympathetic chain neurons are derived from nerve-associated SCPs rather than free-migrating NCCs, contrary to the traditional view of development. However, we note that this lineage tracing does not distinguish the neurogenic contribution of nerve-delivered SCPs from that of intra-ganglionic satellite glia derived from earlier-migrating NCCs.

To address this issue, we performed lineage tracing of boundary cap cells (BCCs), a multipotent neural crest derivative localized exclusively at CNS nerve exit/entry points[22]. BCCs are highly proliferative and differentiate into various neural crest-derived cell types, including nerve-associated SCPs[23]. BCCs markers (*Prss56* and *Egr2/Krox20*) are expressed at CNS entry/exit points but are absent

along spinal nerves and in the sympathetic chain[22]. Hence, Cre alleles based on these cell-specific markers are useful to determine whether BCC-derived SCPs contribute to the sympathetic chain via the nerve. In line with this possibility, we observed *Prss56-Cre*-traced cells primarily at the expected boundary cap location at E13.5, with a few dispersing along ventral root nerves, and some within posterior sympathetic chain ganglia, where they expressed TH (Fig. 3d, e). These results suggest that BCC-derived SCPs travel along the white ramus and are recruited in the ganglia where they give rise to sympathetic neurons.

As a complementary approach to identify BCC-derived nerve-associated SCPs, we undertook single-cell transcriptomics to profile *Prss56-Cre* and *Krox20-Cre*-traced cells dissected from the dorsal and ventral roots at E11.5 and E12.5 (Fig. 3f and Suppl Fig. 2). We detected

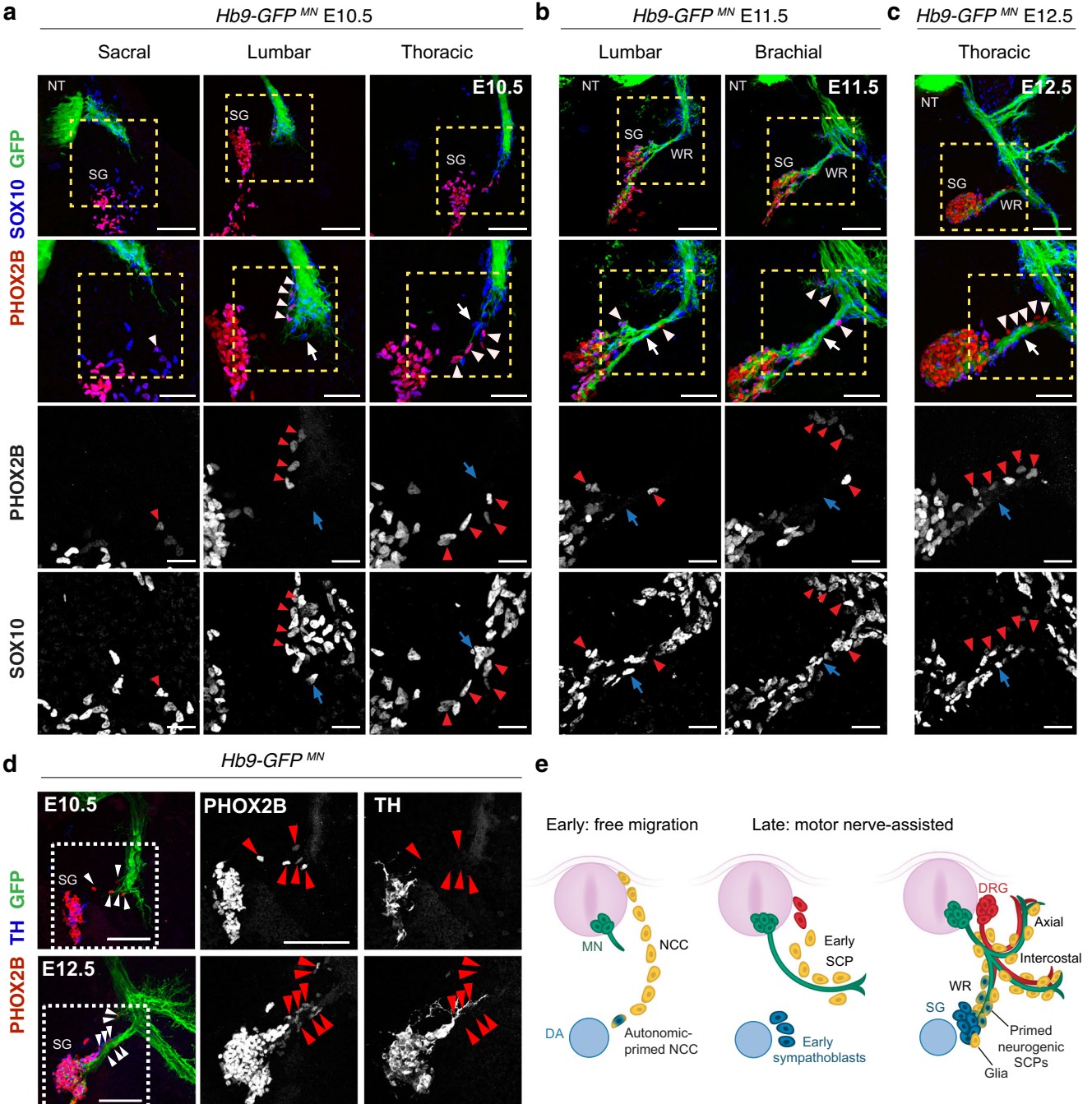

**Fig. 2 | SCPs are primed toward a sympathetic fate while migrating along motor nerves.** Transverse sections from E10.5 (**a**), E11.5 (**b**), and E12.5 (**c**) *Hb9-GFP* embryos immunostained for the NCC/SCP marker SOX10 (blue) and the sympathetic marker PHOX2B (red). Boxed regions are magnified in the corresponding bottom panels. Arrowheads point to committed PHOX2B⁺ SCPs associated with motor axons. Arrows point to PHOX2B⁻ SCPs that may acquire glial fate within the ganglia. Representative of at least n = 4 embryos per stage. **d** Transverse view of E10.5 (upper panel) and E12.5 (bottom panel) *Hb9-GFP* embryo trunks immunostained for the early sympathetic marker PHOX2B (red) and differentiated sympathetic neuron marker TH (blue). The boxed regions are magnified to show individual PHOX2B (middle) and TH (right) staining. Arrowheads point to PHOX2B⁺/TH⁻ SCPs

associated with motor axons. The images are representative of at least 3 embryos per stage. **e** Schematic of motor nerves (green) assisting the late wave of SCP migration (yellow) towards sympathetic ganglia (blue). Motor axons represent a permissive substrate for autonomic priming (yellow-blue hybrid cells; PHOX2B⁺/ TH⁻) but not for neuronal maturation (blue, TH⁺). SCPs that do not acquire PHOX2B expression might differentiate into satellite glial cells within sympathetic ganglia. DA dorsal aorta, DRG dorsal root ganglia, MN motor neurons, NCCs neural crest cells, NT neural tube, SCPs Schwann cell precursors, SG sympathetic chain ganglia, WR white ramus communicans. Scale bars: **a**–**c**: 100 μm, 50 μm, 25 μm (from top to bottom panels); **d**: 100 μm.

*Krox20/Prss56*-traced BCC derivatives expressing the hub state markers *Sox8* and *Itga4* (Fig. 3g). RNA velocity analysis revealed multiple trajectories including BCC-to-SCP and neuronal differentiation (Fig. 3f), which is primarily dominated by sensory neurogenesis (Fig. 3g). As expected from our analysis of lineage-traced cells on tissue

sections, some BCC-traced cells were found to express the pro-autonomic neuronal markers *Ascl1* and *Gata2*, as well as *Maoa*, *Mapt*, *Cartpt*, and *Th* (Fig. 3g).

From these studies, we conclude that *Plp1*⁺ SCPs and satellite glia contribute to late sympathetic neurogenesis (after E10.5), and that

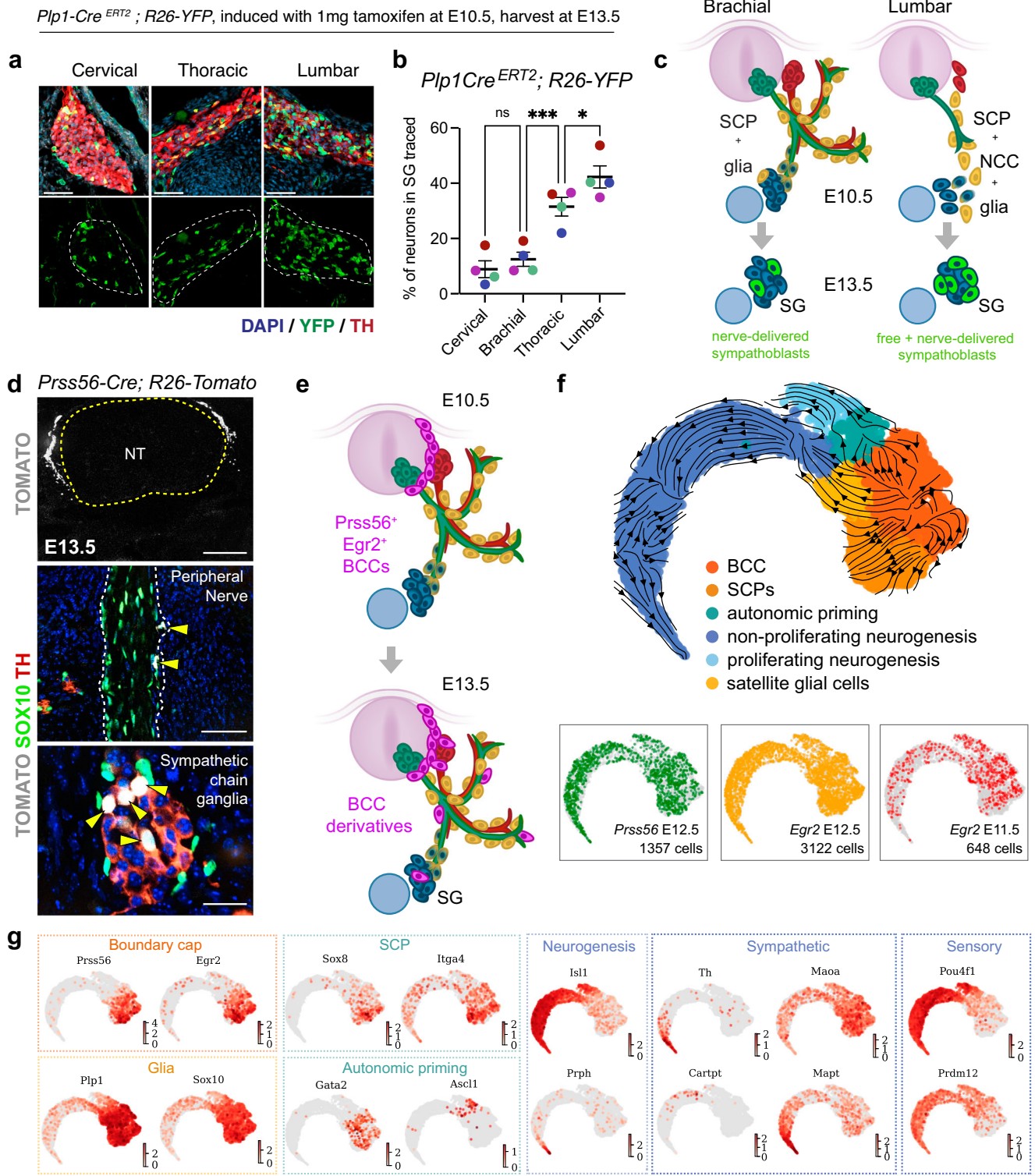

BCCs can give rise to sympathetic chain neurons via a nerve-associated SCP state.

## Motor ablation leads to misplacement of sympathoblasts associating with sensory fibers

To directly assess the requirement of motor nerves in sympathetic ganglia development, we analyzed *Olig2-Cre; R26-DTA* (referred to as *Olig2-Cre; DTA*) embryos in which motor neurons are selectively ablated following expression of diphtheria toxin (DTA) in *Olig2*[+] motor neuron progenitors. *Olig2-Cre; DTA* embryos displayed a near total loss

of *HB9*[+] motor neurons by E10.5 (Suppl Fig. 3a, b, c), resulting in complete depletion of motor axons in the ventral root (Fig. 4a, Suppl Fig. 3d)[24]. Therefore, the *Olig2-Cre; DTA* model effectively and selectively eliminates motor neurons and their associated axons from early embryonic stages. In these embryos, the pelvic ganglion, which forms independently from motor nerves[18], developed normally (Suppl Fig. 3e, f), whereas chromaffin cells in the adrenal medulla, largely originating from nerve-associated SCPs[11], were severely reduced (Suppl Fig. 3g). In addition, the genetic ablation of motor neurons affected the projection of DRG-derived sensory axons (Suppl Fig. 3h),

**Fig. 3 | Nerve-delivered SCPs contribute to sympathetic ganglia neurogenesis.** **a** *Plp1-Cre^ERT2; R26-YFP* embryos traced by tamoxifen injection at E10.5 and harvested at E13.5. Immunostaining for TH and YFP on sagittal sections through sympathetic ganglia at different anatomical locations of the traced *Plp1-Cre^ERT2; R26-YFP* embryos. **b** Quantifications of YFP⁺/TH⁺ cells traced at different levels of the sympathetic chain along the body axis. Colors represent individual embryos (n = 4). Mean ± SEM, one-way ANOVA and post hoc Tukey's Multiple Comparison Test, (\*\*\*) p = 0.001, (\*) p = 0.0343, (ns) p = 0.6829. **c** Schematic of lineage tracing experiment. Induction of YFP expression in *Plp1*⁺ cells (yellow) at E10.5 results in differential contribution to sympathetic ganglia along different body segments. At brachial levels (left), in E10.5 embryos all YFP-labeled NCC derivatives are associated with nerves, revealing the extent of SCP contribution to sympathetic ganglia neurons (green cells in the ganglia at E13.5). At lumbar levels (right), YFP is induced in both nerve-associated SCPs and residual free-migrating NCCs at E10.5, resulting in mixed NCC/SCP contribution to sympathetic neurons (larger fraction of green cells in the ganglia at E13.5). **d** Transverse sections of *Prss56-Cre; R26-Tomato* E13.5 embryos immunostained for TH (red) and SOX10 (green) visualized alongside endogenous Tomato fluorescence (TOM, gray). Nuclei are in blue. (Top) TOM⁺ boundary caps adjacent to the neural tube (outlined). (Middle) BCC-derived SCPs (TOM⁺/SOX10⁺, arrowheads) in the ventral root (outlined). (Bottom) TH⁺/TOM⁺ traced cells (arrowheads) in sympathetic chain ganglia from the lower lumbar region (observed in n = 5/5 embryos). **e** Schematic showing the restricted expression of BCC markers, *Prss56* and *Egr2* in the boundary caps at E10.5 (upper panel) and tracing of BCC derivatives (purple) along nerves and sympathetic ganglia (bottom panel). **f** Combined UMAP embedding with color-coded scRNAseq clusters (top) and sample origin (bottom). Arrows show RNA velocity-determined transcriptional flows. **g** Feature plots showing expression of selected cell type marker genes. SG sympathetic ganglia, BCCs boundary cap cells, NCCs neural crest cells, NT neural tube, SCPs Schwann cell precursors. Scale bars: **a**: 50 μm; **d**: 100 μm (top), 50 μm (middle), 25 μm (bottom).

in agreement with previous studies[17]. Lacking the motor component, peripheral nerves of the ventral root were shorter (Suppl Fig. 3i) and significantly thinner (Suppl Fig. 3j) than control embryos.

Notably, in line with the predominance of motor fibers, the white ramus was entirely absent in *Olig2-Cre; DTA* embryos, leaving only occasional mis-patterned viscerosensory axons reaching the sympathetic ganglia (Fig. 4b). This was accompanied by a marked decrease in SOX10⁺ glial cells delivered to the ganglia at E12.5 (Fig. 4c, d, Suppl Fig. 4a), as well as a 20% reduction in PHOX2B⁺ sympathoblasts inside the ganglia at this stage (Fig. 4e). These alterations were not caused by an overall developmental delay since mutant embryos did not exhibit defects in body length at E12.5 (Suppl Fig. 4b). Moreover, early placement/induction of the sympathetic chain was unaffected in *Olig2-Cre; DTA* (Suppl Fig. 4c–f) and the size of TH⁺ sympathoblasts was normal (Suppl Fig. 4g), although the ganglia appeared aberrantly aggregated (Suppl Fig. 4a).

We hypothesized that loss of the white ramus would prevent SCPs from reaching the sympathetic anlagen. Indeed, in *Olig2-Cre; DTA* embryos between E10.5 and E12.5, PHOX2B⁺ cells accumulated along the spared sensory nerves at the prospective branching point of visceral motor axons (Fig. 4f). These ectopically primed sympathoblasts spread aberrantly along the entire sensory nerve, from the DRG to the nerve endings (Fig. 4f–h) and were detected hundreds of micrometers away from the usual location near the sympathetic chain ganglia (Fig. 4i).

Overall, these results indicate that motor nerves influence the position of sympathetic progenitor cells. In the absence of motor axons, nerve-associated SCPs engage in a PHOX2B⁺ state, and these primed cells disperse along sensory fibers to distant locations away from their normal induction site near the dorsal aorta (Fig. 4j), at the expense of satellite glia and autonomic progenitor cells in the forming ganglia. These results distinguish sympathetic neurons and glia into two separate categories: an early population derived directly from free-migrating NCC, and a late population derived from nerve-associated SCPs. Whether these cell waves become functionally identical or not remains the subject of future investigation.

### Motor axons influence the timing and position of sympathetic differentiation and axonal projection

We noticed that many misplaced PHOX2B⁺ sympathoblasts associated with sensory nerves in motor-ablated embryos differentiated into TH⁺ neurons even at abnormal positions far from the dorsal aorta (Fig. 5a, Suppl Fig. 4h). This phenotype, in which sympathetic neurons mature and project along sensory nerves, worsened from E11.5 to E13.5, leading to the formation of several independent TH⁺ neuronal clusters of misplaced sympathoblasts and the concomitant "fragmentation" of the sympathetic chain (Fig. 5a, b).

Whole mount immunofluorescent staining revealed a constellation of ectopic TH⁺ cell clusters around the sympathetic chain throughout the thoracolumbar region (Fig. 5c, d), that were numerous in motor-ablated embryos but virtually absent in controls (Fig. 5e, f). These cells were found at a considerable distance from the sympathetic chain, extending as far as the limbs (Fig. 5g). The same phenotype was observed in a second independent mouse model of motor nerve ablation, *Hb9-Cre; Isl2-DTA* in which the DTA is expressed in postmitotic motor neurons based on overlapping expression of *Hb9* and *Isl2* (Suppl Fig. 5). Possibly as a consequence of sympathetic neuron misplacement, *Olig2-Cre; DTA* embryos presented severe disruption of the thoracic (Fig. 5h) and lumbar (Fig. 5i) sympathetic chain. In motor nerve-ablated embryos, the normally continuous, smoothly curved sympathetic chain became fragmented, with individual ganglia exhibiting a rounded, amorphous morphology (Fig. 5h, i, yellow arrowheads, and additional examples in Suppl Fig. 6), and was significantly smaller than in Cre-negative controls (Fig. 5j).

Aberrant sympathetic axon growth was also observed in nerve-ablated embryos, both from ectopic sympathetic neurons, as well as those properly positioned in the sympathetic chain (Fig. 5d, h, i). The misplaced sympathoblasts in nerve-ablated embryos extended axonal projections by E11.5, when sympathetic outgrowth had not yet commenced in controls (Fig. 5a, d). At later stages (E12.5-E13.5) in mutant embryos, sympathetic axons grew longer and visibly altered, spreading throughout the thoracolumbar region (Fig. 5d), ultimately invading the DRGs (Fig. 5h, i, red arrowheads). Sympathoblasts misplaced near the forelimbs (Fig. 5k–m) and hindlimbs (Suppl Fig 7) extended very long projections that abnormally innervated these regions in the mutant embryos.

The observation that removal of motor nerves enables the neurogenic potential within nerve-associated SCPs far away from traditional sites (sympathetic chain and dorsal aorta) (Fig. 5n) implies that motor nerves might inhibit this differentiation process during normal development. In addition, motor nerves appear to prevent premature and inappropriate innervation patterns, hinting they might serve as outgrowth and navigational controllers for sympathetic neuron projection.

### Early interactions with motor nerves direct morphogenesis of cervical ganglia

Although the fragmentation phenotype was mostly evident in the thoracic and lumbar parts of the trunk, the morphology of the superior cervical ganglion (SCG) was also affected in *Olig2-Cre; DTA* embryos. In mutants, the SCG was normal until E12.5 (Suppl Fig 8a) but became gradually more deformed starting from E13.5 (Suppl Fig 8b). Since these defects developed at later stages than those observed in the sympathetic chain, we used *Chat-Cre* allele to drive *R26-DTA* in postmitotic motor neurons starting at E11.5 (as opposed to *Olig2-Cre* that is expressed in progenitors). Unexpectedly, *Chat-Cre; R26-DTA* embryos exhibited a less severe SCG phenotype compared to *Olig2-Cre; DTA* embryos, showing only a moderate elongation of the ganglia in a few

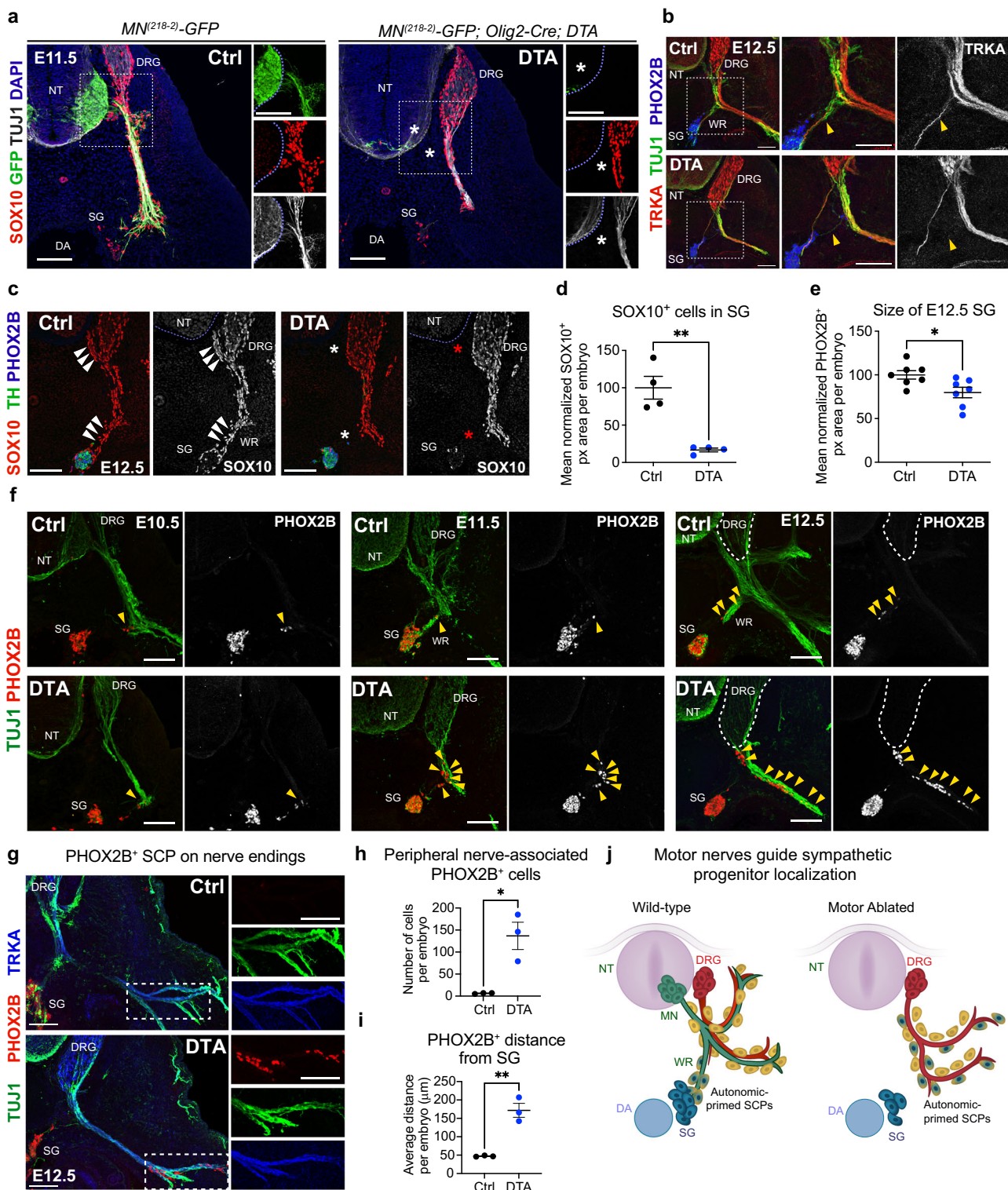

mutants (Suppl Fig 5b). Therefore, the different severity of the phenotypes observed following DTA-mediated cell ablation might depend on the timing of Cre-dependent recombination.

By E14.5 sympathoblasts in the cervical ganglia of *Olig2-Cre; DTA* embryos displayed a twofold reduction in volume and a corresponding enlargement of the medial cervical ganglion (MCG, Suppl Fig 8c–e). Importantly, these defects were exclusively dependent on the ablation of motor nerves, as they were not observed in *Neurog1* knockout embryos in which cranial sensory innervation is disrupted

(Suppl Fig 8c, f). Altogether, these results suggest that early recruitment of motor nerve-associated SCPs into cervical ganglia, prior to the complete phenotypic maturation of sympathetic neurons, is critical for later development and morphogenesis of the SCG. These changes occurred without apparent differences in the levels of cell proliferation and apoptosis in the SCG of mutant embryos at E12.5-E13.5 compared to controls (Suppl Fig 8g), suggesting a collective cell migration event might occur along motor nerves during formation of the SCG.

**Fig. 4 | Genetic ablation of motor neurons leads to ectopic autonomic priming along sensory nerves. a** Transverse sections showing complete loss of motor neuron cell bodies and axons (asterisks) in E11.5 *Olig2-Cre; DTA* embryos (right, n = 3) compared to control littermates (left, n = 2). Motor neurons are labeled with the cell-specific transgenic reporter *MN(218-2)-GFP*. In mutants, SCPs (SOX10+, red) migrate exclusively along sensory nerves (GFP-/TUJ1+). Individual labeling from the boxed regions are shown separately in the right panels. **b** Transverse sections at E12.5 showing mispatterning of viscerosensory projections (TRKA+/TUJ1+; arrowheads) in *Olig2-Cre;DTA* (bottom, n = 3) compared to controls (top, n = 3). PHOX2B (blue) identifies sympathetic ganglia. The boxed regions are magnified in middle and right panels. TRKA (gray) is shown separately in the right panels. **c** Transverse view of SCPs (SOX10, red) migrating on motor nerves (arrowheads) in E12.5 controls (left). SCP delivery to sympathetic ganglia (PHOX2B, TH) along the white ramus is interrupted (asterisks) in *Olig2-Cre;DTA* (right). SOX10 (gray) is shown separately in the right panels. **d** Area of glial cells (SOX10+ pixels) in sympathetic ganglia from E12.5 controls (black dots) and *Olig2-Cre;DTA* (blue dots). Mean (normalized to control) ± SEM, Unpaired two-sided t test (**) p = 0.0017; controls n = 4, mutants n = 4. **e** Sympathetic chain ganglia area (PHOX2B+ px) measured in transverse sections from E12.5 controls and *Olig2-Cre;DTA*. Mean (normalized to

control) ± SEM, Unpaired two-sided t test (*) p = 0.0226; controls n = 7, mutants n = 7. **f** Time course of autonomic priming identified by PHOX2B+ cells associated with peripheral nerves (TUJ1+) outside sympathetic ganglia (arrowheads) in E10.5, E11.5, and E12.5 *Olig2-Cre; DTA* (bottom) and control (top) embryos. PHOX2B is shown separately in gray. Images are representative of at least 3 embryos per genotype, per stage. **g** Ectopic autonomic priming (PHOX2B+ cells) along sensory nerves (TRKA+/TUJ1+) away from the ganglia chain in E12.5 *Olig2-Cre;DTA* (bottom), but not in controls (top). The merged channels in the boxed regions are shown separately in the right panels. Average number of nerve-associated primed SCPs (PHOX2B+ cells outside the ganglia) (**h**) and distance from ganglia (**i**) in controls and *Olig2-Cre; DTA*. Mean ± SEM, Unpaired two-sided t test (*) p = 0.0136; (**) p = 0.0027; controls n = 3, mutants n = 3. **j** Schematics of autonomic priming (yellow-blue hybrid cells) on motor axons of the white ramus in the vicinity of the dorsal aorta in controls (left) compared to uncontrolled aberrant priming along sensory nerves far away from the sympathetic chain in motor nerve-ablated mutants (right). DA dorsal aorta, DRG dorsal root ganglia, MN motor neurons, NT neural tube, SCPs Schwann cell precursors, SG sympathetic chain ganglia, WR white ramus communicans. Scale bars: 100 μm.

## Motor nerves prevent intermixing of sensory and autonomic ganglia

We next examined the organization of peripheral ganglia at intermediate and late developmental stages in motor nerve ablated embryos. *Olig2-Cre; DTA* mice are not viable at birth, but survive in the womb, allowing investigation of embryos until birth. At E15.5, the ectopic clusters of sympathetic neurons often appeared larger than the regular sympathetic ganglia (Fig. 6a) and were frequently abutting the DRGs, resulting in a significantly shorter distance between sympathetic and sensory ganglia (Fig. 6b). Even more drastic alterations were observed at E18.5, wherein the DRGs and sympathetic chain ganglia were interspersed, with sensory neuroblasts intermingling within the sympathetic ganglia (Fig. 6c, insets). These defects were observed in most of the peripheral ganglia in mutant embryos, but never in controls (Fig. 6d). Conversely, sympathetic neurons and their axons aberrantly invaded the DRGs thereby disrupting their structure (Fig. 6c, f). Consequently, sensory ganglia were generally smaller (Fig. 6e) and exhibited an abnormal morphology (Fig. 6f).

These findings suggest that motor nerves, potentially through the regulation of progenitor cell placement and axon navigation (or via other mechanisms), function as "insulators", safeguarding the integrity of boundaries between different types of peripheral ganglia (Fig. 6g).

## Signaling interactions between motor neurons and sympathetic progenitor cells

To investigate how motor neurons participate in signaling interactions with neural crest-derived cells like SCPs and sympathetic neurons, we analyzed single-cell transcriptomic datasets of motor neurons[24] as well as the neural crest lineage, including hub-state SCPs, boundary cap cells, satellite glia, and sympathetic neurons at E12.5 (Suppl Fig 9a)[20]. The CellChat algorithm[25] was used to predict ligand-receptor interactions between motor neurons and neural crest-derived cell types harvested at E12.5. The highest predicted scores for signaling outgoing from motor neurons, while incoming to peripheral neuroglial cell types, were assigned to class-3 Semaphorin (SEMA3), Pleiotrophin (PTN), Neuregulin (NRG), Macrophage migration inhibitory factor (MIF), Energy Homeostasis Associated (ENHO), and Growth Arrest Specific (GAS) pathways (Suppl Fig 9b, c). Most motor neuron subtypes, including preganglionic motor neurons (PGCa and PGCb), were predicted to signal via these selected pathways to boundary cap cells, hub-state SCPs, satellite glia and sympathetic neurons (Suppl Fig 9b–d). Gene expression analysis of receptor/ligand pairs for two of the highest-scoring pathways, SEMA3 and NRG, identified multiple specific ligands in motor

neurons, and high levels of cognate receptors in neural crest derivatives (Suppl Fig 9e, f). Focusing on SEMA3 pathway, the top-ranking ligands expressed in PGC motor neurons were Sema3C, followed by Sema3A (Suppl Fig 9e, g). Therefore, we hypothesized that deregulation in SEMA3 signaling might be partly responsible for the phenotypes observed in motor nerve-ablated embryos.

To address this possibility, we examined sympathetic chain development in SEMA3-deficient embryos. The top predicted signal, SEMA3C, is highly expressed by motor neurons (Suppl Fig 10a) and has been shown to control neurovascular interactions through Neuropilin-1 (Nrp-1)/Neuropilin-2 (Nrp-2)/Plexin-D1 receptor complexes in endothelial cells[26–28]. However, the analysis of *Sema3C* knockout embryos at E12.5 and E13.5 did not reveal abnormalities in sympathetic ganglia (Suppl Fig 10b, c). The other candidate class-3 semaphorins, Sema3A and Sema3F, were expressed in motor neurons (Suppl Fig 11a, b)[28] and have been shown to provide chemo-repulsive cues for migrating NCCs and sympathetic fibers expressing cognate Neuropilin and Plexin receptors[29–33]. Sympathetic ganglia abnormalities resembling the phenotypes induced by motor nerve ablation were observed in both *Sema3a/3f* double knockout embryos (Suppl Fig 11) and *Wnt1-Cre;Nrp1flox/flox* embryos in which *Nrp1* was deleted from the neural crest[31] (Suppl Fig 12). In both mutant models, the sympathetic chain appeared moderately fragmented (Suppl Fig 11c, d, Suppl Fig 12a–f, arrows) and small clusters of misplaced sympathetic neurons were observed at ectopic locations at multiple axial levels mostly dispersing in the forelimbs where they extended aberrant projections (Suppl Fig 11c–f and Suppl Fig 12c–f, arrowheads). In addition, mutants displayed severe disruption of the cervical sympathetic ganglia (Suppl Fig 11d, and Suppl Fig 12f). Fragmentation of the sympathetic chain and ectopic clusters were also visible in *Nrp2-/-; Nrp1Sema/Sema* mutants in which all class-3 semaphorin signaling is abolished[34] (Suppl Fig 11g). However, the abnormally robust outgrowth of sympathetic fibers in the hindlimb and thoracic regions observed in motor nerve-ablated embryos was not recapitulated in *Nrp1* and *Sema3a/3f* signaling mutants. These results suggest that motor nerve derived SEMA3A/3 F may be partly responsible for the placement of SCPs and structural integrity of sympathetic ganglia, but that motor nerves also use other mechanisms to regulate sympathoblast maturation and axonal outgrowth. These aspects of the motor nerve-driven phenotype, independent of semaphorin signaling, warrant future investigation.

## Discussion

Besides transmitting information through synaptic contacts, peripheral axons influence the innervated tissues in a variety of ways, by

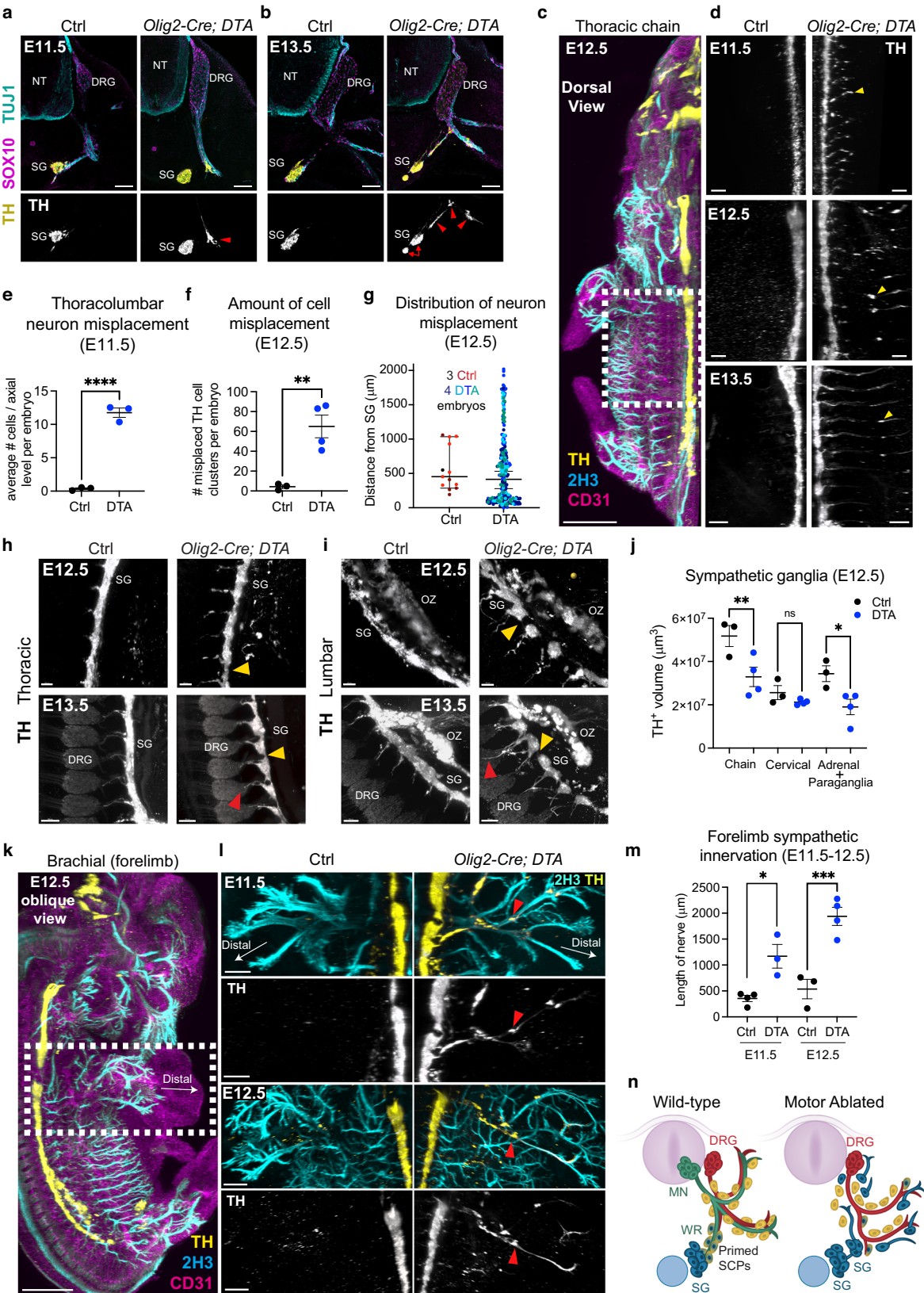

secreting morphogenic navigating signals[35], organizing nerve tracts through axon-axon interactions[36,37], guiding and remodeling blood vessels[28,38,39] and delivering progenitors of multiple cell types (such as melanocytes, chromaffin cells, autonomic neurons) to their final destination during embryonic development[9,11]. Because the PNS is multifunctional, the formation of innervation patterns has far-reaching consequences for organismal physiology[40]. The co-dependency of different nerve types reflects a parsimonious solution for neural wiring and represents an appealing developmental mechanism for generating evolutionary novelty[41]. Here, we addressed whether assembly of the sympathetic autonomic system depends on early-projecting and pioneering visceral motor neurons of the spinal cord[42].

**Fig. 5 | Motor innervation controls sympathetic neuron differentiation, axonal projection and chain ganglia morphology.** Transverse sections at E11.5 (**a**) and E13.5 (**b**) showing misplaced sympathoblasts differentiating into sympathetic neurons (TH⁺, yellow) along sensory nerves (TUJ1⁺, cyan) and forming ectopic mini-ganglia in *Olig2-Cre; DTA* (arrowheads) but not in controls. SOX10 labels SCPs (magenta). TH staining is shown separately in the bottom panels (gray). Arrows point to fragmented sympathetic chain ganglia. Representative images of 6 embryos per genotype at E11.5 and 4 embryos per genotype at E13.5. **c** Dorsal view of a whole mount immunostaining for TH, Neurofilament (2H3), and vascular marker CD31 of E12.5 control embryo. The region encompassing the sympathetic chain visualized in (d) is outlined. **d** Dorsal view of a whole mount immunostaining for TH in control (left) and *Olig2-Cre; DTA* (right) E11.5 (top), E12.5 (middle), and E13.5 (bottom) embryos. Arrowheads point to sensory fiber-associated TH⁺ sympathetic neurons distant from the chain ganglia. Premature and aberrant extension of sympathetic fibers is visible in mutants. Representative images of at least n = 5 embryos per genotype at each stage. **e** Quantification of misplaced sensory nerve-attached sympathetic neurons per DRG between brachial and lumbar levels, averaged per embryo. Mean ± SEM, Unpaired two-sided t test (****) p < 0.0001; controls n = 3, mutants n = 3. **f** Quantification of misplaced TH⁺ cells outside the chain ganglia at E12.5, per embryo. Mean ± SEM, Unpaired two-sided t test (**) p = 0.0066; controls n = 3, mutants n = 4. **g** Distribution of the distances between misplaced TH⁺ cells and the borders of the chain ganglia. Each color of dot represents a different embryo (n = 3 controls, n = 4 mutants). Sagittal view of thoracic (**h**) and lumbar (**i**) regions

from whole mount TH immunostaining of E12.5 (top) and E13.5 (bottom) *Olig2-Cre; DTA* embryos (right) and control littermates (left). Yellow arrowheads point to fragmented sympathetic chain in mutants; red arrowheads indicate abnormal sympathetic axon growth. Representative images of at least n = 5 embryos per genotype at each stage. **j** Volume measurements of sympathetic chain ganglia, cervical ganglia, and adrenal/paraganglia. Datapoints represent the average volume of ganglia from individual embryos (n = 3 control, n = 4 mutant). Mean ± SEM, one-way ANOVA using Šidák correction for multiple comparisons (**) p = 0.0032; (*) p = 0.0251; ns: p = 0.9815. **k** Oblique view of whole mount immunostaining for TH, Neurofilament (2H3), and vascular marker CD31 of E12.5 control embryo. The outlined forelimb region is shown in (**l**). **l** Forelimbs of E11.5 (top) and E12.5 (bottom) controls (left) and *Olig2-Cre; DTA* (right) littermates. Red arrowheads point to ectopic TH⁺ neurons extending axons aberrantly into the limb. **m** Length of TH⁺ sympathetic axons innervating the forelimb in control littermates vs *Olig2-Cre; DTA* embryos (at E11, controls n = 4, mutants n = 3; at E12.5, controls n = 3, mutants n = 4). Mean ± SEM, one-way ANOVA using Šidák correction for multiple comparisons (***) p = 0.0003; (*) p = 0.0118. **n** Schematic showing both ectopic and normally positioned sympathetic neurons (blue) projecting prematurely and along inappropriate paths in association with sensory fibers in motor-ablated mutants. The sympathetic chain is fragmented. DRG dorsal root ganglia, MN motor neurons, NT neural tube, OZ organ of Zuckerkandl, SCPs Schwann cell precursors, SG sympathetic chain ganglia. Scale bars: **a**, **b**: 100 μm; **c**: 500 μm; **d**: 200 μm (top and middle), 300 μm (bottom); **h**, **i**: 100 μm (top), 200 μm (bottom); **k**: 500 μm; **l**: 200 μm (top), 300 μm (bottom).

First, we report that the developing sympathetic chain requires motor innervation to provide a cell source to supplement its growth. The motor nerves are covered by multipotent and neurogenic SCPs, which are neural crest derivatives similarly to sympathetic neurons[43]. In the classical paradigm, migratory neural crest cells arrive at the dorsal aorta and coalesce into primary sympathetic ganglia[7]. We show that as motor axons emerge from the neural tube, they encounter part of the freely migrating neural crest, which becomes associated with growing motor nerves, turning into transcriptionally-defined SCPs—residing along the entire length of the peripheral nerve[43]—and BCCs—residing at CNS nerve exit points[22]. Consequently, the timing of motor axon outgrowth at each rostrocaudal segment coincides with a switch from free- to axon-associated dispersion of neural crest derivatives, consistent with our earlier studies[21]. Early-extending motor axons appear to function as an adhesive barrier that intersects the ventral migratory path of neural crest cells converting them into BCCs and SCPs (Fig. 1b–f). These highly plastic cell types later contribute to the sympathoblasts and satellite glia of the sympathetic chain, following the initial seeding of the ganglia by early migrating NCCs.

Unexpectedly, we found that SCPs are primed towards the autonomic fate while still moving along the extending visceral motor nerves to reach the sympathetic anlagen. To validate this, we lineage-traced nerve exit point-associated BCCs and their SCP derivatives, revealing their contribution to sympathetic chain neuroblasts and satellite glia. These results were supported by pseudotime trajectory analysis of single-cell transcriptomics data of traced progeny of BCCs. Furthermore, we observed a 20% loss of sympathoblasts in the developing sympathetic chain upon genetic ablation of motor neurons. The contribution of motor nerve-associated BCCs and SCPs to the sympathetic chain is reminiscent of the SCP-dependent origin of parasympathetic neurons, melanocytes, and chromaffin cells of the adrenal medulla[8,9,11]. Also, it supports the evolutionary concept proposing nerve-assisted tissue invasion as the archetypical mode of neural crest dispersal in prehistoric vertebrates[44].

The genetic ablation experiment pointed to additional roles for motor nerves besides supplying SCPs to the developing sympathetic chain. For instance, the induction of sympathetic ganglia stationed along sensory fibers in the limb far from the dorsal aorta cannot be attributed solely to the misplacement or limited availability of SCPs for two reasons. First, the number of observed ectopic primed cells outnumbers the corresponding reduction inside the ganglia. Second, the ectopic priming resulting from motor neuron ablation occurs in

regions where motor nerves are normally covered with migrating SCP, implying that potential inductive signals present at those location, for instance in the forelimbs, are kept dormant in the presence of motor nerves. Thus, sensory nerve-associated SCPs retain the autonomic neuron differentiation potential, and in the absence of repressing signals from motor axons, they differentiate into sympathetic neurons in response to pro-neurogenic factors secreted by the local environment[7,45]. In motor-ablated embryos, the misplaced sympathoblasts, as well as those appropriately located in the ganglia, matured and projected at an accelerated pace compared to controls, suggesting that motor nerves not only serve as a scaffold to guide sympathoblasts but also regulate their neuronal maturation. It would follow that a fine balance exists between gliogenic and neurogenic potential in individual SCPs, with local signaling cues regulating the probability of nerve-associated neurogenesis[10,46]. Knowing how motor nerves regulate cell migration and neuroglial cell fate might have implications for neuroblastoma pathogenesis. Specifically, future studies of nerve-derived cues affecting sympathetic priming, especially functional differences between the sensory versus motor niches, might improve our understanding of cell origins of tumors initiated in non-canonical locations[47].

Previous studies highlighted how early-established motor nerve tracts influence the subsequent trajectories of other nerve fiber types[17,37]. This was elegantly demonstrated by Wang et al. showing how different types of nerve ablations affect co-dependent patterns of motor, sensory and sympathetic innervation in skin and limbs[17]. According to Wang et al., motor axons are essential for the subcutaneous navigation of sensory axons, and in turn, sympathetic efferent fibers require those correctly positioned sensory afferents to innervate the dermis. Furthermore, genetic removal of sensory afferent fibers during development showed that sympathetic fibers successfully follow motor nerve trajectories before entering the skin, but subsequently fail to innervate the skin entirely. This dependence of sensory axon on motor axons is anatomy-dependent, because in the absence of motoneurons, the majority of trunk sensory axons successfully navigate along normal peripheral pathways in the ventral root, showing major projection abnormalities only at further extremities such as the limbs and skin. Indeed, our motor ablated embryos show a mainly normal distribution of sensory fibers in the vicinity of the sympathetic chain and DRGs, with only minor or rare deviations (such as slightly shifted branching point of the white ramus). Together with the paucity of viscerosensory axons in the white ramus, these observations suggest that the premature sympathetic nerve

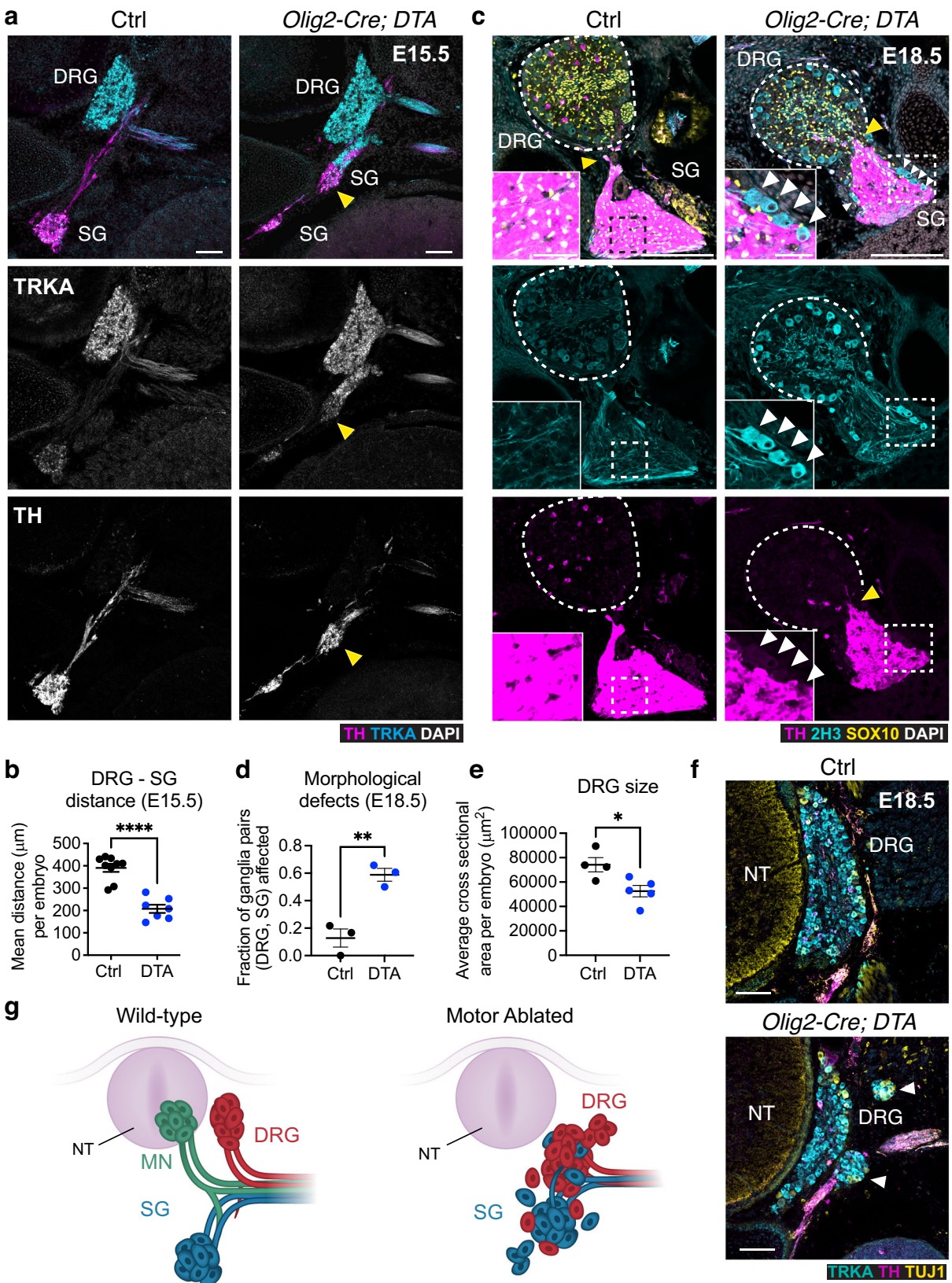

outgrowth and abnormal navigation along the ventral root can be most appropriately attributed to the absence of motor fibers.

Given this logic, our findings imply a specific role of motor nerves in inhibiting precocious SG axon outgrowth and sympathetic innervation of DRGs. Indeed, in this experimental setting, both properly positioned and ectopic sympathetic neurons mis-projected into DRGs, away from their canonical path. Ultimately, at later developmental stages, sympathetic and sensory ganglia started to intermingle, causing inappropriate mixing of different neuronal types. This aberrant configuration resembles or may even model human cases of sympathetic pain syndrome, where chronic pain arises from abnormal synaptic connections between sympathetic

**Fig. 6 | Motor nerves serve as insulators to prevent abnormal intermixing of sympathetic and sensory ganglia. a** Transverse embryo sections at E15.5 showing fusion of sympathetic (TH⁺, magenta) and dorsal root ganglia (TRKA⁺, cyan) in *Olig2-Cre; DTA* (right), but not in control littermates (left). TRKA and TH are shown separately in middle and bottom panels, respectively. **b** Distance between sympathetic and dorsal root ganglia in *Olig2-Cre;DTA* and controls. Mean ± SEM, Unpaired two-sided t test (****) p < 0.0001; controls n = 9, mutants n = 7. **c** Immunostaining for TH (magenta), SOX10 (yellow), and 2H3 (cyan) in sagittal sections of *Olig2-Cre; DTA* embryos and control littermates. The dorsal root ganglion is outlined with a dotted line. Boxed regions are magnified in the insets showing interspersed sensory neurons in sympathetic ganglia (white arrowheads). Yellow arrowheads point to fusion of DRG and sympathetic ganglia in mutants. **d** Fraction of ganglia pairs showing aberrant organization such as misplaced sympathoblasts around the sensory ganglia, or misplaced sensory neurons in the sympathetic ganglia. Mean ± SEM, Unpaired two-sided t test (**) p = 0.0046; controls n = 3, mutants n = 3. **e** Average cross-sectional area of DRGs per embryo. Mean ± SEM, Unpaired two-sided t test (*) p = 0.0196; controls n = 4, mutants n = 5. **f** Transverse view of fragmented DRG (TRKA⁺, cyan) (arrowheads) in E18.5 *Olig2-Cre; DTA* (bottom) but not controls (top). TH (magenta) and TUJ1 (yellow) mark sympathetic projections. Representative images of n = 4 controls and n = 5 mutants. **g** Schematic showing intermixing of sympathetic (blue) and sensory (red) ganglia in the absence of motor nerves (green). NT neural tube, SG sympathetic ganglia, DRG dorsal root ganglia, MN motor neurons. Scale bars: **a**: 100 μm; **c**: 200 μm (insets 50 μm); **f**: 100 μm.

neurons and pain somatosensory neurons located in DRGs[48–50]. It would be interesting to test whether perturbation of specific motor-derived signals, or transient redirection of SCPs, is sufficient to recreate such syndromes in a mouse model. Altogether, our results point to a restraining effect of preganglionic motor fibers on the projection of sympathetic nerves.

Unmasking the molecular mechanisms underlying the regulatory action of motor nerves on sympathoblast priming, maturation, and outgrowth remains an important biological question for future studies. Notably, the phenotype observed in the absence of motor axons is reminiscent of the defects that arise in the sympathetic chain when Semaphorin-3/Neuropilin signaling is impaired in mutant embryos. Sema3/NRP pathway is required for sympathetic nervous system development[29] and for placing of chromaffin cell precursors in the adrenal medulla following visceral motor nerves[31]. Motor neurons express multiple semaphorins[51] and use them to regulate guidance receptors in an autocrine fashion[52], and as paracrine signals to control the interactions between developing motor axons and the cells in the innervated tissues, including vascular endothelial cells[27,28]. Our results support the possibility that *Sema3A* and *Sema3F* ligands released by extending preganglionic motor nerves orchestrate the local induction and spatial organization of sympathoblasts, because the clusters of ectopic nerve-associated sympathoblasts observed in *Sema3a/3f* and *Nrp1* knockout embryos strongly resemble those in motor nerve-ablated embryos. Other aspects of the motor nerve ablation phenotype in the sympathetic chain were not recapitulated in Sema pathway mutants, suggesting that motor nerves may utilize multiple mechanisms to influence the developing sympathetic system. In addition to this logic, because the activation of *Sema3a/3f−Nrp1* signaling has been implicated in peripheral nerve targeting to muscles and adrenal primordia[31,53,54], a contribution of motor axonal misrouting to SCP disorganization and ectopic mini-ganglia formation cannot be excluded.

The exact identification of motor-derived signals that influence sympathetic development is complicated by the fact that this effect could be in part mediated by co-extending sensory axons[17,36,37]. Also, in our experiments we cannot exclude the influence of semaphorins and other signals derived from local mesodermal populations, including the developing somites[55]. It is possible that a combination of extrinsic cues from different cell sources directs the induction and maturation of sympathetic neurons, as well as controls the navigation of sympathetic nerves. Altogether, these features complicate the unambiguous dissection of the underlying molecular signals, making it a compelling subject for future studies aimed at clarifying how motor fibers influence the surrounding cellular microenvironment.

In conclusion, the role of motor nerves in directing sympathetic chain development is pivotal and multifarious at the same time. Our results demonstrate that motor nerves function as a regulatory brake on the maturation and outgrowth of sympathetic neurons, "insulate" distinct PNS elements to safeguard their integrity, and provide an essential niche and migratory substrate for a portion of motor nerve-associated progenitors, thereby contributing to the development of the sympathetic nervous system.

## Methods

### Mouse lines
All animal work was permitted by the Ethical Committee on Animal Experiments (Stockholm North committee) and Animal Research Committee of IRCCS San Raffaele Hospital, and conducted in compliance with The Swedish Animal Agency's Provisions and Guidelines for Animal Experimentation recommendations under I.A.'s ethical protocol #15907-19; 18314-21 and The Italian Ministry of Health under D.B.'s protocols #1131/2016-PR and 668/2022-PR. Mice were kept in standard conditions: 24 °C; 12h-12h light dark cycle; 40–60% humidity; food and water ad libitum. *R26-Tomato* mice were ordered from The Jackson Laboratory (stock number 007914). *Plp1-Cre^ERT2* mice were received from U. Suter laboratory (ETH Zurich, Switzerland) (http://www.informatics.jax.org/allele/MGI:2663093). *R26-YFP* mice were received from The Jackson Laboratory (stock number 006148, full strain name B6.129×1-Gt(ROSA)26Sortm1(EYFP)Cos/J). *Hb9-Cre* (also known as *Mnx1-Cre*) mice were received from The Jackson Laboratory, stock number 006600 (full strain name B6.129S1-Mnx1tm4(cre)Tmj/J). *Isl2-DTA* mice were received from The Jackson Laboratory, stock number 007942 (full strain name B6.Cg-Isl2tm1Arbr/J). *R26-DTA* alleles were received from The Jackson Laboratory, stock numbers 006331 (full strain name Gt(ROSA)26Sortm1(DTA)Jpmb/J) and 010527 (full strain name B6;129-Gt(ROSA)26Sortm1(DTA)Mrc/J)[56]. *Chat-Cre* mice were received from K. Meletis lab (Karolinska Institutet) also available from the Jackson Laboratory, stock number 006410 (full strain name B6;129S6-Chattm2(cre)Lowl/J). *Olig2-Cre* mouse line (C57BL6/n background) was donated by T. Jessel laboratory (Columbia University, New York, US)[57]. *Hb9-GFP* and *MN^(2I8-2)-GFP* mouse lines (C57BL6/n background) were donated by S. Pfaff laboratory (Salk Institute, San Diego, US)[58,59]. Mouse mutants deficient in semaphorin and neuropilin signaling (*Wnt1-Cre Nrp1^flox/flox, Sema3a/3f-DKO, Nrp1^SEMA Nrp2 KO*) as well as strains used for boundary cap cell tracing (*Egr2-Cre, Prss56-Cre*) have been described previously[22,29–31,34]. *Sema3C KO* mouse line (CD1 background) was donated by J. Raper (University of Pennsylvania, Philadelphia, US) and (S. Chauvet Aix-Marseille Université, Marseille, France)[60]. For all experiments, the day the plug was detected was considered E0.5. For *Plp1-Cre; R26-YFP* lineage tracing experiments, tamoxifen (Sigma, T5648) was dissolved in corn oil (Sigma, 8267) and delivered via intra peritoneal (i.p.) injection to pregnant females (0.05 mg/g body weight). For embryo collection, euthanasia was performed via isofluorane overdose followed by cervical dislocation. Sex information was not collected for the experimental analysis of embryonic tissue because sexual dimorphisms are not prominent at the stages investigated.

### Motor neuron ablation and lineage tracing experiments
For targeted ablation of the pre-ganglionic neurons, *Isl2-DTA* or *R26-DTA* mice were bred to *Hb9-Cre*, mice, *Chat-Cre* mice, or *Olig2-Cre* mice to generate experimental *Hb9-Cre/+;Isl2-DTA/+* and control *Isl2-DTA/+* embryos, *Chat-Cre/+;R26-DTA/+* and control *R26-DTA/+* embryos, or *Olig2-Cre/+; R26R-DTA/+* and control *R26R-DTA/+* embryos. All tracing experiments using *Plp1-CreERT2; R26R-YFP* were performed using

heterozygotes for both the Cre and the reporter *R26R-YFP* or *R26-Tomato*. Sample size was determined by availability of mutant embryos from each litter. Images shown in the figures represent comparisons between at least three mutant and three control embryos, for each developmental stage from E10.5-E18.5.

## Immunohistochemistry

Embryos were harvested and fixed 4–6 h or overnight using 4% paraformaldehyde dissolved in a PBS buffer (pH 7.4) at 4 °C. Samples were washed in PBS at 4 °C for 1 h and cryopreserved by submerging at 4 °C for 6–24 h in 30% sucrose, diluted in PBS. The samples were then embedded using OCT media, frozen on dry ice, and stored at −20 °C. Tissue blocks were sectioned on an NX70 cryostat at a section thickness of 14–40 μm. Slides were stored at −20 °C after drying at RT for 1 h. For antigen retrieval, slides were immersed in 1x Target Retrieval Solution (Dako, S1699) for 1 h, pre-heated to 90 °C. Sections were washed three times in PBS containing 0.1% Tween-20 (PBST), incubated at 4 °C overnight with primary antibodies diluted in PBST in a humidified chamber. Finally, sections were washed in PBST and incubated with secondary antibodies and Hoechst stain diluted in PBST at RT for 2 h, washed again three times in PBST, and mounted using Mowiol mounting medium (Dako, #S3023). Rabbit polyclonal anti-TH (1:800, Pel-Freez Biologicals, #P40101-150, RRID:AB_2617184), sheep polyclonal anti-TH (1:2000, Novus Biologicals, #NB300-110), chicken polyclonal anti-TH (1:500, Abcam, #ab76442, RRID:AB_1524535), rabbit polyclonal anti-Hb9 (1:8000, gift from Samuel Pfaff's laboratory[61]), mouse monoclonal anti-bIII tubulin/TUJ1 (1:500, Promega, #G712A), mouse monoclonal anti-bIII tubulin/TUJ1 (1:1000, Abcam #ab7751, clone TU-20, RRID:AB_306045), rabbit polyclonal anti-bIII tubulin/TUJ1 (1:1000, Synaptic Systems, cat#302302, RRID:AB_10637424), chicken polyclonal anti-GFP (1:500, Aves Labs Inc., #GFP-1020, RRID:AB_10000240), chicken polyclonal anti-GFP (1:1000, Abcam, ab13970, polyclonal, RRID:AB_300798), rabbit polyclonal anti-GFP (1:5000, Thermo/LifeTech, #A6455, lot#2126798, RRID:AB_221570), goat polyclonal anti-PHOX2B (R&D, 1:1000, #AF4940), mouse monoclonal anti-Neurofilament/NF200 (1:200, Developmental Hybridoma Studies Bank, clone 2H3), goat polyclonal anti-SOX10 (1:500, Santa-Cruz, #sc-17342), rabbit monoclonal anti-SOX10 (1:2000, Abcam, # ab155279, clone EPR4007, RRID:AB_2650603), goat anti-human SOX10 (1:800, R&D Systems, #AF2864, RRID:AB_442208), rabbit polyclonal anti-PRPH (1:500, Chemicon, #AB1530, RRID:AB_90725), rabbit monoclonal anti-KI67 (1:500, Thermo Scientific, #RM-9106, clone SP6, RRID:AB_2341197), rat monoclonal anti-PECAM (1:300, BD Pharmingen, #553370, RRID:AB_394816), goat polyclonal anti-PECAM (1:300, R&D systems #AF3628, RRID:AB_2161028), rabbit monoclonal anti-ITGA4/CD49d (1:500, Invitrogen, #MA5-27947, clone RM268, RRID:AB_2744984), rabbit monoclonal anti-Cleaved Caspase 3/Asp175 (1:500, Cell signaling #96645, clone 5A1E), rabbit polyclonal anti-TrkA (1:500, # 06-574 Sigma-Aldrich, RRID:AB_310180). DAPI (Thermo Fisher Scientific, 1:10,000, #D1306) was used concomitantly with secondary antibodies diluted in PBST buffer. For detection of the primary antibodies, secondary antibodies raised in donkey and conjugated with Alexa-405, −488, −555, and −647 fluorophores were used (1:1000, Molecular Probes, Thermo Fisher Scientific).

## In situ hybridization

For HCR in Fig. S10, probe for Sema3C was ordered directly from Molecular Instruments, and the procedure was performed according to the protocol described online (https://files.molecularinstruments.com/MI-Protocol-RNAFISH-FrozenTissue-Rev3.pdf) using tissue cryosections from the trunks of day 11.5 mouse embryo. Briefly, slides were pre-treated via a fixation in paraformaldehyde at 4 °C, dehydration with an ethanol gradient, and a 10 μg/μL proteinase K digestion for 10 min before transcript detection. Then, slides were hybridized to 0.4 pmol probes diluted in 100 μL probe hybridization buffer (Molecular Instruments) at 37 °C overnight. After unbound probe was washed away with probe

wash buffer (Molecular Instruments) the bound probes were amplified using snap-cooled H1 and H2 hairpins overnight in a dark humidified chamber. After washing excess amplification solution away using SSCT, slides were mounted and imaged on a Zeiss LSM980-Airyscan confocal microscope. For traditional in situ hybridization, embryos were fixed overnight in cold 4% paraformaldehyde/PBS, washed in PBS, dehydrated in methanol, and stored at −20 °C. Then, in situ hybridization was performed using digoxigenin-labeled riboprobes transcribed from plasmids containing *Sema3a* and *Sema3f* cDNAs.

## Whole mount immunostaining of mouse embryos

Fixed embryos were dehydrated with a methanol gradient, bleached overnight at 4 °C using Dent's Bleach (20% dimethyl sulfoxide in methanol, mixed 2:1 with 30% hydrogen peroxide), incubated overnight in Dent's fix (20% dimethyl sulfoxide in methanol) at 4 °C and stored at −20 °C. Antibody dilution was done in a blocking buffer containing 5% normal donkey serum and 20% dimethyl sulfoxide. Embryos were incubated with primary antibody solution for 6–7 days, and incubated with secondary antibody solution for 2–3 days. After incubation of embryos with secondary antibodies and washing steps, the embryos were optically cleared using two different methods according to their size. Embryos at or younger than E11.5 were cleared in BABB solution (1 part benzyl alcohol / 2 parts benzyl benzoate) for 1 h with rotation before imaging on a confocal LSM800 microscope[8]. Embryos at or older than E12.5 were imaged using the CUBIC method (3–7 days in CUBIC1 at 37 degrees, 2–3 days in CUBIC2 solution at room temperature) before imaging on a Zeiss Z1 light sheet microscope[62]. 3D reconstructions of the sympathetic ganglia were performed using IMARIS software (version 9.5, Bitplane) based on segmentations of the TH/PHOX2B staining. In some cases, autofluorescence was used to aid the identification of the vasculature, the outline of the embryo, and the neural tube.

## Microscopy

Images were acquired using LSM800 Zeiss confocal microscope, Olympus FLUOVIEW FV3000RS Confocal, or Zeiss Z1 light sheet microscope. Confocal microscope was equipped with 10x/0.45, 20x/0.8, 40x/1.2 and 63x objectives. Laser lines used for excitation included 405 nm, 488 nm, 561 nm, and 640 nm. All images using the light sheet were taken with 5X/0.16 air objective, using 405 nm, 488 nm, 561 nm, and 638 nm for excitation. Light sheet images were acquired in the .czi format in Zen (Black edition, version 3) and processed by stitching with Arivis Vision 4D (Zeiss, version 4.0), down-sampling (1:2 in the XY plane) in FIJI (ImageJ, version 2.14.0), conversion to .ims files using Imaris File Converter (Bitplane, version 9.5), and downstream analysis with Imaris (Bitplane, version 9.5). Confocal images were acquired in .czi format and analyzed in FIJI.

## Image analysis of *Plp1-Cre^{ERT2}* embryos

Quantification of lineage tracing experiments was performed using FIJI (ImageJ, version 2.14.0). Multicolor, multi-tile, Z-stack images in CZI format with channels for DAPI, TH, and YFP were converted into 8-bit, and all Z-slices were combined using a maximum intensity projection. Binarization of images was based on manual thresholding per image, after it was determined that a fixed threshold per embryo gave similarly trending results across the anteroposterior axis. Image calculation in FIJI was used to find single, double, and triple positive regions. The ratio between the amount of triple positive YFP+ TH+ DAPI+ objects and the amount of double positive TH+ DAPI+ objects was used to determine the percent of traced neurons in the sympathetic ganglia. Over 1000 cells were counted per axial region per embryo, and the graphical results are from the averages of four analyzed embryos from two different litters.

## Image analysis of sympathetic ganglia size

Volumetric quantification of pelvic ganglia was performed using Imaris (version 9.5, Bitplane) with the surface generation tool and a fixed

intensity threshold for all samples. Volumetric quantification of sympathetic chain volumes was performed using Imaris (version 9.5, Bitplane) using the surface generating tool. 5 to 7.5 micrometer smoothing was used for all light sheet images, 1.25 micrometer smoothing was used for confocal images. Manual segmentation was used to separate cervical ganglia, sympathetic chain, adrenals, and ectopic micro-ganglia before surface generation. Signal intensity thresholds, which were automatically recommended by the Imaris software for each whole unsegmented image, were used for automatic surface generation. Volumes of right and left sympathetic chain ganglia were averaged.

### Boundary cap lineage tracing single-cell sequencing study

*Krox20-Cre; R26-Tomato* and *Prss56-Cre; R26-Tomato* mouse strains were used for lineage tracing boundary cap cells. For single-cell analysis at E11.5- E12.5, embryos were dissected to separate the meninges with all DRGs and ventral and dorsal roots. Specifically, E11.5 and 12.5 *Egr2Cre/+; R26-Tomato and Prss56cre/+; R26-Tomato* embryos were identified on the basis of Tomato expression under fluorescent stereomicroscope (Leica, Nussloch, Germany). Meninges and DRGs with dorsal and ventral roots were dissected and then digested with collagenase/dispase type I (Merck/Roche) for 15 min at 37 °C. Digestion was stopped by addition of 0.1 ml of fetal calf serum. Samples were slowly mechanically dissociated, and the cell suspension was filtered. Dissociated cells were then resuspended in PBS, 1% BSA, and subjected to FACS. Tomato-positive cells were isolated, while dead cells and doublets were excluded by gating on a forward-scatter and side-scatter area versus width. Log RFP fluorescence was acquired through a 530/30 nm bandpass. Internal Tomato-negative cells served as negative controls for FACS gating. Tomato-positive cells were sorted directly into PBS, 0.04% BSA for scRNA-seq experiments. Around 5,000 cells were loaded into one channel of the Chromium system using the V2 single-cell reagent kit (10X Genomics). Following capture and lysis, cDNAs were synthesized, then amplified by PCR for 12 cycles as per the manufacturer's protocol (10X Genomics). The amplified cDNAs were used to generate Illumina sequencing libraries that were each sequenced on one flow cell NextSeq500 Illumina.

### Single cell transcriptomics pre-processing and analysis

10x Genomics Cell Ranger v7.0.0[63] was used to process raw sequencing data. This pipeline converted Illumina base call files into Fastq format, aligned sequencing reads to a mm39 transcriptome using the STAR aligner[64], and quantified the expression of transcripts in each cell using Chromium barcodes. The Cell Ranger outputs were given to the velocyto.py pipeline (version 0.17.17)[65] to generate spliced/unspliced expression matrices further used for RNA velocity estimation. Scanpy package pipeline (version 1.9.3)[66] was used for the downstream analysis. To retain only high-quality cells, we filtered out the cells with high mitochondrial content (more than 10%); the cells with less than 2000 UMIs (1000 UMIs for Egr2_E12 dataset); and cells defined as putative doublets (with a doublet score equal to or greater than 0.2, calculated by Scrublet version 0.2.3)[67]. The filtered datasets first were analyzed separately to extract the cells belonging to BCC lineage and then integrated with Harmony (3000 highly variable genes, 30 principal components (PC), max number of iterations = 20)[68]. The new PCs adjusted by Harmony were used to compute a nearest neighbor graph with further clustering and embedding by the Leiden algorithm[69] and UMAP (Uniform Manifold Approximation and Projection), respectively. Cell types were identified based on the Leiden clusters and marker gene expression. For RNA velocity estimation, we used the *Scvelo* package (version 0.3.1)[70] with a dynamical model to learn the transcriptional dynamics of splicing kinetics.

### Cell-cell interaction analysis

Single cell transcriptomics datasets used for the study were downloaded from (https://github.com/LouisFaure/glialfates_paper) and ArrayExpress

accession: E-MTAB-10571. Standard pre-processing workflows such as QC, cell selection, data normalization, identification of highly variable features, scaling, dimensional reduction, clustering, and integration of the datasets was performed in R (version 4.3.2) using Seurat (version 5.0.3) according to the instructions provided (https://satijalab.org). Cell interaction analysis was performed using the R package CellChat (v 1.6.1). A subset of CellChatDB ("Secreted signaling") was used to seek cell-cell interactions between cell clusters in the merged motor-crest Seurat object (https://github.com/sqjin/CellChat). For specific code pipelines used for the analysis of single-cell data, please visit the GitHub link: https://github.com/ipoverennaya/motor_nerve_paper[71].

### Statistics and reproducibility

Statistical analysis was performed with GraphPad Prism (version 9.5.1) software. Description of statistical tests used can be found in each figure legend. No statistical method was used to predetermine sample size. No data were excluded from the analyses. The experiments were not randomized. The investigators were not blinded to allocation during experiments and outcome assessment. Analysis of covariation was performed in the case of determining the correlation between peripheral nerve length and features of neural crest migration using linear regression.

### Reporting summary

Further information on research design is available in the Nature Portfolio Reporting Summary linked to this article.

## Data availability

Source data are provided with this paper as a Source Data File. Information provided in the text, figures and supplementary information contained in the present manuscript is sufficient to assess whether the claims of this study are supported by the evidence. Neural crest single-cell transcriptomics datasets used for the cell interaction study are available from GEO via accession GSE201257 and are viewable from https://adameykolab.hifo.meduniwien.ac.at/cellxgene_public/. Motor neuron datasets can be found using the ArrayExpress accession: E-MTAB-10571. Dataset for the lineage tracing study has been submitted to GEO under accession code GSE261748. Raw microscopy files generated for this research project are available to interested parties upon request. Source data are provided with this paper.

## Code availability

The code used for single cell analysis, both tracing boundary cap cells and for predicting cell interactions between motor neurons and SCPs, can be found at the GitHub link: https://github.com/ipoverennaya/motor_nerve_paper[71].

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

## Acknowledgements

A.G.E. was supported by StratNeuro SRP Postdoctoral Research Fellowship and a Ruth L. Kirschstein National Research Service Award (F32/NRSA) from the National Institute of Dental and Craniofacial Research of the National Institute of Health (NIDCR/NIH) under award number 1F32DE029662. I.A. was supported by ERC Consolidator grant STEMMING-FROM-NERVE and ERC Synergy Grant KILL-OR-DIFFERENTIATE, Swedish Research Council, Knut and Alice Wallenberg Foundation, Bertil Hallsten Research Foundation, Cancerfonden, Paradifference Foundation, Austrian Science Fund (FWF) Stand-Alone grants, Austrian Science Fund (FWF) SFB F78 consortium grant, and the Austrian Science Fund (FWF) Emerging Fields "Brain Resilience" consortium grant. D.B. was supported by European Research Council Starting Grant 335590 and a Career Development Award from the Giovanni Armenise-Harvard Foundation. I.P. was supported by the European Union's Horizon 2020 Research and Innovation Program under Marie Sklodowska-Curie (grant agreement No. 860635, ITN NEUcrest). C.R. was supported by Wellcome Investigator Award 205099/Z/16/Z. We graciously thank Hjalmar Brismar and Hans Blom from the national Advanced Light Microscopy unit in SciLifeLab for permission to use the light sheet microscope. We thank the advanced microscopy laboratory (ALEMBIC) of San Raffaele Hospital for expertise and instrumentation. Finally, we give thanks to Emma Erickson for the lovely illustrations, and to Hanna Helene Daryapeyma for technical assistance.

## Author contributions

A.G.E., I.A., D.B., A.M. [Motta] conceived the study. A.G.E., A.M. [Motta], M.E.K., F.C., E.T., A.M. [Murtazina], G.C., and Q.S. performed experiments. A.G.E., A.M. [Motta], E.T., I.P., D.W., J.G., F.L., S.H., C.R., V.C., D.B., I.A. analyzed data. M.E.K., S.E., F.C., E.T., G.C., F.L., P.T., S.H., K.F., C.R., Q.S., V.C. provided experimental support and materials. D.B., K.F., P.T., and I.A. funded the project and supervised the study. A.G.E., A.M. [Motta], D.B., and I.A. wrote the initial draft of the manuscript; all other authors contributed to writing and approved the final version.

## Funding

## Competing interests

The authors declare no competing interests.
