## [Peer Review File · Nature Communications]

Motor innervation directs the correct development of the mouse sympathetic nervous systemREVIEWER COMMENTS

Reviewer #1 (Remarks to the Author):

In this manuscript, Erickson et al., selectively ablate motor neurons during embryogenesis and study the impact on sympathetic nervous system (SNS) development. The authors find that in the absence of motor neurons, neural crest cells do not migrate as expected and structures do not form properly during organogenesis. This is shown using several different genetic mouse models and at several different embryonic time points, imaging, data measurements, and single cell analysis.

The findings are very interesting, and a relevant addition to the enigmatic field of neural crest development. Methodology is solid and experiments well executed. Immunofluorescent experiments and images are beautiful, clear, and interpretation well supported. It will be very interesting to see which potential impact loss of motor neuron-regulated neural crest development has on terminal embryogenesis, viability and disease. This is beyond the scope of this project, but the manuscript could benefit even more from including a short discussion on the potential biological implications, see also point raised below.

I have many comments, but these are mainly discussion points and errors in figures/legends/corresponding text. I therefore consider this manuscript acceptable with minor revisions.

Comments:

Related to Fig. 1: Since migration is not affected, how come these projections form? What kind of property of the cell does not regulate migration, but growth? It seems that there are more cells in mutant mice? And/or bigger cells? Interpret in terms of individual cell morphology.

Fig. 1: Scale bars for insets in panel B are missing.

Legend for Fig. 1: "(C-E) ...in E12.5 embryos...". But 1C is denoted E11.5. This must be a mistake in the figure and legend, since 1E (denoted E12.5) is described as measurements from segmented portions of the volumes in C?

Legend for Fig. 1: There is no description of 1F.

Fig. 2C and E: In 2E, abbreviations NT, DRG and SG are spelled out. In this figure, only SG is used. All three abbreviations are however used in 2C, and should be spelled out in description text for this panel (or collect for all in the beginning/end).

Fig. 2D: In text, results are reported in μm , but in figure in #cells/axial level. Change to coherence.

Fig. 2F-G is mentioned/described before 2E in text.

Fig. 2F: Spell out abbreviation SC.

Fig. 2G: Huge difference in data points despite $n=3$ and $n=4$ embryos per group. Why?

Related to Supplementary Fig. 3A (second model): Why E14.5 time point? This is later than in the previous model. Explain and mention in text. For studies like this, investigating specific development events, it is important to take the developmental stages used into consideration to interpret the results.

Related to Fig. 3A: How does SOX10+ SCPs reach SG when both motor and sensory nerves are disconnected from SG (can it really be nerve/axon-free migration for such a far distance)?

Related to Fig. 3: Motor neuron ablation leads to that sensory neurons become thinner and further away from SG. Nerve thickness (i.e., here thinner) causes slower migration of SOX10+ SCPs. But migration is unaffected according to previous figures? How does this add up?

Fig. 3A: Spell out nt and sg in legend (or collect for all in the beginning/end).

Fig. 3C: Scale bar mentioned in legend but not present in actual images.

Figs. 2-3: In Fig. 2, 2H3 is used to show peripheral nerves, in Fig. 3 TUBIII is used for this. In Fig. 2 TUBIII is used to show sensory neurons. This is a bit confusing?

Fig. 4A: Stated that embryos from Fig. 1C-D are used, but see comment on this above (regarding E11.5 and E12.5 confusion. Here it is stated both ages, which does not add up with Fig. 1E).

Fig. 4A: Spell out abbreviations in legend (or collect for all in the beginning/end).

Fig. 4B: Scale bar only in upper left (WT). Are top and bottom images really the same magnification?

Fig. 4D: Not mentioned in the text.

Fig. 4D: I'm confused about what the scg+mcg quantification is?

Fig. 5A: Scale bars in insets? Are images from WT and R26R DTA the same magnification? drg is bigger in size in WT?

Fig. 5B: I'm confused about "Fraction of DRG/SG", is it images with aberrant organization in drg divided by images of aberrant organization in sg?

Fig. 5C: NT abbreviation not spelled out.

Supplementary Fig. 5A: How do the authors know that the SOX10+ cells along the peripheral nerves (2H3+) are exclusively SCPs?

Page 7: "Thus, ChatCre becomes active days later..." The number of days is missing.

Fig. 5C, Supplementary 5E: Schematics are unclear, e.g., what do the different colors represent? Schematics are extremely helpful and appreciated, but these need to be made much clearer and more instructive.

Fig. 6A: Spell out PHMC, MMC, PGC in legend.

Fig. 6B-C: What do the colors above the heatmap refer to? And what is the y-axis?

Fig. 7B-C: Scale bar in figure, but length not mentioned. Insets have no scale bars.

Fig. 7G-I: Insets have no scale bars.

Page 8, following Fig. 7D-I: The authors state that they did not observe any deficits in SG development in Olig2Cre-VEGF KO embryos, but do not provide any data ("data not shown"). The authors need to add (supplemental) figure on this, or remove the sentence.

Supplementary Fig. 2A legend: "Body length of the same E12.5 ..." The same embryos as in?

Supplementary Fig. 2E legend: "Data not shown". Remove.

Supplementary Fig. 2E: Why images from E10.5 (A) and measurements from E12.5 (B)?

Supplementary Fig. 5A legend: Missing E in E10.5.

Supplementary Fig. 5D legend: Symbol should be color

Supplementary Fig. 5E legend: Spell out abbreviation (or collect for all in the beginning/end).

Supplementary Fig. 8A: Abbreviations are not in A (move or collect for all in the beginning/end).

Supplementary Fig. 8: Length of scales not described in legend or image.

Discussion, first paragraph: Although mentioned in the results section, I recommend to include a sentence stating that whether these effects would be noticed also when sensory neurons are selectively ablated remains to be seen. This does not weaken or undermine any of the data presented here, but since sensory neurons are obviously an important player in these processes, they should be addressed in this context.

The authors examine embryos at E18.5, which is fairly late in development. Was there any phenotype observed, apart from SG/AG morphology? What will the biological consequences be? Are these embryos postnatally viable? Experiments to elucidate the resulting biology of later development are beyond scope and not required for this manuscript, but should be addressed. Could this have implications for NC-derived diseases for example?

Methods:

Image analysis of lineage tracing section includes "data not shown". This should be removed.

Otherwise, the Methods section is written in a structured and sufficiently detailed manner.

Reviewer #2 (Remarks to the Author):

The manuscript entitled "Motor nerves direct the development of the sympathetic nervous system" by Erickson et al. addresses the interesting and timely topic of how developing axons may direct cell migration and autonomous ganglion assembly, thus coordinating peripheral extension and guidance with the assembly of future synaptic targets by the very same axons. Using genetic tools, the authors generated mouse embryos lacking CNS motoneurons, including preganglionic motoneurons normally innervating the sympathetic chain ganglia (SCGs). They found that ablating motoneuron axons led to defective assembly of SCGs, misplacement of SCG neuron somas and aberrant SCG axon outgrowth, with similar defects in cervical sympathetic ganglia. They found that during normal development, a subset of PHOX2B-expressing Schwann cell precursors migrated along motoneuron axons towards the SCGs, which had previously aggregated in an apparently axon-independent manner from earlier migrating neural crest cells, suggesting that the observed defects were due to aberrant migration of the late SCP-derived SCG progenitor population. The authors further found that, without motoneuron axons, SCG cells aberrantly intermingled with dorsal root ganglion cells (DRGs). Next, the authors used previously published single-cell data to search for potential signaling molecules that could underlie the interactions between SCG progenitors and motoneuron axons, which suggested a potential involvement of the Semaphorin signaling pathway. Lastly, the authors analyzed mouse embryos lacking both Semaphorins 3A and 3F, as well as embryos lacking functional Semaphorin receptors Nrp1 and Nrp2. They found defects in SCG assembly and displacements of SCG neurons in both mutant models and based on these results proposed that *Sema3A/3F-Nrp1/Nrp2* signaling underlies the motoneuron axon-dependent migration and aggregation of SCP-derived SCG progenitors into defined SCGs.

General comments

Overall, I found the study interesting, in principle technically sound, but lacking in substance. The importance of motoneuron axons for the proper developmental assembly of the SCGs becomes very clear. However, the study comes short of offering a convincing explanation accounting for the observed defects in SCG assembly in the absence of motoneuron axons and vice versa, they do not come up with a good model of what the role of the interactions between motoneuron axons and SCG neurons and axons would be during normal development. The developmental events leading to faulty SCG and cervical ganglion assembly in the absence of motoneuron axons is not at all well documented, beyond a description of some late-migrating SCP-derived SCG progenitors apparently associate with growing preganglionic motoneuron axons. But how does the absence of the latter exactly lead to misinformed SCGs? Likewise, besides a relatively superficial description of SCG defects in the Semaphorin and Neuropilin mutant embryos, the authors do not come up with a good model of what these molecules could actually be doing during the migration of SCGs along preganglionic motoneuron axons. They largely ignore a host of previously published data that could offer a range of alternative explanations for the defects observed in the motoneuron-less embryos, and particularly in the *Sema* and *Nrp* mutants. These include (but are by far not limited to) the long-established roles of somitic Semaphorins in channeling neural crest cell migration or the previously reported impacts of *Nrp1/2* or motoneuron-removal on sensory axons, which in turn

could indirectly affect SCG neurons and axons. Thus, several of the SCG assembly defects observed in these mutants could be entirely unrelated to motoneuron axon-SCG interactions, a possibility that is very briefly mentioned in the discussion, albeit buried within. In short, I find the topic of the study interesting and the data presented in principle of good quality, but the findings of the manuscript are too descriptive and preliminary to recommend publication in Nature Communications at this stage.

Specific comments

1. Page 6, paragraph "Ventrally migrating neural crest cells are drawn to motor axons exiting the spinal cord". The paragraph and the Supplementary Figures (5A-E) mentioned actually do not contain any evidence supporting the statement in the paragraph heading of ventral neural crest cells being "drawn" to motoneuron axons, which is quite confusing.
3. Motoaxons have long been thought to form a lattice for the extension of later-developing sensory axons (e.g. Honig et al. Dev. Biol. 118, 532). While the authors acknowledge the latter (page 10, last sentence), they generally fail to consider the extent to which these defects depend on direct interactions of SCGs with motoaxons (or the lack thereof) or the indirect motor axon-dependent misrouting of sensory axons on which in turn SCGs and axons in turn might depend (the same goes for: Page 5, paragraph heading: "Motor ablation causes misrouting of sensory nerves away from sympathetic ganglia").
4. Throughout the manuscript claims are made about "repulsive navigational cues" or cells being "drawn to" motor neuron axons or the axons "sponge" neural crest cells (Page 6, paragraph 2, last sentence), without any direct evidence for the underlying cellular interactions. For instance, would neural crest cells indeed be attracted by motoaxons and are the latter indeed repulsive for SCGs? A possibility to more directly dissect such interactions could be through relatively straightforward in vitro co-culture assays.
5. Figure 7: The authors provide only a brief documentation of trunk SCG developmental defects in *Sema* and *Nrp* mutants (Figure 7), without providing information of other aspects disrupted by motoneuron axon-ablation, such as the intermingling of DGRs and SCGs, the SCG axon defects or the defects in the cervical sympathetic ganglia. Moreover, the different markers and the assigning of different colors to markers, as well as the lack of comparable wholemount images for the *Sema/Nrp* mutants makes it altogether difficult for the reader to compare these defects with those obtained by ablating motoneurons (Figures 1-4) and to appreciate the extent to which they are indeed similar as claimed (Page 9, first paragraph). One has the impression that both data sets (obtained by motoneuron ablation, *Sema/Nrp1* knockouts, respectively) were obtained by two different research groups with little prior coordination. All in all, these data are too preliminary for the authors to convincingly make their case that the *Sema/Nrp* signaling between preganglionic motoneuron axons fully account for the SCG assembly and axon defects observed upon motoneuron ablation.

Reviewer #3 (Remarks to the Author):

This manuscript describes an involvement of motor neurons in shaping the developing sympathetic ganglia via schwann cell precursor (SCP) contribution after the sympathetic ganglia have initially been formed by the migrating neural crest cells. The finding is relevant as sympathetic ganglia are thought to arise solely from the migrating, primary neural crest cells with no input from SCPs. The manuscript is based on high resolution confocal images from several mouse lines. However, the manuscript is not well written and it is very hard to follow; findings throughout the text and figure legends are poorly explained to the point that it's difficult to judge the data. To make better sense of the results and the phenotype caused by motor-nerve depletion, the novel finding of SCP contribution to the sympathetic ganglia in normal development should be described in a much more detailed manner. None of the experiments describe the sequence of WT events in a satisfactory manner, which leaves open questions: at what stage do the SCPs start moving towards the sympathetic ganglia, what markers do they express that differentiate them from glial cells, do all the cells migrate along the sensory nerves that are guided by the presence of motor nerves, or do some also migrate along the motor nerves (the results seem contradictory on this topic in the text)? It also remains unclear if all the migrating primary neural crest cells that populate the sympathetic ganglia migrate around the sensory ganglia or do some go through them, and at what stage do they become SCPs and stop being migrating crest (as the manuscript does occasionally refer to the cells as migrating crest) and when can they be called sensory glial cells (the lab has just published on this topic so marking these different SOX10+ populations is a

realistic request). The data itself also raises several concerns, some of which may be due to misunderstanding from my part caused by the poor communication of the results. Overall, the work in its current form leaves a lot of open questions and seems premature.

Specific concerns:

Introduction:

Paragraphs 2 and 3 are redundant.

Images and explanations

The manuscript is written to a very limited audience that is assumed to have full knowledge on the literature and the detailed stage by stage peripheral nervous system development and anatomy in mouse embryos, which does not serve the readership of any journal. Importantly, since this work is solely based on imaging data, every microscope images need an explanatory cartoon with an orientation of where the image was taken from in the embryo and what are the surrounding structures are to guide the reader and to show how the WT embryo forms as compared to the mutants. The labels need to be significantly improved, one can not say 1) "SCPs found at close vicinity of the dorsal aorta" without marking the DA in the images, or 2) talk about TH+ and 2H3+ cells without first explaining why the markers were used and thus what those cells are, or 3) talk about Sox10 -positive cells without mentioning what they mark in the specific context (the Neural crest, SCPs or glial cells), or 4) talk about nerve fibers without mentioning which nerves are authors referring to (visceral motor, sensory or sympathetic?). All these small deficiencies and unmarked structures in the figures make the manuscript exhausting to follow. In sum, the text including figures and figure legends need to be re-written with a systematically more explanatory style.

Figure 1

- Fig 1A,B: The results show the main phenotype, constellation of prominent ectopic TH positive cells along the entire trunk. The WT embryo has no TH positive cells projecting from the sympathetic chain, which is inconsistent with previous findings that do show these branches in WT (PMID: 21325504), which raises a concern. Please address the difference between the published work and your data.

Are the WT and DTA embryos images from the same focal level and do both images contain the same amount of stacks in the maximum projection? The WT embryo images have less 2H3 positive nerve fibers than the DTA counterpart (indicating the DTA image consists of either more z-stacks or same the amount but captured from a different focal plane) and seem to have been imaged from the focal plane of the sensory ganglia, whereas the ganglia are out of focus in the DTA – are these differences shown in the images real or caused by technical imaging discrepancy? Similarly, the fluorescence intensity of the sympathetic chain in fig 1B is much lower in the WT images as compared to DTA, why?

Finally, I was not sure how the quantifications were done to get a result of the general TH volume to be lower in DTA, that does not match with what the images show. Additionally, as the sympathetic chain ganglia get more fragmented over developmental time, the range of developmental stages within a timepoint, (even amongst littermates) can be broad. The number of embryos is only 3 or 4, respectively – were the developmental stages aligned by additional measures like somite counting? The imaging discrepancies need to be addressed before conclusions can be made in a convincing manner.

- Figure legend for 1f is missing

Figure 2

As mentioned above, the authors should mark the structures that are discussed and be specific.

For example: "Sympathetic somas" on "which" nerve?

Why are the sg not TubIII-positive? Please show the channels separately in addition to the merged figures.

Why don't the authors show WT images of the normal SCP contribution into the sg via the motoneurons/ motoneuron guided-sensory nerves? If TH is not expressed in the motor nerve associated SCPs and only comes on when they reach the condensed sg, please use another, SCP-specific marker to show the cells and explain this clearly in the text. Or is Sox10 marking also glial cells here, the use of Sox10 for three different purposes is very confusing and requires an additional marker to separate between the stages (NC, SCP, Glia)? Also please verify that the difference between the condensation of the sympathetic ganglia between WT and DTA is not due to analysis at a later developmental stage.

In the second DTA panel in 2E, what is the tubIII-positive nerve-like looking structure below the sensory ganglia originating from the neural tube, which contains some TH positive cells, is the motor nerve ablation not 100%?

Figure 3:

- Please add additional data and cartoons to demonstrate the step by step sequence of events during normal development in order for readers to understand what is the difference as compared to the DTA phenotype. (also B and C should be introduced before going into the DTA phenotype in more detail.) Double staining of Phox2b and TH would be essential to combine information from figures 2 and 3.
- Please use additional markers to differentiate between SCP and glial cells and the migratory neural crest.
- Can the SCP-derived portion of the sympathetic ganglia be quantitatively shown to be missing from the motor-nerve-ablated mice? (by using specific genetic reporter lines or by photoconversion or by an onset of a Cre-reporter, or an injected fluorescent dye (which may not be possible to correctly target in utero))
- Please clarify what the authors mean by the following sentence on page 5: "Although this experiment lacks a sensory-ablated control to tease out a specific role for sensory fibers, the current data suggests that the misplaced sympathetic neurons might result from the migration of neural crest and SCPs along improperly positioned sensory axons." Is the migration of primary neural crest cells also dependent on the guidance of the sensory nerves? Which data supports this and how can those cells be separated from the SCPs this manuscript is afocused on?
- 3A: What does "next to sg" mean? Why are there TH positive cells in the neural tube? Please show the TH channel separately for better interpretation of the data. Why are there no TH positive cells on the nerve approaching the WT sg? Can the authors separate, by using specific markers, the motor and sensory axons and show which ones are used by the SCPs? Please also use specific markers to separate the SCPs and glial cells that are associated with the nerves (the text uses both terms); this part is very confusing.
- 3B: why are the Phox2B+ cells called sox10/Phox2b double positive cells – were the double positive cells selected by using Imaris or an equivalent image analysis software?
- 3B and C: can the authors add a fourth channel to use specific markers to separate between motor and sensory axons? Similarly, add a fourth channel for TH to separate between Phox2b and TH. Please show the Sox10 channel alone to show whether it overlaps with the Phox2b.
- 3C: Why are there so many (in proportion) more sox10+ cells as compared to Phox2b positive cells in the sacral ganglia, I'm not following why the sox10+ SCPs on the tip of the nerves (in the text now referred to as motor nerves and not sensory) in the sacral axial level are not Phox2B positive (and most cells in the sg are Phox2b negative as well) while Phox2b is turned on in the equivalent cells along the nerves in the more anterior images. If this demonstrates that sacral cells entering the sg are still primary neural crest cells, why are they also using the motor axons for guidance? Please clarify.

Figure 4

- Fig A raises the same central concern regarding the focal plane of the imaging (similar to fig 1). Please explain why the images don't show TH projections into the drg in the WT. In the fig A E12.5 DTA image, traces of the DRG are clearly visible in the background whereas they are not shown in the WT. This systematic discrepancy in the imaging is concerning.
- A misshape analysis in 3D (trace the ganglia shape, overlay them and calculate the differences) would define and quantify the results better, as the provided size calculation gives no significant change. The n only equals 2 in some cases (n should be increased).

Figure 5

- In the text the authors say: "Unexpectedly, we found that in mutant embryos sympathoblasts expressing high levels of TH (in contrast to typical sparse, low-TH sensory neurons in normal DRG) were inappropriately located around and within the dorsal root ganglia and, conversely, that sensory neurons were misplaced in the sympathetic ganglia (Figure 5A)." Looking at the images, the authors may want to soften their statement to concluding that sensory somas are found in the symphatetic ganglia, whereas the evidende for the presence of TH -positive cell bodies in the drg is not convincing as the conclusion relies on the subjective interpretation on where the sg ends and the drg starts, and some TH positive cells are also found in the WT drg.

Figure 6

- The RNAseq data to show putative molecular signals seems far fetched and problematic. Importantly, the joint data sets are from significantly different developmental stages (neural crest is from E9.5 whereas the Amin et al data set is E12 (and not 10.5 as falsely stated in the text and figure legend). The search for putative receptor-ligand interactions between these data sets from completely different stage embryos does not provide a realistic base for a search on putative interactions.
- How were the five candidate pathways predicted to mediate motor neuron-to-neural crest cell signaling? Why was SEMA3 pathway considered the best candidate? Please explain

Figure 7

- The selection of the knockout models needs to be justified in much more detail to convince the reader on the motor neuron- SCP- sg connection. The knockout phenotypes are thus extremely preliminary and not convincing, and nothing is quantified. The in situ expression pattern of Sema3A and 3F don't per se support the hypothesis of involvement in the NC/SCP process. Furthermore, where are Nrp1 and 2 expressed at the time of the potential involvement in the motoneuron initiated guidance of migration? How did the authors rule out that the phenotypes are not a consequence of neural crest development/migration related defects much before the cells reach the stage of the motoneuron involvement stage of sg formation? In order to make credible claims, the expression patterns and phenotypes need to be studied in a much more detailed manner, which, if properly done, will require a significant amount of additional experiments.
- In sum, Figures 6 and 7 seem tangential to the main finding of the manuscript and the authors should consider removing them and instead focus on improving the data on the main points of the story to convincingly test their hypothesis.

Reviewer #1 (Remarks to the Author):

In this manuscript, Erickson et al., selectively ablate motor neurons during embryogenesis and study the impact on sympathetic nervous system (SNS) development. The authors find that in the absence of motor neurons, neural crest cells do not migrate as expected and structures do not form properly during organogenesis. This is shown using several different genetic mouse models and at several different embryonic time points, imaging, data measurements, and single cell analysis. The findings are very interesting, and a relevant addition to the enigmatic field of neural crest development. Methodology is solid and experiments well executed. Immunofluorescent experiments and images are beautiful, clear, and interpretation well supported. It will be very interesting to see which potential impact loss of motor neuron-regulated neural crest development has on terminal embryogenesis, viability and disease. This is beyond the scope of this project, but the manuscript could benefit even more from including a short discussion on the potential biological implications, see also point raised below.

Thank you for the kind comments. We have now included a discussion about the disease-related aspects of our findings. We believe that the effects of this neural-crest-related phenotype would be overshadowed by the complete loss of motor neurons, but this does have implications for diseases such as neuroblastoma, as it may take only a single mis-located primed autonomic glial progenitor cell to initiate a tumor, and short delays in white ramus communicans axonal navigation might already cause subtle variations of the Olig2Cre-DTA phenotype (a few misplaced cells). See the portion of the text:

“It would follow that a fine balance exists between gliogenic and neurogenic potential in individual SCPs, with local signaling cues regulating the probability of nerve-associated neurogenesis [10, 47]. Knowing how motor nerves regulate cell migration and neuroglial cell fate might have implications for neuroblastoma pathogenesis. Specifically, future studies of nerve-derived cues affecting sympathetic priming, especially functional differences between the sensory versus motor niches, might improve our understanding of cell origins of tumors initiated in non-canonical locations [48].”

Furthermore, aberrant sympathetic innervation of DRG can be a cause of idiopathic pain, and it seems that motor guidance might be relevant for preventing this. See the relevant discussion in the text:

“Ultimately, at later developmental stages, sympathetic and sensory ganglia started to intermingle, causing inappropriate mixing of different neuronal types. This aberrant configuration resembles or may even model human cases of sympathetic pain syndrome, where chronic pain arises from abnormal synaptic connections between sympathetic neurons and pain somatosensory neurons located in DRGs [49-51]. It would be interesting to test whether perturbation of specific motor-derived signals, or transient redirection of SCPs, is sufficient to recreate such syndromes in a mouse model. Altogether, our results point to a previously unappreciated restraining effect of preganglionic motor fibers on the projection of sympathetic nerves.”

I have many comments, but these are mainly discussion points and errors in figures/legends/corresponding text. I therefore consider this manuscript acceptable with minor revisions.

We have done our best to respond to each point. Thank you for the advice and guidance.

Comments:

Related to Fig. 1: Since migration is not affected, how come these projections form? What kind of property of the cell does not regulate migration, but growth? It seems that there are more cells in mutant mice? And/or bigger cells? Interpret in terms of individual cell morphology.

Following this comment, we examined carefully the development of sympathoblasts in the motor nerve ablated embryos, also by including additional whole mount staining experiments (updated panels in Figure 5). We consistently find that the misplaced sympathetic neurons start to project earlier and along abnormal paths. However, according to the reviewer's recommendation, we have quantified the overall neuronal soma size on sections (see updated Figure S4) within the ganglia and found no difference between mutants and controls.

Overall, we are convinced that most of the early neural crest migration is not affected because the detectable differences in sympathetic chain of mutants only come at E11.5, while neural crest migration in the trunk and the initial seeding of sympathetic anlagen is completed between E9.5-10. From E10.5, loss of motor innervation affects the late stream of nerve-assisted SCP migration to the ganglia, leading to the misplacement of sympathoblasts. This can give the impression of more cells because they are more spread out, but we actually detected a significant decrease in volume and cell numbers in the sympathetic chain (revised Figure 4e and 5j). However, we do observe cells priming at far-away locations in the limb, suggesting that some mis-placed cells can be converted into sympathoblasts inappropriately when motor nerves are not present to restrain this induction. This is the most plausible explanation because such phenotype cannot be explained by abnormal free neural crest migration based on the timing of the crest migrating and motor nerve outgrowing, and also because the mis-placed sympathetic cells are found in tight association with distant sensory nerves.

Fig. 1: Scale bars for insets in panel B are missing.

Scale bars have been added to all panels, including insets.

Legend for Fig. 1: "(C-E) ...in E12.5 embryos...". But 1C is denoted E11.5. This must be a mistake in the figure and legend, since 1E (denoted E12.5) is described as measurements from segmented portions of the volumes in C?

This was indeed a mistake and has been amended. This is now Figure 5j.

Legend for Fig. 1: There is no description of 1F.

We have added in the much-needed description of the panel, which after reorganizing the manuscript is now Figure 5m. Please see the description:

*“(M) Length of TH⁺ sympathetic axons innervating the forelimb in control littermates vs Olig2-Cre; DTA embryos. Mean ± SEM, unpaired t test (**) p<0.01; controls n=3, mutants n=4.”*

Fig. 2C and E: In 2E, abbreviations NT, DRG and SG are spelled out. In this figure, only SG is used. All three abbreviations are however used in 2C, and should be spelled out in description text for this panel (or collect for all in the beginning/end).

We have fixed this inconsistency. We collected all abbreviations at the end of figure legends.

Fig. 2D: In text, results are reported in μm , but in figure in #cells/axial level. Change to coherence.

We have rearranged the reporting of results so that the panel (now Figure 5e) reflects the statement in the text. Here is how we refer to this panel:

“Whole mount immunofluorescent staining revealed a constellation of ectopic TH⁺ cell clusters around the sympathetic chain throughout the thoracolumbar region (Fig 5c-e), that were numerous in motor-ablated embryos but virtually absent in controls (Fig 5f). These cells were found at a considerable distance from the sympathetic chain, extending as far as the limbs (Fig 5g).”

In the figure legend we describe this panel as such:

*“(E) Quantification of the number of observed misplaced sensory-nerve-attached sympathetic neurons per DRG at all axial levels from brachial to lumbar in Olig2-Cre; DTA versus control littermates (Mean \pm SEM, Unpaired t test (***) $p < 0.0001$; control $n = 3$ embryos, mutant $n = 3$ embryos).”*

Fig. 2F-G is mentioned/described before 2E in text.

In the revised manuscript, the order has been changed to match exactly across text and figures.

Fig. 2F: Spell out abbreviation SC.

SC was referring to “sympathetic chain”, but in the revised manuscript we removed this abbreviation, and the new title of the chart (updated Figure 5f) is: Amount of cell misplacement (E12.5).

Fig. 2G: Huge difference in data points despite $n = 3$ and $n = 4$ embryos per group. Why?

There is an abundance of misplaced cells in the DTA group but very few misplaced cells in the wildtype control embryos. Because of that, we have a lot of datapoints (technical n per embryo) for the DTA group but not for the control. The plot is now Figure 5g.

Related to Supplementary Fig. 3A (second model): Why E14.5 time point? This is later than in the previous model. Explain and mention in text. For studies like this, investigating specific development events, it is important to take the developmental stages used into consideration to interpret the results.

This slight incoherence of stages occurred because of a practical reason: the laboratory of Francois Lallemand, who provided these *Hb9-Cre; Isl2-DTA* embryos, had the mice in very low numbers so we were only able to examine E14.5 embryos. However, despite this discrepancy, both *Olig2Cre; DTA* and *Hb9-Cre; Isl2-DTA* displayed similar phenotypes, reinforcing our major finding. To respond to other comments, we had to expand our analysis with *Olig2* line towards intermediate stages prior to ganglia fusion and opted for E15.5 as good middle ground between E18.5 and E12.5-13.5. Because this revision already took over a year, and we did not have enough mice to perform analysis on every embryonic day, we decided that given the principal similarity of results and the ancillary position of *Hb9-Cre; Isl2-DTA* data in the context of the paper, we could still use this information to rule out model-specificity. We hope the reviewer doesn't take this practical side negatively.

Related to Fig. 3A: How does SOX10+ SCPs reach SG when both motor and sensory nerves are disconnected from SG (can it really be nerve/axon-free migration for such a far distance)?

The former figure 3A showed the situation directly after the end of the neural crest migration in control and nerve-ablated embryos at E10.5. We performed more staining and analysis during revision, and substituted this panel with Figure 4a and Figure S1 that specifically highlight the role of motor nerves, and show the whole progression from early neural crest migration that seeds the sympathetic ganglia to later waves of SCPs, which supplement and fuel ganglia growth. When motor and sensory nerves are disconnected from SG, Sox10⁺ SCPs do not reach SG, meaning that the only remaining cells in the SG are those that come from the initial wave of neural crest migration (i.e., the cells migrating prior to the outgrowth of motor nerves). We do not think axon-free migration of neural crest cells take place after they begin to associate with nerves becoming SCPs around E10.5 as supported by previous publications from our team and others (Adameyko et al Development 2012 PMID:22186729, Dyachuk et al Science 2014 PMID: 24925909). This important point has been clarified in the revised text:

“In the classical paradigm, migratory neural crest cells arrive at the dorsal aorta and coalesce into primary sympathetic ganglia [7]. We show that as motor axons emerge from the neural tube, they encounter part of the freely migrating neural crest, which becomes associated with growing motor nerves, turning into transcriptionally-defined SCPs – residing along the entire length of the peripheral nerve [43] – and BCCs – residing at CNS nerve exit points [22]. Consequently, the timing of motor axon outgrowth at each rostrocaudal segment coincides with a switch from free- to axon-associated dispersion of neural crest derivatives, consistent with our earlier studies [21].”

Related to Fig. 3: Motor neuron ablation leads to that sensory neurons become thinner and further away from SG. Nerve thickness (i.e., here thinner) causes slower migration of SOX10+ SCPs. But migration is unaffected according to previous figures? How does this add up?

We are grateful for this comment. We believe it would be important to state here that when we talk about “migration” we refer to free nerve-independent active neural crest migration. Therefore, the “unaffected migration” being referred to in previous figures is related only to early waves of nerve-independent neural crest migration that initially seed the sympathetic anlagen, which is not perturbed by motor neuron ablation.

When it comes to SCPs, they spread on the nerve surface via intense proliferation and only local tethering along the nerve length. Basically, they travel with the nerve on its surface. Therefore, invoking the different nerve thickness itself to explain changes in migration was a bit of a speculation on our part in the first version of the manuscript. In the revised manuscript, we have clarified our model with respect to the effects on neural crest migration or SCPs traveling with innervation. Indeed, motor neuron ablation leads to the peripheral nerves being thinner and further away from SG due to the fact that ventral white ramus communicans, the part of the ventral root leading to the SG, is almost entirely composed of motor fibers. The spared sensory fibers are misguided, and some fail to reach the SG. The loss of the white ramus leading to SG appears to be the major reason why eventual misplacement of sensory-associated SCP cells occurs. It appears that SCPs are able to travel significant distances along sensory nerves, but have trouble making it to the sympathetic chain ganglia in the absence of the mostly motor white ramus.

Fig. 3A: Spell out nt and sg in legend (or collect for all in the beginning/end).

Thanks for pointing this out. We have been more meticulous with our abbreviations in the revised manuscript.

Fig. 3C: Scale bar mentioned in legend but not present in actual images.

We have added scale bars throughout the panels systematically. We thank the reviewer for observing these details.

Figs. 2-3: In Fig. 2, 2H3 is used to show peripheral nerves, in Fig. 3 TUBIII is used for this. In Fig. 2 TUBIII is used to show sensory neurons. This is a bit confusing?

Both 2H3 (neurofilament) and TUBIII/TUJ1 antibodies are widely accepted pan-neuronal projection markers and provide a virtually identical view of these neuronal structures. Because we performed experiments at different institutions, we used these reagents interchangeably based on availability in the labs. We are confident that this does not affect any conclusions of the study. In former Figure 2, the peripheral nerves in a motor-ablated embryo are coming solely from sensory neurons, and therefore both TUBIII/TUJ1 or 2H3 would stain only sensory fibers in that setting. We don't include the panel from Figure 2C in the revised manuscript because, while performing new staining experiments, we found new panels that show the phenotype of sympathoblast misplacement more clearly (Figure 4f and g).

Fig. 4A: Stated that embryos from Fig. 1C-D are used, but see comment on this above (regarding E11.5 and E12.5 confusion. Here it is stated both ages, which does not add up with Fig. 1E).

We have resolved the issue.

Fig. 4A: Spell out abbreviations in legend (or collect for all in the beginning/end).

Fixed! Thanks.

Fig. 4B: Scale bar only in upper left (WT). Are top and bottom images really the same magnification?

Scale bars have been added to each group of panels (now Supplementary Figure 8b).

Fig. 4D: Not mentioned in the text.

This issue has been resolved (now Supplementary Figure 8e).

Fig. 4D: I'm confused about what the scg+mcg quantification is?

We have measured the volume of the superior cervical ganglia (scg) and medial cervical ganglia (mcg), by employing an automatic calculation using a surface generation in Bitplane-Imaris image analysis software, applied to our whole mount fluorescence images immunostained for TH to label sympathetic ganglia. In *Olig2-Cre; DTA*, the SCG appears smaller in size, while the MCG –which is normally a very tiny ganglia– becomes enlarged in mutants. However, we noticed that the sum of the volumes of these two structures (“scg + mcg”) is not different between control and motor-ablated embryos. For this reason, it seems likely that the enlargement of mcg, and reduction of scg, in the motor-ablated embryos is caused by deficiency in motor nerve-associated collective cell movements that affect the distribution of sympathoblasts in the two ganglia.

Fig. 5A: Scale bars in insets? Are images from WT and R26R DTA the same magnification? drg is bigger in size in WT?

The magnification is the same. During this revision, in response to this comment, we quantified DRG sizes in both WT and DTA. There is a significant decrease in the size of DRGs in the *Olig2-Cre; DTA* embryos at E18.5. We included this data in Figure 6e. Additionally, we have added the appropriate scale bars to each inset.

Fig. 5B: I'm confused about “Fraction of DRG/SG”, is it images with aberrant organization in drg divided by images of aberrant organization in sg?

We apologize for the unclear labeling. In the previous version of the manuscript, this label was supposed to refer to the number of ganglia pairs (meaning a DRG and the corresponding region of SG at the same axial level) observed with aberrant organization, divided by the total ganglia pairs analyzed. We changed label to avoid confusion, and hope the message of the panel is clearer now (please see revised Figure 6d).

Fig. 5C: NT abbreviation not spelled out.

This has been added to the figure legend. Thanks for pointing out the error.

Supplementary Fig. 5A: How do the authors know that the SOX10+ cells along the peripheral nerves (2H3+) are exclusively SCPs?

When the neural crest cells begin associating to nerves, they are henceforth referred to as SCPs according to the canonical and historical definition. However, we addressed this question in our previous manuscript by Kastri et al 2022 (PMID: 35815410), where we showed that the phenotypic transition from free migrating neural crest cells into nerve-associated state correlates with gradual transcriptional changes that progressively define the molecular signatures of SCPs. The association to peripheral nerves coincides with the switch from free NCCs to the SCP state, which is highlighted by the gradual upregulation of a defined set of genes, including *Itga4*. We have added some clarifying text about this in the manuscript:

“While SCPs were previously distinguished from NCCs and ganglia-residing satellite glia solely on the basis of their association with nerves, a SCP-specific gene signature referred to as a “hub state” marked by Itga4 expression has been recently described [19]. Indeed, at E10.5 ITGA4 levels were greater in nerve-associated SOX10+ cells compared to NCCs that were still freely migrating in the caudal region (Fig 1f), suggesting that nerve association coincided with the adoption of the SCP hub state.”

Based on ITGA4 staining performed during the revision (Figure 1f), we show that nerve-associated SCPs express ITGA4, while free-migrating NCCs rapidly downregulate ITGA4 after delamination from the dorsal neural tube. Therefore, this new result indicates that the cells at the peripheral nerves are SCPs and not “touching-the-nerve migrating neural crest”.

Also, in the revision we determined the effect of motor nerve outgrowth on the neural crest migration, showing how neural crest streams are interrupted, diverted, and consolidated by the growing motor nerve. We find that this is largely due to the nerve-association of these cells and their transition into SCP state (Figure 1b-e and Figure S1a). From the reconstructed sequence of events, we can say that NCC transition to SCP involves attachment to nerves, apparent alterations in the migratory behavior, and ITGA4 expression.

Page 7: “Thus, ChatCre becomes active days later...” The number of days is missing.

This has been now clarified:

“Since these defects developed at later stages than those observed in the sympathetic chain, we used Chat-Cre allele to drive R26-DTA in postmitotic motor neurons from E11.5 (as opposed to Olig2-Cre that is expressed in progenitors)”.

Fig. 5C, Supplementary 5E: Schematics are unclear, e.g., what do the different colors represent? Schematics are extremely helpful and appreciated, but these need to be made much clearer and more instructive.

In the revision, we improved significantly the schematics, and included more of them to illustrate the models and conclusions. The color code is used to distinguish the different types of neurons is now consistent across the manuscript.

Fig. 6A: Spell out PHMC, MMC, PGC in legend.

This has been fixed.

Fig. 6B-C: What do the colors above the heatmap refer to? And what is the y-axis?

This comment refers to CellChat analysis that is now in Supplementary Figure S9. During the revision, we improved the single-cell transcriptomic analysis using a more appropriate time-matched dataset including more relevant cell types. Colors on top of heatmaps (Figure S9b,c) match the clusters in the UMAP (Figure S9a). The Y-axis (the length of the bar in each bar graph) is the overall predicted activity of the signaling pathway across all cell types. We provided detailed description of the plots in figure legends.

Fig. 7B-C: Scale bar in figure, but length not mentioned. Insets have no scale bars.

The scale bars for all panels in Figure 7 measure 20 microns. For all figures, the lengths of scale bars are reported at the end of figure legend. The scale bars are now included for insets.

Fig. 7G-I: Insets have no scale bars.

The scale bars are now included for insets.

Page 8, following Fig. 7D-I: The authors state that they did not observe any deficits in SG development in Olig2Cre-VEGF KO embryos, but do not provide any data ("data not shown"). The authors need to add (supplemental) figure on this, or remove the sentence.

We removed the sentence. Since VEGF-derived from motor nerves doesn't impact sympathetic development, we decided this negative result is at best tangential to the primary findings of the paper.

Supplementary Fig. 2A legend: "Body length of the same E12.5 ..." The same embryos as in?

We used the images of whole mounts of embryos shown in former Figure 1 (updated Figure 5) to measure body length. The quantification is now in Supplementary Figure S4b along with an explanation in the corresponding figure legend.

Supplementary Fig. 2E legend: "Data not shown". Remove.

This has been removed.

Supplementary Fig. 2E: Why images from E10.5 (A) and measurements from E12.5 (B)?

New images for E12.5 have been included in the updated Figure 4b, showing how peripheral nerves mostly fail to reach the sympathetic ganglia when only sensory nerves remain (quantifications in Supplementary Figure S3j). Still, we decided to keep the E10.5 whole mounts because they clearly show the same effect but in 3D volumes (Supplementary Figure S3h).

Supplementary Fig. 5A legend: Missing E in E10.5.

This has been amended.

Supplementary Fig. 5D legend: Symbol should be color

This has been amended.

Supplementary Fig. 5E legend: Spell out abbreviation (or collect for all in the beginning/end).

Collected at the end of the figure legend.

Supplementary Fig. 8A: Abbreviations are not in A (move or collect for all in the beginning/end).

Collected at the end of the figure legend.

Supplementary Fig. 8: Length of scales not described in legend or image.

Scale descriptions added.

Discussion, first paragraph: Although mentioned in the results section, I recommend to include a sentence stating that whether these effects would be noticed also when sensory neurons are selectively ablated remains to be seen. This does not weaken or undermine any of the data presented here, but since sensory neurons are obviously an important player in these processes, they should be addressed in this context.

We agree about this interesting nuance and have since added this discussion point during the revision:

“The exact identification of motor-derived signals that influence sympathetic axon guidance is complicated by the fact that this effect could be in part mediated by sensory axons, which also depend on pioneering motor axons to establish their connectivity patterns [16]. Whether sympathetic nervous system defects would manifest when sensory neurons are selectively ablated remains to be seen, however as the viscerosensory fibers are fewer in numbers compared to the motor fibers, we predict this would lead to a milder phenotype.”

The authors examine embryos at E18.5, which is fairly late in development. Was there any phenotype observed, apart from SG/AG morphology? What will the biological consequences be? Are these embryos postnatally viable? Experiments to elucidate the resulting biology of later development are beyond scope and not required for this manuscript, but should be addressed. Could this have implications for NC-derived diseases for example?

We clarify that *Olig2-Cre; DTA* mice do not survive at birth:

“Olig2-Cre; DTA mice are not viable at birth, but survive in the womb, allowing investigation of embryos until birth.”

We have improved the discussion about the later stage phenotype and biomedical implications for dysregulated PNS patterning and we have added a discussion point related to neuroblastoma, as cited in the first reply to the reviewer’s comment.

Methods:

Image analysis of lineage tracing section includes “data not shown”. This should be removed.

It has been removed.

Otherwise, the Methods section is written in a structured and sufficiently detailed manner.

Thank you!

Reviewer #2 (Remarks to the Author):

The manuscript entitled "Motor nerves direct the development of the sympathetic nervous system" by Erickson et al. addresses the interesting and timely topic of how developing axons may direct cell migration and autonomous ganglion assembly, thus coordinating peripheral extension and guidance with the assembly of future synaptic targets by the very same axons. Using genetic tools, the authors generated mouse embryos lacking CNS motoneurons, including preganglionic motoneurons normally innervating the sympathetic chain ganglia (SCGs). They found that ablating motoneuron axons led to defective assembly of SCGs, misplacement of SCG neuron somas and aberrant SCG axon outgrowth, with similar defects in cervical sympathetic ganglia. They found that during normal development, a subset of PHOX2B-expressing Schwann cell precursors migrated along motoneuron axons towards the SCGs, which had previously aggregated in an apparently axon-independent manner from earlier migrating neural crest cells, suggesting that the observed defects were due to aberrant migration of the late SCP-derived SCG progenitor population. The authors further found that, without motoneuron axons, SCG cells aberrantly intermingled with dorsal root ganglion cells (DRGs). Next, the authors used previously published single-cell data to search for potential signaling molecules that could underlie the interactions between SCG progenitors and motoneuron axons, which suggested a potential involvement of the Semaphorin signaling pathway. Lastly, the authors analyzed mouse embryos lacking both Semaphorins 3A and 3F, as well as embryos lacking functional Semaphorin receptors Nrp1 and Nrp2. They found defects in SCG assembly and displacements of SCG neurons in both mutant models and based on these results proposed that Sema3A/3F-Nrp1/Nrp2 signaling underlies the motoneuron axon-dependent migration and aggregation of SCP-derived SCG progenitors into defined SCGs.

General comments:

Overall, I found the study interesting, in principle technically sound, but lacking in substance. The importance of motoneuron axons for the proper developmental assembly of the SCGs becomes very clear.

We thank the reviewer for the very careful analysis and for seeing in-depth. We would like to assure the reviewer that we took all comments very seriously and dedicated all possible time and resources to address them. During the revision, we performed several new experiments and revised substantially the manuscript to clarify the significance of the findings.

We agree with the reviewer that demonstrating the importance of motoneuron axons for SCG assembly is our main and clear-cut finding. To further examine the contribution of nerve-associated SCPs to sympathetic chain, we have now performed lineage tracing experiments with boundary cap tracers and characterized the progeny with single cell transcriptomics. Boundary cap cells exist and maintain their specification only in nerve-associated mode (they reside in nerve exit and entry points of the CNS). We observed that a considerable fraction of their progeny loses boundary cap identity and becomes nerve-associated SCPs. We show that BCC-derived SCPs travel along peripheral nerves toward the sympathetic chain, where they differentiate into sympathetic neurons.

Finally, we analyzed the Sema-related phenotypes more carefully by performing additional whole mount staining, which revealed that the phenotypes observed in Sema mutants recapitulate only in part the defects found in motor nerve-ablated embryos. This suggests that there must be other motor nerve-associated signals that are important for some aspects of sympathetic nervous system development, including neurite outgrowth and navigation.

We rewrote the manuscript in a more systematic way providing explanations or suggestions behind the observed phenomena. We believe that the addition of new key experiments and discussion points has significantly strengthened the study and reinforced our conclusions regarding the previously unappreciated contribution of motor innervation to the assembly of the sympathetic ganglia system.

However, the study comes short of offering a convincing explanation accounting for the observed defects in SCG assembly in the absence of motoneuron axons and vice versa, they do not come up with a good model of what the role of the interactions between motoneuron axons and SCG neurons and axons would be during normal development. The developmental events leading to faulty SCG and cervical ganglion assembly in the absence of motoneuron axons is not at all well documented, beyond a description of some late-migrating SCP-derived SCG progenitors apparently associate with growing preganglionic motoneuron axons. But how does the absence of the latter exactly lead to misinformed SCGs?

We pushed a lot in this direction during the revision and are thankful to the reviewer for pointing out these important issues. We hope that our point-by-point response below and the amount of the new work alleviates most or all of the reviewer's concerns. We managed to establish a convincing model of SCP-supplementation during the later stages of sympathetic ganglia development by adding new lineage tracing experiments (with *Krox20-Cre* and *Prss56-Cre*) and better studying intermediate developmental stages with motor nerve ablation. We show that the shortage of motor nerve-associated progenitors not only leads to the smaller sympathetic chain ganglia, but also reduces both PHOX2B⁺ neural progenitor cells and more drastically satellite glial cell populations (updated figures 4d and 4e). The absence of these cells during the process of ganglia coalescence and growth apparently leads to a fragmented sympathetic chain, ultimately causing hypoplasia of the ganglia.

The other, related defect caused by the absence of motor fibers is the misplacement of autonomic primed SCPs that accumulate along the peripheral sensory nerves in the *Olig2-Cre; DTA* mutant, leading to the formation of ectopic clusters of sympathetic neurons throughout the embryo. One explanation for this phenomenon, is that the misplaced cells would otherwise have made it to the ganglia to fuel its growth during the stages E11.5-E13.5. However, some of these clusters appear in atypical locations quite far from the dorsal aorta, such as the brachial plexus, where they are not observed normally. These observations raised the hypothesis that motor fibers may influence the neurogenic (pro-sympathetic) potential of SCPs, and that the observed phenomenon cannot be explained by abnormal neural crest migration (based on timing of neural crest migration and motor nerve outgrowth, and also because mis-placed sympathoblasts are sensory nerve-associated in specific distant locations, where they get via nerve-assisted transportation).

Likewise, besides a relatively superficial description of SCG defects in the Semaphorin and Neuropilin mutant embryos, the authors do not come up with a good model of what these molecules could actually be doing during the migration of SCGs along preganglionic motoneuron axons. They largely ignore a host of previously published data that could offer a range of alternative explanations for the defects observed in the motoneuron-less embryos, and particularly in the *Sema* and *Nrp* mutants. These include (but are by far not limited to) the long-established roles of somitic Semaphorins in channeling neural crest cell migration, or the previously reported impacts of *Nrp1/2* or motoneuron-removal on sensory axons, which in turn could indirectly affect SCG neurons and axons. Thus, several of the SCG assembly defects observed in these mutants could be entirely unrelated to motoneuron axon-SCG interactions, a possibility that is very briefly mentioned in the discussion, albeit buried within.

We apologize for the shortage of potential explanations in our early submission regarding the phenotypes observed in *Sema* and *Nrp* mutants. We have improved this part of the study by showing phenotypic aspects shared between *Sema*-related mutants and motor nerve-ablation: both models display ectopic nerve-associated, aberrantly projecting clusters of sympathetic neurons in the forelimb, far away from dorsal aorta. We interpret this defect as ectopic neuronal induction after neural crest migration, since the phenotype develops at later stages (after E11.5). We have taken this striking resemblance of phenotypes, along with previous knowledge that motor nerves both produce and respond to *Sema3* signals, as an indication that this pathway may be implicated in the control exerted by motor nerves on sympathetic chain development. It remains possible that the effect observed in *Sema3* mutants is secondary to abnormal navigation or fasciculation of motor axons in these embryos. However, since motor nerves are not disrupted in neural crest-specific deletions of *Nrp1* –and we show in the revised manuscript how sensory fibers are negligible at this stage at the dorsal aorta area– the presence of ectopic clusters of sympathoblasts in this mutant suggests an active signaling role of Semaphorins in either guiding nerve-associated neuroblasts or preventing ectopic induction of neurogenesis.

In all, because the *Sema-Nrp* mutants analyzed in the study do not fully recapitulate all the defects observed in motor nerve-ablated embryos, there must be other motor nerve-dependent mechanisms (but Semaphorin-independent), controlling some aspects of sympathetic ganglia development (particularly in hindlimbs, compare new Figure S7 against new Figures S11-12). We recognize that achieving a complete understanding of these potentially combinatorial signaling processes would require a significant number of knockout transgenic mice, which, regrettably, exceeds the scope of this specific paper. Therefore, despite we further developed this part in the revised manuscript (both Results and Discussion), we decided to present the data as Supplementary Figures (as also suggested by reviewer #3), and we removed any claims of a direct motor-derived molecular mechanism via Semaphorin signaling affecting sympathetic gangliogenesis.

We have extended the Discussion in this direction mentioning the role of Semaphorins in neural crest migration (Goldstein and Kalchauer 1991 Development, PMID: 1769337) and we acknowledge the role of somitic Semaphorins in cell migration and axon guidance. In agreement with the reviewer, we are convinced that the mechanisms related to Semaphorin signaling are complex, as they involve

multiple cellular sources and targets. We are now very careful (and thanks to the referee's advice) not claiming that this pathway fully accounts for the observed phenotype in motor-ablated embryos, but it seems to play a possible role when it comes to increase of late (post neural crest) neurogenic potential in nerve-associated SCPs, as this phenotype is fully consistent with motor nerve ablation. Importantly, we downplayed this part and placed all related results to supplementary information according to the request by Reviewer 3.

This is the part of the main text dedicated to the discussion of these results:

“Unmasking the molecular mechanisms underlying the regulatory action of motor nerves on sympathoblast priming, maturation and outgrowth remains an important biological question for future studies. Notably, the phenotype observed in the absence of motor axons is reminiscent of the defects that arise in the sympathetic chain when Semaphorin-3/Neuropilin signaling is impaired in mutant embryos. Sema3/NRP pathway is required for sympathetic nervous system development [29] and for placing of chromaffin cell precursors in the adrenal medulla following visceral motor nerves [31]. Motor neurons express multiple Semaphorins [52] and use them to regulate guidance receptors in an autocrine fashion [53], and as paracrine signals to control the interactions between developing motor axons and the cells in the innervated tissues, including vascular endothelial cells [27, 28]. Our results support the possibility that Sema3A and Sema3F ligands released by extending preganglionic motor nerves orchestrate the local induction and spatial organization of sympathoblasts, because the clusters of ectopic nerve-associated sympathoblasts observed in knockout Sema3a/3f and Nrp1 embryos strongly resemble those in motor nerve-ablated embryos. Interestingly, other aspects of the motor nerve ablation phenotype in the sympathetic chain were not recapitulated in Sema pathway mutants, suggesting that motor nerves may utilize multiple mechanisms to influence the developing sympathetic system. In addition to this logic, because the activation of Sema3a/f – Nrp1 signaling has been implicated in peripheral nerve targeting to muscles and adrenal primordia [31, 54, 55], a contribution of motor axonal misrouting to SCP disorganization and ectopic mini-ganglia formation cannot be excluded.

The exact identification of motor-derived signals that influence sympathetic axon guidance is complicated by the fact that this effect could be in part mediated by sensory axons, which also depend on pioneering motor axons to establish their connectivity patterns [17]. Whether sympathetic nervous system defects would manifest when sensory neurons are selectively ablated remains to be seen, however as the viscerosensory fibers are fewer in numbers compared to the motor fibers, we predict this would lead to a milder phenotype.

Also, in our experiments we cannot exclude the influence of Semaphorins and other signals derived from local mesodermal populations, including the developing somites [56]. It is possible that a combination of extrinsic cues from different cell sources directs the induction and maturation of sympathetic neurons, as well as controls the navigation of sympathetic nerves. Altogether, these features complicate the unambiguous dissection of the underlying molecular signals, making it a compelling subject for future studies aimed at clarifying how motor fibers influence the surrounding cellular microenvironment.”

Regarding the involvement of sensory axons, we believe that an indirect effect of motor removal through sensory axons is unlikely for three reasons. First, sensory nerves begin to extend after motor axons have intersected the neural crest streams (according to the updated data shown in Figure 1g and Supplementary Figure S1). Second, since the white ramus is vastly dominated by motor fibers, (see updated Figure 1i), the contribution of few viscerosensory fibers projecting to sympathetic ganglia is likely to be minor. Third, after motor ablation, the sensory component of the ventral root is still in place despite minor mis-patterning (updated Figure 4b). Given all this information, it seems that the striking phenotype in sympathetic chain is unlikely to result indirectly via effects on the sensory system. Nevertheless, we highlight the importance of sensory nerves failing to reach the ganglia in the absence of motor axons (e.g., Suppl Fig 3h, i) for the development of ectopic clusters of sympathetic neurons observed in *Olig2-Cre; DTA* embryos. We note, however, that ectopic

neurogenesis also occurs on sensory nerves that appear normal after motor-nerve ablation (e.g., in the brachial plexus of *Olig2Cre; DTA* in Figure 5L and in *Wnt1cre; NRP1^{fl/fl}* embryos in Figure S12d'), not only on defasciculated fibers, supporting the idea that motor axons exert a regulatory/inhibitory effect on sympathetic neurogenesis.

In short, I find the topic of the study interesting and the data presented in principle of good quality, but the findings of the manuscript are too descriptive and preliminary to recommend publication in Nature Communications at this stage.

We understand the referee's point of view, and have undertaken a substantial revision that, we believe, has improved significantly the quality of the study. We performed a number of new experiments, including boundary cap (BCC) and SCP lineage tracing with two new Cre lines and single cell transcriptomics to examine the progression of nerve-associated cells (updated Figure 3d-g). We provide a detailed description of the natural sequence of events leading to sympathetic ganglion formation (updated Figure 1, S1, 2) and improved our interpretation of the phenotypes of motor-ablated embryos (updated Figure 4, 5, 6). We sought to determine the mechanism of action by which motor neurons affect SCG axons by increasing the amount of data related to Semaphorin signaling (whole mount of *Sema3A/3F* mutant in updated Figure S11, whole mount of neural-crest specific conditional knockout of *Nrp1* in Figure S12) and generating a more informative receptor-ligand interaction prediction using more complete and time-matched single cell datasets (updated Figure S9). Following the reviewer's advice, we improved the Discussion, paying special attention to the potential contributions of Sema-related molecules and different nerve types. We hope the reviewer finds that our manuscript has matured positively, given all the new additional data and better-informed interpretations. A detailed description of the changes is given in response to specific comments in the next section.

Specific comments:

1. Page 6, paragraph "Ventrally migrating neural crest cells are drawn to motor axons exiting the spinal cord". The paragraph and the Supplementary Figures (5A-E) mentioned actually do not contain any evidence supporting the statement in the paragraph heading of ventral neural crest cells being "drawn" to motoneuron axons, which is quite confusing.

Yes, we agree with reviewer, and we apologize for the wrong language. We did not mean a specific chemoattraction here, and we removed such wording in the revised text. Also, during the revision, we performed more analysis about the early interactions between neural crest and the outgrowing motoneuron axons, including the onset of *ITGA4* expression - a marker of emerging hub/SCP state (Kastriti et al., EMBO J 2022 PMID: 35815410). The wording "sponge" and "drawn" is indeed more evocative than necessary, and we have changed the text to more appropriately describe the observed sequence of events:

"To track the dynamics of motor axons during neural crest migration and subsequent steps of sympathetic ganglia development, we utilized Hb9-GFP transgenic mice in which motor neurons are labeled with green fluorescent protein (GFP) (Fig 1b). We observed that outgrowing motor axons partly interrupted the neural crest wave, and freely migrating

NCCs became associated with the nerve as SCPs. This transition is evidenced by a gradual shift in the migratory stream angle (Fig 1c), widening gap between SCPs and free migrating NCCs (Fig 1d), and consolidation of the SOX10⁺ stream as the nerve grows towards the sympathetic anlagen (Fig 1e)."

In this setting, we do not imply that the migratory neural crest is attracted to motor nerves via a morphogen gradient or specific signal, and we rather assume based on our latest data in this revision that the stream of the neural crest intersects with outgrowing motor axons, causing the nerve-association of the neural crest and their transformation into SCPs. We thank the reviewer for pointing this out.

3. Motoaxons have long been thought to form a lattice for the extension of later-developing sensory axons (e.g. Honig et al. Dev. Biol. 118, 532). While the authors acknowledge the latter (page 10, last sentence), they generally fail to consider the extent to which these defects depend on direct interactions of SCGs with motoaxons (or the lack thereof) or the indirect motor axon-dependent misrouting of sensory axons on which in turn SCGs and axons in turn might depend (the same goes for: Page 5, paragraph heading: "Motor ablation causes misrouting of sensory nerves away from sympathetic ganglia").

We indeed do not have the direct data that would show how sympathetic neurons interact with the sensory fibers, and how this interaction might cause effects on motor nerve. However, the development of sympathetic nerves in the trunk is relatively late compared to motor and sensory nerves, which are very much extended and navigated to all sites when outgrowth from sympathetic neuroblasts at the dorsal aorta begins (after E11.5-E12). The late timing of sympathetic outgrowth suggests that SCGs are more likely to be influenced by, rather than to affect, motor-sensory neuron projections. Given the known hierarchical interactions among nerve fiber types (Wang et al. PMID: 24700820), we cannot exclude that misrouting of sensory nerves after motor neuron ablation alters the patterning of sympathetic nerves. A significant contribution of our work is to report the precocious and excessive growth of SCG axons, from both ectopic clusters and ganglia-located neurons, which we interpret as the result of lack of motor-derived inhibitory signals that would normally control the timing and extent of SCG outgrowth. A corollary of this observation is that sensory fibers represent a highly permissive substrate for SCG projection that necessitates modulation by motor signals.

The previous version of the manuscript included claims about sensory nerve misrouting, when the comparisons were actually the peripheral nerve with and without motor components (i.e., motor + sensory versus sensory-only). The reviewer was correct to ask clarification related to the precise involvement of sensory nerves. During the revision, we performed additional experiments to refine our model of the developing neuroanatomical pathways and confirmed that the viscerosensory component of the white ramus is minimal and develops after motor neurons start to project toward the dorsal aorta. For establishing this, we relied on TrkA staining of sensory fibers (which became clearly visible by E12.5, Figure 1i) as well as a double staining at earlier stages (E10.5 and E11.5) using *HB9-GFP* for motor and pan-neuronal TUJ1, in which TUJ1+/GFP- were identified as sensory axons (Figure 1g-h). Using *Hb9-GFP*-labeled sections, we show that the initial nerve-association of neural crest is mainly with motor axons, prior to the outgrowth of the bulk of sensory axons. Given their

minimal presence and delayed development, it is highly unlikely that the sensory fibers will play a significant role here (though not impossible).

The revised manuscript contains a better description of the sequence of events in this part:

“During the early recruitment of SCPs, between E10.5 and E11.5, the ventral root is dominated by motor axons, as most TUJ1⁺ nerve fibers were also Hb9-GFP⁺ (Fig 1g and 1h). By E12.5, the outgrowing spinal nerves carried both motor (efferent) and sensory (afferent) fibers that could be distinguished by Hb9-GFP and TRKA (NTRK1) labeling, respectively (Fig 1i). Notably, the majority of the white ramus communicans, connecting to sympathetic ganglia, was composed of Hb9-GFP⁺ preganglionic visceral motor axons branching off the common nerve bundle, while TRKA⁺ viscerosensory axons extending from the dorsal root ganglia (DRGs) were present in small numbers (Fig 1i). Therefore, SCPs predominantly migrate along developing motor axons to reach the sympathetic anlagen (Fig 1j). “

and the new Discussion mentions the potential role of sensory fibers:

“The exact identification of motor-derived signals that influence sympathetic axon guidance is complicated by the fact that this effect could be in part mediated by sensory axons, which also depend on pioneering motor axons to establish their connectivity patterns [17]. Whether sympathetic nervous system defects would manifest when sensory neurons are selectively ablated remains to be seen, however as the viscerosensory fibers are fewer in numbers compared to the motor fibers, we predict this would lead to a milder phenotype. “

4. Throughout the manuscript claims are made about “repulsive navigational cues” or cells being “drawn to” motor neuron axons or the axons “sponge” neural crest cells (Page 6, paragraph 2, last sentence), without any direct evidence for the underlying cellular interactions.

As we mentioned above, we agree that that imprecise and inappropriate terminology needed to be removed, according to the reviewer’s advice. We implement this throughout the manuscript.

For instance, would neural crest cells indeed be attracted by motoaxons and are the latter indeed repulsive for SCGs?

In the past, we performed co-culture assays, and we learned that the nerve-association of the neural crest is largely driven by nerve-anchored NRG1, in agreement with previous classic studies (Semin Cell Dev Biol. 2010 Dec; 21(9): 922–928, PMID: 20832498. and Cells. 2022 Dec; 11(23): 3753, PMID: 36497014). This does not imply that there is an attractant signal from the nerve to the migrating neural crest or vice versa. We revised manuscript for clarity and consistency in accordance with this comment. Furthermore, to address this comment, during the revision we further characterized the behavior of neural crest nerve-association based on quantification of the temporal sequence of events (updated Figure 1b-e, updated Figure S1). We show that the stream of the trunk neural crest migrating ventrally is broad, and upon intersecting the outgrowing motor nerve it engages into massive nerve-association, while an earlier wave reaches the dorsal aorta via free migration prior to motor nerve outgrowth.

A possibility to more directly dissect such interactions could be through relatively straightforward in vitro co-culture assays.

During revision, we attempted to perform the suggested co-culture assays modifying the available protocols (Wang et al. PMID: 22281870) but did not obtain interpretable results due to a survival problem with cultured sympathoblasts in motor neuron media. We performed MN-DRG explant co-culture using DRGs as source of migrating SCPs, and tried to induce glial cells to become sympathetic neurons in culture. However, we only obtained few sparse TH+ neurons that were insufficient to test axon-axon interactions. Thus, although we put considerable effort into these experiments over the past several months, regrettably, we did not reach conclusive results that could be included in the manuscript (See panels below). However, we are eager to optimize these assays in the future to dissect the combinatorial signals mediating motor-sympathetic-sensory interactions, including and beyond Semaphorins.

This data above is a summary of our attempt to use co-cultures to test for direct interactions between motor axons and sympathetic axons. However, our transgenic mouse lines facilitating efficient

dissection of motor explants (isolated by signal from Hb9-cre) and sympathetic explants (targeted for dissection using Wnt1-cre) were in different countries.

We reasoned that a model of motor and sensory co-cultures would include many multipotent SCPs, which in principle should allow us to test whether motor versus sensory niche could influence: 1) SCP migration 2) SCP neurogenic differentiation, and 3) sympathetic axon growth. We tried to induce sympathetic neurogenesis from SCPs by adding a cocktail of growth factors (BMP4, SDF1, NGF, and GDNF). While we managed to get a few differentiated TH⁺ neurons within DRG explants in culture with MN explants using these growth factor treatments, it was not enough to test for a significant interaction.

5. Figure 7: The authors provide only a brief documentation of trunk SCG developmental defects in Sema and Nrp mutants (Figure 7), without providing information of other aspects disrupted by motoneuron axon-ablation, such as the intermingling of DGRs and SCGs, the SCG axon defects or the defects in the cervical sympathetic ganglia.

Yes, we understand why reviewer thinks this would be interesting to explore. Importantly, during this revision we implemented the analysis of Sema mutant embryos by performing wholemount immunostaining which allowed us a better comparison between nerve ablated and Sema mutant embryos. We found that some of the phenotypes observed after motor nerve ablation were not recapitulated in Sema mutants, and vice versa. This means that there are other mechanisms involved in mediating motor nerve/sympathetic chain interactions. However, we observed that the formation of ectopic sympathoblasts along nerves in the brachial plexus appeared conserved between Sema-related mutants and motor nerve ablation embryos, suggesting that Semaphorins might influence the neurogenic potential of nerve associated SCPs.

We did not observe a large accumulation of thoracic nerve-associated sympathoblasts near DRGs in Sema mutants at E13.5 versus E11.5 – rather the misplaced sympathetic cells appeared to decrease in this region over time (compared to an obvious accumulation in the *Olig2-Cre; DTA*). Therefore, we doubt that we would see commonalities between Sema and motor ablation (beyond ectopic sympathetic clusters) at later stages when it comes to sympathetic nerve navigation to thoracic DRGs and sensory-sympathetic ganglia fusion. As the sympathetic nerve navigation defects are apparently not the same in motor ablation and Sema-related KO models, these aspects of *Olig2-Cre; DTA* phenotype are most likely controlled by other mechanisms. Please see updated Figure S9-S12 and here is the discussion related to this point in the main text:

“Fragmentation of the sympathetic chain and ectopic clusters were also visible in $Nrp2^{-/-}$; $Nrp1^{Sema/Sema}$ mutants in which all class-3 Semaphorin signaling is abolished [34] (Suppl Fig 11g). However, the abnormally robust outgrowth of sympathetic fibers in the hindlimb and thoracic regions observed in motor nerve-ablated embryos was not recapitulated in $Nrp1$ and $Sema3a/3f$ signaling mutants. These results suggest that while motor nerve-derived SEMA3A/3F may be partly responsible for the placement of SCPs and structural integrity of sympathetic ganglia, there are other means by which motor nerve regulates sympathoblast maturation and axonal outgrowth. These aspects of the motor nerve-driven phenotype, independent of Semaphorin signaling, warrant future investigation.”

Despite this logic, we were still motivated to carry out the suggested experiment, but we could not revive that part of a mouse colony due to the time required for strain rederivation and expansion at

our institutions, and could only obtain a few litters to answer the most urgent comments. The rest of the work would require time which is beyond this year-long manuscript revision. We hope that the study of *Sema*-mutants at later stages is less essential to our story, especially given that we moved all *Sema*-related data to Supplementary as requested by Reviewer #3 and overall downplayed this part of the study.

Moreover, the different markers and the assigning of different colors to markers, as well as the lack of comparable wholemount images for the *Sema*/*Nrp* mutants makes it altogether difficult for the reader to compare these defects with those obtained by ablating motoneurons (Figures 1-4) and to appreciate the extent to which they are indeed similar as claimed (Page 9, first paragraph). One has the impression that both data sets (obtained by motoneuron ablation, *Sema*/*Nrp1* knockouts, respectively) were obtained by two different research groups with little prior coordination. All in all, these data are too preliminary for the authors to convincingly make their case that the *Sema*/*Nrp* signaling between preganglionic motoneuron axons fully account for the SCG assembly and axon defects observed upon motoneuron ablation.

We thank the referee for this very important comment, and we largely agree. We did our best to improve the results in this direction, and for this, we managed to generate new whole mount data as requested with comparable staining and quality (updated Supplementary Figure S11-12). We also understand that the description of the role of the *SEMA3* pathway is complex when it comes to this topic because motor neurons are capable of secreting as well as responding to *SEMA3* signaling, and *SEMA3* signals are secreted from somites as well as MNs, meaning that phenotypes of full knockouts for Semaphorin ligands cannot be interpreted at face value as fully phenocopying the MN ablation (and they do not phenocopy fully, which we confirmed in this revision). To respond to this comment, we have provided a greater detail to the characterization of these mutant embryos in the revised manuscript.

Given this new information, and also as requested by Reviewer 3, we brought all *Sema3* data into Supplementary information to downplay this part, but we wanted to include it since we still believe these data might be useful to the next generations of researchers addressing the same problem. We carefully rephrased our discussion about *Sema*-related observations:

“Unmasking the molecular mechanisms underlying the regulatory action of motor nerves on sympathoblast priming, maturation and outgrowth remains an important biological question for future studies. Notably, the phenotype observed in the absence of motor axons is reminiscent of the defects that arise in the sympathetic chain when Semaphorin-3/Neuropilin signaling is impaired in mutant embryos. Sema3/NRP pathway is required for sympathetic nervous system development [29] and for placing of chromaffin cell precursors in the adrenal medulla following visceral motor nerves [31]. Motor neurons express multiple Semaphorins [52] and use them to regulate guidance receptors in an autocrine fashion [53], and as paracrine signals to control the interactions between developing motor axons and the cells in the innervated tissues, including vascular endothelial cells [27, 28]. Our results support the possibility that Sema3A and Sema3F ligands released by extending preganglionic motor nerves orchestrate the local induction and spatial organization of sympathoblasts, because the clusters of ectopic nerve-associated sympathoblasts observed in knockout Sema3a/3f and Nrp1 embryos strongly resemble those in motor nerve-ablated embryos. Interestingly, other aspects of the motor nerve ablation phenotype in the sympathetic chain were not recapitulated in Sema pathway mutants, suggesting that motor nerves may utilize multiple mechanisms to influence the developing sympathetic system. In addition to this logic, because the activation of Sema3a/f – Nrp1 signaling has been implicated in peripheral nerve

targeting to muscles and adrenal primordia [31, 54, 55], a contribution of motor axonal misrouting to SCP disorganization and ectopic mini-ganglia formation cannot be excluded.”

...

“Also, in our experiments we cannot exclude the influence of Semaphorins and other signals derived from local mesodermal populations, including the developing somites [56]. It is possible that a combination of extrinsic cues from different cell sources directs the induction and maturation of sympathetic neurons, as well as controls the navigation of sympathetic nerves. Altogether, these features complicate the unambiguous dissection of the underlying molecular signals, making it a compelling subject for future studies aimed at clarifying how motor fibers influence the surrounding cellular microenvironment.”

Overall, we hope that the reviewer will appreciate the significant amount of new experiments and our fair attempts to address all the comments. By providing a more detailed description of how SCG development depends on motor innervation, we hope that we have delivered a more comprehensive and convincing story.

Reviewer #3 (Remarks to the Author):

This manuscript describes an involvement of motor neurons in shaping the developing sympathetic ganglia via Schwann cell precursor (SCP) contribution after the sympathetic ganglia have initially been formed by the migrating neural crest cells. The finding is relevant as sympathetic ganglia are thought to arise solely from the migrating, primary neural crest cells with no input from SCPs. The manuscript is based on high resolution confocal images from several mouse lines.

However, the manuscript is not well written and it is very hard to follow; findings throughout the text and figure legends are poorly explained to the point that it's difficult to judge the data. To make better sense of the results and the phenotype caused by motor-nerve depletion, the novel finding of SCP contribution to the sympathetic ganglia in normal development should be described in a much more detailed manner.

We thank the reviewer for recognizing the main finding of our study. We did our best to improve all parts of the manuscript in accordance with reviewer's advice during these almost 1.5 years of revision. The manuscript has been essentially rewritten and the structure of the figures has changed substantially. We have added schematics throughout the manuscript, which should facilitate understanding the main message of each figure. Importantly, we performed a number of additional experiments to improve the lineage tracing of SCPs and nerve-associated boundary cap (BCC) derivatives by adding the analysis of *Krox20-Cre* and *Prrss56-Cre* BCC-specific transgenic embryos. We characterized nerve-associated lineage-traced cells on sections and also using single cell transcriptomics approach. These new data provided a strong support for that the white ramus nerve serves as a transportation route for progenitors complementing sympathetic ganglion development. Please find new data in updated Figure 3. We also improved the quality and number of staining showing the progression of normal development – both neural crest migration to the sympathetic chain and SCP formation (new Figure 1, 2 and S1). We hope the reviewer will find these changes and additions satisfactory.

None of the experiments describe the sequence of WT events in a satisfactory manner, which leaves open questions: at what stage do the SCPs start moving towards the sympathetic ganglia, what markers do they express that differentiate them from glial cells, do all the cells migrate along the sensory nerves that are guided by the presence of motor nerves, or do some also migrate along the motor nerves (the results seem contradictory on this topic in the text)?

We apologize for the poor organization of the previous version of the manuscript and some important details being omitted. In the revised study, we focused a lot on describing the series of events underlying the transition from free neural crest migration to nerve-associated SCP wave, and we hope that the reviewer will appreciate our efforts (updated Figure 1b-e and S1). We additionally provided the comprehensive picture of motor and sensory fibers within the vicinity of the

sympathetic chain (updated Figure 1g-i), and clearly identified nerve-associated SCPs with the use of a prospective SCP marker ITGA4 (updated Figure 1f). We find that ITGA4 is differentially expressed between the nerve-associated SCPs and the freely migratory neural crest as predicted from our previous transcriptomic analysis (Kastriti et al 2022, PMID: 35815410). Therefore, ITGA4 seems to be a reasonably good indicator that the cells newly associated to the outgrowing motoneuron axon express markers of the “SCP hub-state” (Figure 1f). 4) Of note, we find a reduction in the number of satellite glia in putative sympathetic ganglia in the absence of motor nerves, indicating that SCPs contribute to the satellite glia population, not only to sympathetic neuroblasts (Figure 4c-d).

Based on our immunohistochemical analysis, SCPs travel along both motor and sensory fibers in mixed nerve bundles, as well as in pure sensory/motor nerves. Thus, the contributions of various nerve types to the migration process depend on their relative amounts. We now tracked the relative contribution of motor and sensory axons to the migration of the SCPs using TrkA (NTRK1) for sensory axon labeling and GFP reporters for motor axons (Figure 1g-l and Figure 4a-b). Though the ventral root is a mixed sensory-motor nerve, the “white ramus” that branches from the common nerve tract to project toward the sympathetic ganglia is mostly composed of motor fibers, with only minor contribution of sensory axons (updated Figures 1i and 4b). Consequently, the bulk SCP migration towards the sympathetic chain occurs along motor axons. In addition, SCPs associate robustly to newly extending motor axon prior to sensory nerve outgrowth (E10.5 images in Figure 1g). Hence, the phenotypes observed in the sympathetic chain of motor-ablated *Olig2-Cre; DTA* embryos come from a failure of SCPs to be appropriately transported to the developing sympathetic ganglia via the *white ramus communicans*. While some re-routing of sensory axons occurs in motor ablation models, the bulk of the phenotype appears independent from that re-routing effect.

It also remains unclear if all the migrating primary neural crest cells that populate the sympathetic ganglia migrate around the sensory ganglia or do some go through them, and at what stage do they become SCPs and stop being migrating crest (as the manuscript does occasionally refer to the cells as migrating crest) and when can they be called sensory glial cells (the lab has just published on this topic so marking these different SOX10+ populations is a realistic request).

We have clarified these points adding new panels (Figure 1b-f) and have inserted a descriptive schematic illustrating the events of NCC migration and sympathetic chain formation according to previous studies (Figure 1a). Our results agree with previous studies describing the process of the neural crest migration towards dorsal aorta in wild type embryos (Saito et al 2012 PMID: 22723422; Newbern et al 2015 PMID: 25662262, Serbedzija et al 1989 PMID: 2562671, Kasemeier-Kulesa et al 2005 PMID:15590743, Graham, 2003 PMID: 12747846). In brief, the earliest neural crest migratory streams move ventrally through the site where sensory ganglia will develop at a later time and proceed towards the dorsal aorta, where they coalesce into primordial sympathetic ganglia aggregations. Hence, early-migrating neural crest cells that contribute to the bulk of the sympathetic ganglia make their journey prior to the formation of the sensory ganglia (as evidenced by the near-absence of thoracic Hb9-negative TUJ1-positive sensory fibers at E10.5, see updated Figure 1g).

Later-migrating neural crest cells stop on their way and start building the sensory ganglia. The large portion of the neural crest, which migrated just below sensory ganglia, becomes associated with the outgrowing motor nerve, thus turning into nerve-associated SCPs. The earliest nerve-associated cells contact the ventral root while the dorsal root ganglia are still in the process of coalescing (Figure 1b, showing the rapid nerve-association at 10.5 immediately after motor nerve outgrowth; Figure S1c, comparing 2H3 signal from brachial versus caudal indicating sensory neurons remain immature posteriorly at E10.5). In contrast, later nerve-associated cells, migrate through the forming DRGs to reach the outgrowing motor nerve and eventually the sympathetic ganglia chain. The soon-developing sensory neurons extend axons that follow and fasciculate with pre-extending motor fibers, and they also become associated with SCPs that divide robustly covering the entire surface of sensorimotor nerves. Both motor and sensory axons, albeit in much smaller amounts, reach the primordial sympathetic ganglia carrying SCPs (updated Figure 1g-i), which become recruited as the second complementary cell source to give rise to additional sympathetic neuroblasts and satellite glial cells according to our lineage tracing experiments (updated Figure 3) and revised motor-nerve ablation data (updated Figure 4c-e). This choreographed sequence is now clarified in the manuscript and supported by new data and references to previous work from our team and others [Adameyko et al 2009 PMID:19837037, Adameyko et al 2012 PMID: 22186729, Dyachuk et al 2014 PMID:24925909, Espinosa Medina et al 2014 PMID:24925912].

The data itself also raises several concerns, some of which may be due to misunderstanding from my part caused by the poor communication of the results. Overall, the work in its current form leaves a lot of open questions and seems premature.

We hope that the substantially revised manuscript will be easier to follow and will address the reviewer's concerns. We agree that the organization of the previous version was suboptimal and prone to raising questions about the interpretation of the findings. During the thorough revision, which took us more than one year to complete, we have re-written the text presenting the results and interpretations in a clearer, more logical manner, reorganized the figures, added schematics that illustrate the findings and our new model of sympathetic ganglia formation. We have performed new experiments, and reanalyzed previous data, to define in greater detail the transitions and dynamics of SCPs. We have conducted lineage tracing and single-cell sequencing to determine the contribution of nerve-associated SCPs to the sympathetic chain. Additional improvements and clarifications are outlined below.

Specific concerns:

Introduction: Paragraphs 2 and 3 are redundant.

We agree with the reviewer. We merged those and eliminated redundancies.

Images and explanations

The manuscript is written to a very limited audience that is assumed to have full knowledge on the literature and the detailed stage by stage peripheral nervous system development and anatomy in mouse embryos, which does not serve the readership of any journal. Importantly, since this work is solely based on imaging data, every microscope images need an explanatory cartoon with an orientation of where the image was taken from in the embryo and what are the surrounding structures are to guide the reader and to show how the WT embryo forms as compared to the mutants. The labels need to be significantly improved, one can not say 1) "SCPs found at close vicinity of the dorsal aorta" without marking the DA in the images,

In the revised version we use schematics or overviews to orient the reader for all main messages. To address the specific examples:

"Sympathetic chain development begins with ventrally migrating waves of SOX10⁺ NCCs that coalesce in the vicinity of the dorsal aorta from where they receive inductive signals, such as Bone Morphogenetic Protein (BMP)" corresponds to Figure 1a, a schematic in which the dorsal aorta is labeled, and,

"Together, these data reveal that motor nerves recruit nearby freely migrating NCCs into SCPs, and that these motor nerve-associated SCPs become primed to an autonomic fate en route as they approach the sympathetic ganglia" Corresponds to Fig 2e, a schematic in which the sympathetic ganglia is labeled.

or 2) talk about TH⁺ and 2H3⁺ cells without first explaining why the markers were used and thus what those cells are

The revised Results section and updated Figure 1a describe the utility of TH, PHOX2B, and SOX10 as cell markers:

"Sympathetic chain development begins with ventrally migrating waves of SOX10⁺ NCCs that coalesce in the vicinity of the dorsal aorta from where they receive inductive signals, such as Bone Morphogenetic Protein (BMP) [7]. These precursor cells, known as "sympathoblasts", express early autonomic markers (Phox2b) and later differentiate into bona fide sympathetic neurons (labeled by Tyrosine Hydroxylase, Th) (Fig 1a)"

or 3) talk about Sox10⁺ -positive cells without mentioning what they mark in the specific context (the Neural crest, SCPs or glial cells)

In the revised manuscript we clearly state which type of SOX10⁺ cell we refer to (whether it be NCC, SCP, or satellite glia), only using the identifier "SOX10" when we refer to multiple of these cell groups simultaneously. In some images showing the transition between NCC and SCP states, SOX10 staining identifies both free-migrating neural crest and nerve-associated SCPs (See Figure 1b, f and Figure S1a, c).

or 4) talk about nerve fibers without mentioning which nerves are the authors referring to (visceral motor, sensory or sympathetic?).

In the revised manuscript, we systematically specify the nerve type being examined. There are however cases in which the staining does not distinguish between various subtypes (as we point out above). We explain the use of the Hb9-GFP transgenic line for specific motor axon visualization:

“To track the dynamics of motor axons during neural crest migration and subsequent steps of sympathetic ganglia development, we utilized Hb9-GFP transgenic mice in which motor neurons are labeled with green fluorescent protein (GFP) (Fig 1b).”

The use of a neurofilament (2H3) staining is mentioned within the same paragraph. Note that 2H3 alone does not differentiate between motor and sensory nerves, so we are being as specific as possible in the following passage:

“Indeed, at E10.5 ITGA4 levels were greater in SOX10⁺ cells that were associated to neurofilament-positive (2H3⁺) peripheral nerves of the ventral root, compared to NCCs that were still freely migrating in the caudal region (Fig 1f), suggesting that nerve association coincided with the adoption of the SCP hub state.”

We outline our rationale for a dual antibody stain to distinguish motor from sensory nerves: at early stages, when TrkA expression is low, we used a combination of tubulin-β3 (TUJ1) and GFP labeling in Hb9-GFP mice, while after E12.5 we used TRKA staining to specifically mark sensory nerves. The following passage from the text explains:

“During the early recruitment of SCPs, between E10.5 and E11.5, the ventral root is dominated by motor axons, as most TUJ1⁺ nerve fibers were also Hb9-GFP⁺ (Fig 1g and 1h). By E12.5, the outgrowing spinal nerves carried both motor (efferent) and sensory (afferent) fibers that could be distinguished by Hb9-GFP and TRKA (NTRK1) labeling, respectively (Fig 1i).”

We also made a special effort to introduce new data that estimates the relative contribution of motor versus sensory fibers to mixed nerves in the ventral root, and show that the white ramus is near-purely visceromotor in its composition.

All these small deficiencies and unmarked structures in the figures make the manuscript exhausting to follow. In sum, the text including figures and figure legends need to be re-written with a systematically more explanatory style.

Following the reviewer’s request, we have systematically reorganized the figures, added schematics and rewritten the legends (and Results) to ensure the manuscript is clear, coherent and accessible to a broader audience.

Figure 1

- Fig 1A,B: The results show the main phenotype, constellation of prominent ectopic TH positive cells along the entire trunk. The WT embryo has no TH positive cells projecting from the sympathetic chain, which is inconsistent with previous findings that do show these branches in WT (PMID:

21325504), which raises a concern. Please address the difference between the published work and your data.

The reviewer refers to lack of TH+ nerve outgrowth in WT embryos at E12.5. This is indeed the case in embryos on C57/Bl6 strain (used throughout the study), which often show slower development compared to other strains. TH+ nerve fibers are detected from E13.5. The paper indicated by the reviewer uses colorimetric immunohistochemistry, which has higher sensitivity compared to immunofluorescence staining we used, but also shows higher background. In the cited paper, the youngest axons seem to be very low in TH levels. However, we systematically searched the literature and found many publications that are in agreement with the timeline of sympathetic ganglia development and TH+ neurite outgrowth described in our study (e.g., Maden et al 2012, PMID: 22790009, Figure 2B; Battiste et al 2007, PMID:17166924, Figure 2D).

In conclusion, sympathetic nerve length changes significantly between E12.5 and E13.5. This fast developmental pace, together with differences in specific mouse genetic backgrounds, seems to play a major factor in the variability pointed out by the reviewer. It is important to note that we always compare phenotypes between littermate embryos (e.g., Cre-negative versus Cre-positive littermates on a DTA background), so our experiments are appropriately controlled, and the conclusions are not affected by possible differences in the pace of development. This is further reinforced by our extensive analysis of time series throughout the revised paper, which confirms that the problems in sympathetic axon growth caused by motor ablation cannot be accounted for by differences in developmental timing (Figure 4f, Figure 5, Supplementary Fig S7).

Are the WT and DTA embryos images from the same focal level and do both images contain the same amount of stacks in the maximum projection? The WT embryo images have less 2H3 positive nerve fibers than the DTA counterpart (indicating the DTA image consists of either more z-stacks or same the amount but captured from a different focal plane) and seem to have been imaged from the focal plane of the sensory ganglia, whereas the ganglia are out of focus in the DTA – are these differences shown in the images real or caused by technical imaging discrepancy? Similarly, the fluorescence intensity of the sympathetic chain in fig 1B is much lower in the WT images as compared to DTA, why?

The images contain all stacks from the midline to the end of the sample (tips of the limbs) and they were all imaged in the same way, although some variation across samples in the staining intensities is possible. The same focal planes are shown along the z-stack in the DTA and WT images.

We think that TH fluorescence signal in WT appears slightly lower because these embryos might have lower TH amounts compared to DTA conditions where we observe faster differentiation of some ectopic sympathetic neurons and earlier neurite outgrowth. We decided not to emphasize this observation in the manuscript because we have not measured TH levels by ELISA or other more quantitative assays and it was not essential to our story.

We added different examples of whole mount samples at E12.5 in Supplementary Figure S6 (related to revised Figure 5), to show the variability in staining intensity among different embryos and illustrate how whole mount images of *Olig2-Cre; DTA* and controls were processed for comparison.

Finally, I was not sure how the quantifications were done to get a result of the general TH volume to be lower in DTA, that does not match with what the images show. Additionally, as the sympathetic chain ganglia get more fragmented over developmental time, the range of developmental stages within a timepoint, (even amongst littermates) can be broad. The number of embryos is only 3 or 4, respectively – were the developmental stages aligned by additional measures like somite counting? The imaging discrepancies need to be addressed before conclusions can be made in a convincing manner.

The developmental stages of the compared littermate embryos, when it comes to crown-rump length (Supp Fig S4b) and volume of the sympathetic ganglia at stages E10.5 and E11.5 (Fig S4c-d) is consistent, but we observed some staining variability across embryos, as shown in Fig S6. The panel shown in the previous submission was from a sample with weaker TH staining, which has now been replaced with a more representative image (updated Figure 5d-i).

We added panels in supplementary Figure S6 (right column) showing how the volumetric segmentation was performed. Importantly, we considered the TH⁺ volume from the sympathetic ganglia chain in its standard position, while excluding the pelvic or adrenal regions, as well as all misplaced TH⁺ cells. This is clarified in methods:

“Volumetric quantification of sympathetic chain volumes was performed using Bitplane Imaris 9.5 using the surface generating tool. 5 to 7.5 micrometer smoothing was used for all light sheet images, 1.25 micrometer smoothing was used for confocal images. Manual segmentation was used to separate cervical ganglia, sympathetic chain, adrenals, and misplaced sympathoblasts before surface generation. Signal intensity thresholds, which were automatically recommended by the Imaris software for each whole unsegmented image, were used for automatic surface generation.”

- Figure legend for 1f is missing

This has been corrected.

Figure 2

- As mentioned above, the authors should mark the structures that are discussed and be specific. For example: “Sympathetic somas” on “which” nerve?

During revision we realized that some of the panels were redundant, because they were showing many examples of the same phenotype in principle, when one would suffice. This panel was removed for that reason. We have tried to be more specific in the revised version. For instance, we make distinctions between the axial, intercostal, and white ramus nerve branches of the ventral root. During

early stages in which these divisions have not been set up, we make sure to mention we are referring to the ventral root and the axial level, when applicable.

- Why are the sg not TubIII-positive? Please show the channels separately in addition to the merged figures.

At E11.5, the sympathetic ganglia in the caudal region still contain immature sympathoblasts that have not accumulated high levels of TubIII, which marks mature neurons. At later stages (E12.5-E13.5, and in thoracic E11.5), TubIII is expressed throughout the sympathetic chain (see Figure S10b-c, Figure S11e, Figure 5a-b). Misplaced sympathetic neurons are expected to exhibit lower levels of TubIII if they differentiated more recently. Moreover, they are invariably attached to TubIII+ sensory nerves too, making it difficult to resolve their labeling. Nevertheless, the sequential expression of PHOX2B/TH and extending axonal projections strongly suggests these ectopic cells are sympathetic neuroblasts. In addition, loss of motor nerves in *Olig2-Cre; DTA* embryos would be another reason for a reduction in the TubIII signal around sympathetic chain area. If the reviewer feels it is important, we can bring back these panels containing many different examples of the DTA phenotype and present them with channels separated, but we hope this issue is already clear in the revised and restructured manuscript.

- Why don't the authors show WT images of the normal SCP contribution into the sg via the motoneurons/motoneuron guided-sensory nerves? If TH is not expressed in the motor nerve associated SCPs and only comes on when they reach the condensed sg, please use another, SCP-specific marker to show the cells and explain this clearly in the text. Or is Sox10 marking also glial cells here, the use of Sox10 for three different purposes is very confusing and requires an additional marker to separate between the stages (NC, SCP, Glia)?

We reorganized the manuscript to include a section focused on describing the normal SCP contribution to sympathetic ganglia via the *white ramus communicans* of the ventral root (updated Figure 1g-i, updated Figure 2, and lineage tracing experiments in updated Figure 3). We previously used SOX10 expression to mark NC/SCP/Glia and used their locations to distinguish them (NCC are freely migrating, SCP are nerve-associated, satellite glia are ganglia associated), as these cell populations are very difficult to distinguish normally and they are all commonly identified with the SOX10 marker. Using SOX10 in combination with cell position (free migrating, nerve-associating, ganglia-residing) is often sufficient for distinguishing all 3 populations. However, in the revised manuscript, we show that SCPs, after nerve-association, activate the expression of ITGA4, allowing to distinguish them from free-migrating NCC (updated Figure 1f). For other prospective markers, such as Sox8 and Serpine2 (Kastriti et al 2022, PMID:35815410), we could not find suitable antibodies that clearly delineate these cell populations.

Moreover, our new finding that SOX10+ satellite glia residing in the ganglia are reduced in motor-ablated embryos (updated Figure 4c-e) suggests that both SCP and satellite glial, independently of their subtle transcriptional differences, are supplied by visceral motor nerves of the white ramus and contribute to sympathetic ganglia growth. In addition, we used the PHOX2B marker to identify autonomic priming in SCPs migrating along motor nerves (SOX10+/PHOX2B+ cells).

Overall, we hope the reviewer agrees that 1) in the revised manuscript we more clearly define SCPs, NCCs, and satellite glia, 2) we now use ITGA4 and PHOX2B (in addition to nerve-association) to molecularly define SOX10+ cell populations as SCPs, and that 3) fine distinctions of the glial populations in the vicinity of the SG are overshadowed by the fact that both populations rely on motor nerves for their appropriate placement in the body.

- Also please verify that the difference between the condensation of the sympathetic ganglia between WT and DTA is not due to analysis at a later developmental stage.

All comparisons are made on littermate embryos and we did not observe differences in crown-rump length at E12.5. The unprocessed embryos look remarkably similar. The sympathetic chain volume is not different at stages E10.5 and E11.5, and the phenotypes present in *Olig2-Cre; DTA* embryos are exacerbated over time, and are never found in WT embryos. All experiments are appropriately controlled and we are very confident that the differences reported in the manuscript could not possibly be caused by differences in developmental stages in individual embryos. Please see the time courses in updated Figure 4f, Figure 5d, h, i, l, Supplementary Fig S7.

- In the second DTA panel in 2E, what is the tubIII-positive nerve-like looking structure below the sensory ganglia originating from the neural tube, which contains some TH positive cells, is the motor nerve ablation not 100%?

The motor ablation is extremely efficient and close to 100% from early stages (before E10.5). We have performed additional controls to show the essentially complete loss of motor nerves (updated Figure 4a and Supplementary Figure S3a-d).

The former Figure 2E showed an ectopic TH+ nerve originating from the misplaced sympathetic neurons (not the neural tube), below the sensory ganglia. This panel was removed because we selected just one image for *Olig2-Cre; DTA* to avoid redundancies (other panels in the revised Figure 5 already show the premature projections forming in DTA embryos, visible in Figure 5b on sections and 5h-i via whole mount imaging). Other panels showing aberrant axon growth of the misplaced sympathoblasts are shown in supplementary Figures S4h (insets) and S7.

Figure 3:

- Please add additional data and cartoons to demonstrate the step by step sequence of events during normal development in order for readers to understand what is the difference as compared to the DTA phenotype. (also B and C should be introduced before going into the DTA phenotype in more detail.) Double staining of Phox2b and TH would be essential to combine information from figures 2 and 3.

We took the reviewer's advice and reorganized the Results to discuss the descriptions of normal development prior to any discussion of the DTA mutant (Figure 1, Figure 2 and Figure 3). We also added explanatory schematics to illustrate the sequence of events in normal development (Figure 1a and 1j, Figure 2e, Figure 3c and 3e). To better address the concept of autonomic priming along motor nerves, we performed a double staining for PHOX2B and TH showing that primed migrating SCPs express PHOX2B but not TH (updated Figure 2d and Supplementary Figure S4h).

- Please use additional markers to differentiate between SCP and glial cells and the migratory neural crest.

SCPs, satellite glia, and neural crest have similar transcriptional profiles, and until recently the lack of specific markers made it difficult to resolve them univocally. A recent study from our lab proposed markers for a "Hub state" which corresponds to SCP profile, including integrin alpha 4 (*Itga4*), *Serpine2*, and *Sox8* (Kastriti et al 2022 PMID:35815410). By performing staining with ITAG4 at E10.5, we succeeded in clearly discriminating between freely migrating NCC and nerve-associated SCPs (see the updated Figure 1f). However, unfortunately, ganglia-residing glia display a molecular signature that is still too similar to the SCP profile during embryonic stages, thus these two populations are not easily distinguishable by immunostaining. Lack of good antibodies for these newly identified markers, together with the lack of definitive markers for univocally distinguishing all neural crest subpopulations make it hard to introduce these analyses in the common practice.

Previous publications about the role of nerves in supplying glial progenitor cells to different tissues mostly took advantage of a clear temporal distinctions between the end of free NCC migration (before E11.5) and the onset of target organ formation (after E11.5). For instance, during adrenal gland development (which emerges around E12.5), SCP deposition from visceral motor axons begins after E11.5, when the early wave of NCC free migration is already completely extinct at all axial levels. For this reason, in previous studies (Furlan et al 2017 PMID: 28684471) we used lineage tracing using *Plp1-CreER* from E11.5 to E15.5 to unambiguously demonstrate the contribution of SCP to the developing organ. Instead, because sympathetic ganglia development begins prior to E11.5 and the timing of chain formation coincides with the switch between free to nerve-assisted migration of progenitors, observing actual nerve-association has been the easiest strategy to define NCC and SCP.

- Can the SCP-derived portion of the sympathetic ganglia be quantitatively shown to be missing from the motor-nerve-ablated mice? (by using specific genetic reporter lines or by photoconversion or by an onset of a Cre-reporter, or an injected fluorescent dye (which may not be possible to correctly target in utero))

We could make a few quantitative statements about the SCP-derived portion of the sympathetic ganglia. We estimate that at least 10% of sympathetic neurons in the sympathetic chain ganglia is coming from SCPs after initiation of tracing (see lineage tracing in updated Figure 3b). Given the efficiency of Cre recombination and the time that tamoxifen takes to be activated in a body, this number in reality is higher. In the *Olig2-Cre; DTA* line, we observe 80% reduction of SOX10⁺ satellite glia (updated Figure 4d), a 20% reduction of PHOX2B⁺ sympathoblasts in the putative sympathetic chain ganglia (updated Figure 4e), and a 30% reduction in the TH⁺ volume in the putative sympathetic chain ganglia (updated Figure 5j) with no difference in TH⁺ soma size on sections (updated supplementary Figure S4g). Admittedly, it is not possible to necessarily prove that these reductions are directly caused by loss of the SCP-derived population versus some other mechanism, although this is the most likely explanation considering the parallel accumulation of ectopic TH⁺ and PHOX2B⁺ sympathoblasts that fail to reach their proper location.

To make a definitive statement about the contribution of SCPs to sympathetic ganglia, we employed a *Prss56Cre-Tomato* tracing to label selectively the boundary cap cell population. We found that some of these cells transition to a very specific sub-population of nerve-associated SCPs (updated Figure 3d-g). This experiment does not provide a quantification of the amount of SCP-derived neurons in sympathetic ganglia, since it allowed the detection of only a minor fraction of them (the ones originating from BCCs). However, it clearly demonstrates that BCCs, by transitioning through a SCP-like intermediate state, migrate along peripheral nerves and reach the sympathetic chain where they contribute to late sympathetic neurogenesis.

We are very grateful for the specific experimental suggestions of dye tracing and photoconversion, and we seriously considered it, however we were unable to find such reporters among available Cre lines and using photoconversion would require the in vitro cultivation system which is not currently working well for embryonic days after 10.5 (according to our own experience).

- Please clarify what the authors mean by the following sentence on page 5: "Although this experiment lacks a sensory-ablated control to tease out a specific role for sensory fibers, the current data suggests that the misplaced sympathetic neurons might result from the migration of neural crest and SCPs along improperly positioned sensory axons." Is the migration of primary neural crest cells also dependent on the guidance of the sensory nerves? Which data supports this and how can those cells be separated from the SCPs this manuscript is focused on?

This was a misunderstanding and we take full responsibility for our previously unclear language. We have been careful to avoid such ambiguities in the revised manuscript. Neural crest migration is not

dependent on guidance from sensory nerves, but SCP migration is dependent on motor and sensory nerves, as SCPs do not survive without a nerve (they require NRG1 nerve-associated signal).

- 3A: What does "next to sg" mean? Why are there TH positive cells in the neural tube? Please show the TH channel separately for better interpretation of the data. Why are there no TH positive cells on the nerve approaching the WT sg? Can the authors separate, by using specific markers, the motor and sensory axons and show which ones are used by the SCPs? Please also use specific markers to separate the SCPs and glial cells that are associated with the nerves (the text uses both terms); this part is very confusing.

This panel has been swapped with updated Figure 4a to more clearly show that while in controls SOX10+ SCPs associate with both motor and sensory axons, in the absence of motor nerves all SCPs associate with sensory fibers. The previous images were also showing TH staining, which was not required for the purpose of the panel. TH signal in the spinal cord was actually real since at very early stages (between E9.5 and 10.5) motor neurons express low levels of TH, which is then downregulated after E10.5.

In the updated figures, we have used motor and sensory specific markers to show that SCPs migrating toward the sympathetic chain travel mostly along motor axons of the white ramus (Figure 1g-i). We apologize for the confusion and have made things much clearer in the revised manuscript.

- 3B: why are the Phox2B+ cells called sox10/Phox2b double positive cells – were the double positive cells selected by using Imaris or an equivalent image analysis software?

Yes, we had used Imaris to isolate the double positive cells, but overall the sensitivity of using this method in whole mount images to detect those double positive cells was not enough to detect early stages of priming at E10.5 (when cells just begin to express PHOX2B). Since these priming events were more readily apparent on sections, and the message was redundant with the panels in updated Figure 2, we decided to remove the old panel in former Figure 3b.

- 3B and C: can the authors add a fourth channel to use specific markers to separate between motor and sensory axons? Similarly, add a fourth channel for TH to separate between Phox2b and TH. Please show the Sox10 channel alone to show whether it overlaps with the Phox2b.

We were not able to add a fourth channel to these images specifically, but we did perform extra staining experiments during the revision to address these concerns. Please see Figure 1g, 1h, 1i for distinguishing motor and sensory fibers in wild-type embryos and Figure 4b in the case of motor ablation. For TH/PHOX2B double staining, to show that primed cells do not overtly differentiate on the nerve in the control embryos, please see updated Figure 2d and updated Supplementary Figure S4h. We also have isolated the SOX10 channel for the old Figure 3C (now updated Figure 2a-c).

- 3C: Why are there so many (in proportion) more sox10+ cells as compared to Phox2b positive cells in the sacral ganglia,

This figure (now Figure 2a-c) shows a pseudo-temporal sequence of events reconstructed from a combination of time points and axial levels. On the very left we are looking at a part of the body where nerve outgrowth has only started, so the cells in the sympathetic chain anlagen derive from direct migration of neural crest. They didn't largely express PHOX2B yet because this is such an early point in sympathetic development they are in the process of priming toward the sympathetic fate, whereas SOX10 is expressed throughout migrating neural crest cells and downregulated only during neurogenesis (See also schematic in Figure 1a).

- I'm not following why the sox10+ SCPs on the tip of the nerves (in the text now referred to as motor nerves and not sensory) in the sacral axial level are not Phox2B positive

We refer to these nerves as motor because we specifically traced motor fibers with the *Hb9-GFP* reporter. The most likely explanation is that the sacral SCPs on the tips of the nerves (Figure 2a, left) are not yet close enough to the dorsal aorta, which is a known source for inductive signals to induce sympathetic priming and activate PHOX2B expression.

- (and most cells in the sg are Phox2b negative as well) while Phox2b is turned on in the equivalent cells along the nerves in the more anterior images.

The most sacral image represents such an early time point that we are seeing the nucleation of autonomic priming in the early anlagen.

- If this demonstrates that sacral cells entering the sg are still primary neural crest cells, why are they also using the motor axons for guidance? Please clarify.

This timecourse demonstrates that the earliest wave of primary neural crest cells does not use motor axons for guidance. All the remaining images are meant to contrast that current understanding of sympathetic development with the new model we propose in this study, in which nerve-assisted migration complements the primary colonization of the sympathetic anlagen.

We have now clarified these models and included schematics that should be informative to all readers (for instance Figure 1a and Figure 2e).

Figure 4

- Fig A raises the same central concern regarding the focal plane of the imaging (similar to fig 1). Please explain why the images don't show TH projections into the drg in the WT. In the fig A E12.5 DTA image, traces of the DRG are clearly visible in the background whereas they are not shown in the WT. This systematic discrepancy in the imaging is concerning.

TH expression in the DRG is very dim compared to the SG. Indeed, one can see the outline of DRGs through TH staining on a computer screen but hardly in printouts that have lower contrast. To resolve this apparent discrepancy, we increased the brightness on all panels to show that DRGs are present in every panel and that our imaging is consistent (updated supplementary Figure S8a and S8b).

- A misshape analysis in 3D (trace the ganglia shape, overlay them and calculate the differences) would define and quantify the results better, as the provided size calculation gives no significant change. The n only equals 2 in some cases (n should be increased).

As requested, we have added a shape fitting panel to the figure, which makes it clear how the ganglia forms as two parts instead of one in the *Olig2-Cre; DTA* embryos. We added the additional quantification of the *Neurog1-KO* embryos as requested.

Figure 5

- In the text the authors say: "Unexpectedly, we found that in mutant embryos sympathoblasts expressing high levels of TH (in contrast to typical sparse, low-TH sensory neurons in normal DRG) were inappropriately located around and within the dorsal root ganglia and, conversely, that sensory neurons were misplaced in the sympathetic ganglia (Figure 5A)." Looking at the images, the authors may want to soften their statement to concluding that sensory somas are found in the sympathetic ganglia, whereas the evidence for the presence of TH -positive cell bodies in the drg is not convincing as the conclusion relies on the subjective interpretation on where the sg ends and the drg starts, and some TH positive cells are also found in the WT drg.

It is true that DRGs express low amounts of TH, but that is nowhere near the levels of TH found in sympathetic ganglia. The fact that there is a direct interface between these ganglia is part of the phenotype we observed in *Olig2-Cre; DTA*. During revision, we expanded the characterization of nerve-ablated mutants at late stages. We have added new analysis of E15.5 embryos and quantified the distance between sympathetic and sensory ganglia showing it is significantly reduced in *Olig2-Cre; DTA* compared to controls (updated Figure 6a-b). Moreover, we observed that the DRGs themselves are also apparently misshapen, and aberrant sympathetic projections that traverse DRGs are accompanied by DRG fragmentation. During the revision we also found the DRGs were smaller than observed in wildtype littermates (updated Figure 6e-f). Overall, it is clear that the boundaries between these two ganglia get disrupted in the absence of motor innervation. Here the reviewer can find the description of Figure 6 in the text:

“We next examined the organization of peripheral ganglia at intermediate and late developmental stages in motor nerve ablated embryos. Olig2-DTA mice are not viable at birth, but survive in the womb, allowing investigation of embryos until birth. At E15.5, the ectopic clusters of sympathetic neurons often appeared larger than the regular sympathetic ganglia (Fig 6a) and were frequently abutting the DRGs, resulting in a significantly shorter distance between sympathetic and sensory ganglia (Fig 6b). Even more drastic alterations were observed at E18.5, wherein the DRG and sympathetic chain ganglia were interspersed, with sensory neuroblasts intermingling within the sympathetic ganglia (Fig 6c, insets). These defects were observed in most of the peripheral ganglia in mutant embryos, but never in controls (Fig 6d). Conversely, sympathetic neurons and their axons aberrantly invaded the DRGs thereby disrupting their structure (Fig 6c and 6f). Consequently, sensory ganglia were generally smaller (Fig 6e) and exhibited an abnormal morphology (Fig 6f).”

Figure 6

- The RNAseq data to show putative molecular signals seems far fetched and problematic. Importantly, the joint data sets are from significantly different developmental stages (neural crest is from E9.5 whereas the Amin et al data set is E12 (and not 10.5 as falsely stated in the text and figure legend). The search for putative receptor-ligand interactions between these data sets from completely different stage embryos does not provide a realistic base for a search on putative interactions.

During the revision we resolved this problem by comparing scRNAseq data from E12.5 motor neurons (Amin et al 2021, already used in the previous submission) with time-matched data (E12-E12.5) extracted from neural crest lineage transcriptomes recently published by our lab (Kastriti et al 2022, PMID: 35815410). In both previous and current analysis of cell-cell interactions, we identified Semaphorin signaling as the most promising candidate signaling pathway (updated Figure S9). We have improved the characterization and description of Sema-related phenotypes but we have nevertheless decided to present the data in Supplementary figures since we did not reach a complete understanding of the contribution of this signaling pathway.

- How were the five candidate pathways predicted to mediate motor neuron-to-neural crest cell signaling? Why was SEMA3 pathway considered the best candidate? Please explain Figure 7.

For cell-cell signaling predictions we used the CellChat algorithm that infers the strength of cell communication pathways based on the expression of ligand-receptor pairs in single-cell transcriptional datasets (Jin et al. PMID: 33597522). The algorithm used in CellChat works using the following steps to determine the signaling strength score of individual pathways (as described in Jin et al 2021, PMID:33597522):

“a) Identification of differentially expressed signaling genes. b) Calculation of ensemble average expression c) Calculation of intercellular communication probability by modeling ligand-receptor

interactions using the law of mass action. d) The communication probability of a signaling pathway is computed by summarizing the probabilities of its associated ligand-receptor pairs.”

From the pathways displaying the highest overall score, we retained those that were predicted to be “outgoing” from motor neurons and “incoming” to neural crest derivatives, since we were interested in motor-derived signals influencing SCPs and sympathetic development. This analysis identified *Sema3* pathway as one of the top candidates, but also other pathways emerged as possible mediators of motor-to-neural crest interactions, including NRG signaling. These updated data can be found in Supplementary Figure S9.

- The selection of the knockout models needs to be justified in much more detail to convince the reader on the motor neuron- SCP- sg connection. The knockout phenotypes are thus extremely preliminary and not convincing, and nothing is quantified. The in situ expression pattern of *Sema3A* and *3F* don't per se support the hypothesis of involvement in the NC/SCP process.

Furthermore, where are *Nrp1* and *2* expressed at the time of the potential involvement in the motorneuron initiated guidance of migration?

Since *Sema3* pathway was inferred to mediate motor-SCP signaling we analyzed knockout embryos for the highest predicted ligands (*Sema3C*, *Sema3A*, *Sema3F*) and found that the sympathetic phenotype of *Sema3A/3F* double KO embryos was strikingly similar to *Olig2-Cre; DTA* in the brachial plexus (compare Figure 5l to Supplementary Figure S11), while the similarities were less obvious in other regions in whole mounts (compare Figure 5d to Supplementary Figure S11). We hypothesized that *Sema3* from motor neurons might signal through NRP1 receptor expressed in neural crest derivatives. In fact, NRP1 is highly and broadly expressed by sympathoadrenal SCPs (Lumb et al 2018, PMID:30237243), neural crest cells (showing broad expression at E9.5, Soldatov et al 2019, PMID: 31171666, and Kawasaki et al 2002 PMID: 11830568), and sympathetic neurons between stages of E10.5 and E13.5 when the phenotype of motor nerve ablation is emerging (Maden et al 2012, PMID: 22790009). The transcriptomics data from updated Figure S9 (Kastriti, et al 2022, PMID: 35815410) show broad expression of both Neuropilins across these cell populations. *Nrp2* appears to be expressed in the NCCs near the anterior portions of somites and NRP2-traced neural crest derivatives are reported to give rise to sensory derivatives, whereas NRP1+ neural crest cells are reported to give rise to sympathetic derivatives (Lumb et al 2014 PMID: 25363691).

Interestingly, we observed that conditional NRP1 ablation in neural crest lineage using the *Wnt1-Cre* phenocopies *Sema3* mutant phenotypes in the brachial plexus, suggesting *Sema3A/3F-NRP1* pathway may be involved in the placement and maturation of sympathetic progenitors and may partly explain the nerve-ablation-related defects in the forelimbs (Supplementary Figure S12).

- How did the authors rule out that the phenotypes are not a consequence of neural crest development/migration related defects much before the cells reach the stage of the motorneuron involvement stage of sg formation?

We hope that in the revised version of the manuscript, the reviewer will not get the impression that we are ignoring the well-known effects that *Sema3* signaling has on neural crest migration. However, it is unlikely that the misplaced sympathoblasts in the forelimbs of *NRP1* mutants derive from altered trajectories of NCC free-migration, considering that the ectopic placement happens far-away from the dorsal aorta region in nerve-associated way. Moreover, the migratory pattern of nerve-associated neural crest derivatives (*SOX10+*) in the forelimb seems normal (in both WT and mutants, the cells are attached to the same nerve in the same location), but sympathetic differentiation (*TH+*) occurs at this ectopic location in association with nerves of the brachial plexus in *Wnt1-Cre/NRP1-flox* embryos (Suppl. Figure 12c-d). It is highly unlikely that this aberrant acquisition of sympathetic fate along the extensions of brachial plexus derives from aberrant free migration of the neural crest, but rather it is consistent with the idea that progenitor cells use peripheral nerves to colonize the forelimbs and sympathetic priming/neurogenesis is altered (i.e., takes places at ectopic locations) when *Sema3-NRP1* signaling is perturbed –or when motor axons are ablated in *Olig2-Cre; DTA* embryos. Indeed, the presence of ectopic sympathetic neuroblasts found on sensory nerves in *Olig2-Cre; DTA* embryos supports this logic, as motor nerve outgrowth deficit does not affect the basic neural crest migration patterns (also according to our data). In any case, we downplayed the claims in the revised version of the manuscript, moving all results related to *Sema* pathway to supplementary.

- In order to make credible claims, the expression patterns and phenotypes need to be studied in a much more detailed manner, which, if properly done, will require a significant amount of additional experiments.

In the revised manuscript we expanded the characterization of *Wnt1Cre;Nrp1^{fl/fl}* and *Sema3A/3F-DKO* by performing whole mount immunostaining, which provided insight to the potential contribution of Semaphorin signaling. These experiments also helped us to conclude that some of the defects observed in *Olig2-Cre; DTA* were not recapitulated in SEMA-related mutant (e.g., the aberrant hindlimb sympathetic innervation; compare Suppl. Figure S7 and Suppl. Figure 12). We did not detect obvious alterations in the assembly of the sympathetic chain in *Sema3C* KO embryos (Suppl. Figure 10), which was unexpected given that SEMA3C was top SEMA3-related hit in the CellChat screen.

While the use of conditional mutants allowed us to conclude that *NRP1* signaling is required in the neural crest lineage, we did not have the chance to analyze motor neurons-specific mutant mouse for *Sema3A/3F*. In general, we de-emphasized the claims regarding the involvement of *Sema3-NRP1* pathway showing the data as Supplementary Figures and implemented the discussion regarding the possible implications and limitations of the findings.

- In sum, Figures 6 and 7 seem tangential to the main finding of the manuscript and the authors should consider removing them and instead focus on improving the data on the main points of the story to convincingly test their hypothesis.

We agree with the reviewer and decided to move the Sema3-related data, including the new results, into the Supplementary material. Based on new analysis that we conducted during this revision, it is clear that there is a significant portion of a motor nerve ablation phenotype that cannot be explained by Sema pathway perturbation results. This means that there are other motor nerve-derived signals in addition to Sema, and the overall picture is highly combinatorial and complex, making the full line of investigation impossible to fit into a single study. We hope the reviewer agrees that this aspect of the story is better supported in the revised manuscript, ensuring its inclusion does not disrupt the logical flow of the study or detract from the main findings.

REVIEWERS' COMMENTS

Reviewer #1 (Remarks to the Author):

The authors have made a substantial effort to address all comments. They have done an adequate job, and the figures and text reads much more clearly now.

I do not take the practical limitation of material negatively, as we all know our research is indeed often affected by time, technical-, and practical feasibility. I think that the authors have addressed this nicely in their revised version, and agree that the data should be included.

Reviewer #2 (Remarks to the Author):

I greatly appreciate the efforts by the authors to provide more data and to improve the quality of the images and writing, in addition to answering the concerns and suggestions by this and the other reviewers. They have added additional beautiful data to document the development of sympathetic ganglion neurons relative to the motor nerves. They also added neat single-cell analysis of late-migrating SCPs and their contribution to sympathetic neuro/gliogenesis, which in itself is interesting.

While the manuscript certainly is improved over the previous version and in general addresses an important and interesting topic, its reading still leaves me a little puzzled as to what we actually learn mechanistically about the development of the sympathetic nervous system. The overall conclusions forwarded by the paper still remain quite vague and lean heavily on the descriptive side (e.g. "motor nerves coordinate the placement, maturation, and axonal navigation", "motor fibers regulate neurite outgrowth of sympathoblasts").

The conclusions regarding the role of *Sema/Nrp* in this process has been toned down in the revised manuscript and the corresponding data relegated to supplementary status (it is now stated that they apparently mediated only part of the influence of motor nerves on sympathetic ganglion/axon development). However, it still remains largely unclear how *Sema/Nrp* signaling would be involved these processes. *Sema* mutants were shown to have sympathetic ganglion defects that partially resembled those observed upon motoneuron ablation. But what is the role of *Sema* signaling in this process? Do they serve as guidance molecules for restricting or promoting their migration or an inductive signal regulating gene expression or maturation?

There is another central issue, which I belatedly realized (for which I apologize). There is a previous study actually did directly address the contribution of both motoneurons (using the same approach of motoneuron ablation as the present study) and sensory neurons for sympathetic axon development (Wang et al. *Development* (2014) 141 (9): 1875–1883). This paper is cited in the manuscript, but this issue is not discussed. I am particularly surprised by this omission since one of the corresponding authors of the present study is a co-author of the Wang et al paper. The previous study by Wang et al. had directly addressed the impact of removing sensory neurons and axons on sympathetic axon development and concluded that sympathetic axon guidance actually depends on the presence of preformed sensory axons and only indirectly on motor axons (which in turn strongly influence sensory axon guidance). These previous findings are effectively ignored by the authors, even though the question of whether motor or sensory nerves direct sympathetic guidance is discussed in the manuscript. I am aware that this is inconvenient and complicates their conclusions (i.e. that "motor fibers regulate neurite outgrowth of sympathoblasts"). However, since the present manuscript (in contrast to the previous Wang et al. study) did not address the possibility that it is the sensory component that additionally or primarily directs sympathetic guidance, it is necessary that they reconsider their conclusions in the context of these previous (and at least partially) contradictory findings.

Overall, the data and careful documentation of the paper has the potential to provide a valuable albeit largely descriptive resource for the development of 'late' aspects of the sympathetic nervous system in relationship to the development of the peripheral nerves. Although I appreciate the efforts that went into the study, the limitations that remain to the revised work leave me unable to recommend publication of the revised manuscript by *Nature Communications* at this stage.

Reviewer #3 (Remarks to the Author):

Revision comments to Erickson et al nat comm 315712

Motor Innervation directs the correct development of the mouse sympathetic system

I want to congratulate the authors for turning the previous manuscript into an exciting, well communicated high quality paper I strongly recommended for publication in Nature Communications. I truly value the patience and effort that was put into the revision, the outcome is beautiful, and the cartoons are really nicely done. Excellent work!

I have a few additional minor points I hope the authors can clarify before publication:

1. Please clarify the order of events during sympathetic nervous system glial cell development and the contribution of motor neuron guided SCPs that become glia in the sympathetic ganglia.

As all the necessary data is already there, no new images are needed. However, I'm hoping the authors could better explain how the glial cells get to the sympathetic ganglia. Row (185) states that BCC and satellite glia contribute to sympathetic neurogenesis but it is not clearly stated that they, as I understand from the text, contribute to gliogenesis of the SG as well. This information would be good to pinpoint from normal development figures 2A-C by using distinct colored arrows. Also, addressing the presence and future role of the SOX10+ cells that are not Phox2B+ in the cartoon of figure 2e would benefit the readers. Thus, can one assume that if the cells in the white ramus are Phox2b negative at this stage of nerve-associated migration, they will not ever be primed to sympathetic neurons, but are instead on their way to the SG to become satellite glial cells? Also, row 160 mentions that the intraganglionic satellite glia form from earlier migrating NCC. Please address (if known) in the text and figures how are they different from the sox10+ glial cells that are delivered to the ganglia at E12.5 (row 206). Finally, on row 220, would it be appropriate to say that the motor nerves influence the position of the autonomic/sympathetic nervous system cells rather than autonomic neural progenitor cells?

This way, if the glial input is better communicated in figure 2, it will be easier to understand why the sox10 positive cells delivered to the ganglia (which I understood are SCPs / glial progenitors that are not primed to sympathetic neurons and never will be) are decreased and how that affects the SG formation.

2. Fig 1b-d. Please explain better why the width of the migratory stream is measured (what do we learn from that?). Also, I didn't understand what the red, blue and green colored dots present in the figures c-d. Please explain these results better in the text and figure legend.

3. Fig 3 b: Should caudal be lumbar to be consistent?

4. Are the sox10-positive cells in the left lower corner in fig S1 part of the condensing SG or is it just the smaller clump closer to the motor nerves? Please clarify in the figure legend.

5. Figure 4h: for clarity, change title to "sensory nerve associated"

6. Fig 4j: maybe say "ANS-primed" instead of just primed; the SCPs on the sensory nerves are likely to also be primed (but only for other purposes).

REVIEWERS' COMMENTS

Reviewer #1 (Remarks to the Author):

The authors have made a substantial effort to address all comments. They have done an adequate job, and the figures and text reads much more clearly now. I do not take the practical limitation of material negatively, as we all know our research is indeed often affected by time, technical-, and practical feasibility. I think that the authors have addressed this nicely in their revised version, and agree that the data should be included.

Thank you!

Reviewer #2 (Remarks to the Author):

I greatly appreciate the efforts by the authors to provide more data and to improve the quality of the images and writing, in addition to answering the concerns and suggestions by this and the other reviewers. They have added additional beautiful data to document the development of sympathetic ganglion neurons relative to the motor nerves. They also added neat single-cell analysis of late-migrating SCPs and their contribution to sympathetic neuro/gliogenesis, which in itself is interesting.

While the manuscript certainly is improved over the previous version and in general addresses an important and interesting topic, its reading still leaves me a little puzzled as to what we actually learn mechanistically about the development of the sympathetic nervous system. The overall conclusions forwarded by the paper still remain quite vague and lean heavily on the descriptive side (e.g. "motor nerves coordinate the placement, maturation, and axonal navigation", "motor fibers regulate neurite outgrowth of sympathoblasts").

We are happy that the review appreciated the progress we achieved during this extensive revision. Our conclusions are mainly based on the functional experiment, and therefore are not fully descriptive. The main points are the following:

1. We demonstrated that Schwann cell precursors and boundary cap stem cells contribute to the developing sympathetic chain (both neuronal and glial components)
2. The motor nerve plays a key role in this process, and without the motor nerve the numbers of cells of sympathetic chain dwindle.
3. The role of the motor nerve extends beyond the delivery of the sympathetic progenitor cells, as the motor nerve inhibits the premature maturation of sympathoblasts and early axonal outgrowth.
4. Finally, we show that motor nerves set barriers limiting the navigation of sympathetic fibers and preventing them from entering DRGs and causing mixed ganglia phenotype.

The conclusions regarding the role of *Sema/Nrp* in this process has been toned down in the revised manuscript and the corresponding date relegated to supplementary status (it is now stated that they apparently mediated only part of the influence of motor nerves on sympathetic ganglion/axon development). However, it still remains largely unclear how *Sema/Nrp* signaling would be involved these processes. *Sema* mutants were shown to have sympathetic ganglion defects that partially resembled those observed upon motoneuron ablation. But what is the role of *Sema* signaling in this process? Do they serve as guidance molecules for restricting or promoting their migration or an inductive signal regulating gene expression or maturation?

Answering this question would require the addition of numerous transgenic mouse models (especially motor-nerve-specific knockout of *Sema* ligands, which we do not have access to), for screening the phenotypes, which can be achieved only as a follow up large-scale effort. From our perspective, this is a matter of future studies (not even just one or two studies but many).

There is another central issue, which I belatedly realized (for which I apologize). There is a previous study actually did directly address the contribution of both motoneurons (using the same approach of motoneuron ablation as the present study) and sensory neurons for sympathetic axon development (Wang et al. *Development* (2014) 141 (9): 1875–1883). This paper is cited in the manuscript, but this issue is not discussed. I am particularly surprised by this omission since one of the corresponding authors of the present study is a co-author of the Wang et al paper. The previous study by Wang et al. had directly addressed the impact of removing sensory neurons and axons on sympathetic axon development and concluded that sympathetic axon guidance actually depends on the presence of preformed sensory axons and only indirectly on motor axons (which in turn strongly influence sensory axon guidance). These previous findings are effectively ignored by the authors, even though the question of whether motor or sensory nerves direct sympathetic guidance is discussed in the manuscript. I am aware that this is inconvenient and complicates their conclusions (i.e. that "motor fibers regulate neurite outgrowth of sympathoblasts"). However, since the present manuscript (in contrast to the previous Wang et al. study) did not address the possibility that it is the sensory component that additionally or primarily directs sympathetic guidance, it is necessary that they reconsider their conclusions in the context of these previous (and at least partially) contradictory findings.

We are happy the reviewer brought this up, as we are aficionados of that paper by Wang et al. Firstly, we need to point out that there are major differences between our work and that of Wang et al, primarily the anatomical locations used to investigate the nerve interdependencies. Wang et al. model these interactions in the distal locations, focusing on the limb and dermal/cutaneous tissues, whereas we mostly investigate the axonal navigation in the central and medial locations near dorsal aorta and the ventral root. Importantly, in the more distal locations represented by Wang et al, the sensory axon navigation depends greatly on the motor efferent fibers, whereas in more central locations near the ganglia we show sensory navigation does not depend much on the motor (only mild defects in sensory navigation are observed in this region). This is consistent with observations by Wang et al. in their paper in Figure 2A and elsewhere, where the differences are emerging mainly in limbs or cutaneous level. Most of sensory nerve patterns within the trunk and reasonably proximally to CNS appear similar in control and motor-nerve ablated embryos. Thus, the sympathetic phenotype

observed in our study in the vicinity of the sympathetic chain and DRGs is too dramatic to be explained by misrouted sensory nerves in that area, and instead, the sensory-independent effects of the loss of motor nerves explain the situation.

Following the reviewer's point and to improve our revised manuscript, we added the following text into the Discussion section:

"This was elegantly demonstrated by Wang et al. showing how different types of nerve ablations affect co-dependent patterns of motor, sensory and sympathetic innervation in skin and limbs. According to Wang et al., motor axons are essential for the subcutaneous navigation of sensory axons, and in turn, sympathetic efferent fibers require those correctly positioned sensory afferents to innervate the dermis. Furthermore, genetic removal of sensory afferent fibers during development showed that sympathetic fibers successfully follow motor nerve trajectories before entering the skin, but subsequently fail to innervate the skin entirely. This dependence of sensory axon on motor axons is anatomy-dependent, because in the absence of motoneurons, trunk sensory axons successfully navigate along normal peripheral pathways in the ventral root, showing major projection abnormalities only at further extremities such as the limbs and skin. Indeed, our motor ablated embryos show a mainly normal distribution of sensory fibers in the vicinity of the sympathetic chain and dorsal root ganglia (DRGs), with only minor or rare deviations (such as slightly shifted branching point of the white ramus). Therefore, the observed premature sympathetic nerve outgrowth and abnormal navigation from the sympathetic chain is most appropriately attributed to the absence of motor fibers."

Reviewer #3 (Remarks to the Author):

Revision comments to Erickson et al nat comm 315712

Motor Innervation directs the correct development of the mouse sympathetic system

I want to congratulate the authors for turning the previous manuscript into an exciting, well communicated high quality paper I strongly recommended for publication in Nature Communications. I truly value the patience and effort that was put into the revision, the outcome is beautiful, and the cartoons are really nicely done. Excellent work!

I have a few additional minor points I hope the authors can clarify before publication:

1. Please clarify the order of events during sympathetic nervous system glial cell development and the contribution of motor neuron guided SCPs that become glia in the sympathetic ganglia.

As all the necessary data is already there, no new images are needed. However, I'm hoping the authors could better explain how the glial cells get to the sympathetic ganglia. Row (185) states that BCC and satellite glia contribute to sympathetic neurogenesis but it is not clearly stated that they, as I understand from the text, contribute to gliogenesis of the SG as well. This information would be good to pinpoint from normal development figures 2A-C by using distinct colored arrows. Also, addressing the presence and future role of the SOX10+ cells that are not Phox2B+ in the cartoon of figure 2e would benefit the readers. Thus, can one assume that if the cells in the white ramus are Phox2b negative at this stage of nerve-associated migration, they will not ever be primed to sympathetic neurons, but are instead on their way to the SG to become satellite glial cells? Also, row 160 mentions that the intraganglionic satellite glia form from earlier migrating NCC. Please address (if known) in the text and figures how are they different from the sox10+ glial cells that are delivered to the ganglia at E12.5 (row 206). Finally, on row 220, would it be appropriate to say that the motor nerves influence the position of the autonomic/sympathetic nervous system cells rather than autonomic neural progenitor cells? This way, if the glial input is better communicated in figure 2, it will be easier to understand why the sox10 positive cells delivered to the ganglia (which I understood are SCPs / glial progenitors that are not primed to sympathetic neurons and never will be) are decreased and how that affects the SG formation.

To address this, we:

a) modified figure 2a-c to include arrows highlighting the Phox2b negative cells, some of which are likely contributing to sympathetic ganglionic gliogenesis. As a brief note, one cannot distinguish molecularly whether a migrating SOX10⁺ PHOX2B⁻ cell will certainly generate satellite glia, or simply have not activated PHOX2B yet. Similarly, some portion of those observed autonomic-primed SCPs may remain PHOX2B⁺ glia without differentiating into neurons. These nuances are difficult to rule out.

We took the reviewer's advice and modified Figure 2e to include some Phox2b negative SCPs, and mention in the text and figure legend that these cells remain gliogenic (which can be certainly said for at least the stages analyzed). We report our broader conclusion in the text:

"Together, these data are consistent with a model in which motor nerves recruit nearby freely migrating NCCs into SCPs, and that some of these motor nerve-associated SCPs become primed to an autonomic neurogenic fate en route as they approach the sympathetic ganglia, while others remain gliogenic (Fig 2e)."

b) we clarify the dichotomy of NCC- versus SCP- derived cells in the text around line 224: *"These results distinguish sympathetic neurons and glia into two separate categories: an early population derived directly from free-migrating NCC, and a late population derived from nerve-associated SCPs. Whether these cell waves become functionally identical or not remains the subject of future investigation."*

c) we clarify these labels on row 224 by labeling them *"sympathetic progenitor cells"* instead of "autonomic neural progenitor cells".

2. Fig 1b-d. Please explain better why the width of the migratory stream is measured (what do we learn from that?). Also, I didn't understand what the red, blue and green colored dots present in the figures c-d. Please explain these results better in the text and figure legend.

We added a blurb in the text around line 110:

"This transition is evidenced by a gradual shift in the migratory stream angle (Fig 1c), widening gap between SCPs and free migrating NCCs (Fig 1d), and consolidation of the SOX10+ stream as the nerve grows towards the sympathetic anlagen, showing that SCPs do not leave the nerve once attached (Fig 1e). These relationships suggest that nerves influence migratory patterns of SCPs from an earlier stage than previously appreciated."

We added detail to the relevant part of the figure legend:

"(c-e) Scatterplots of measurement of (c) the angle created by intersecting the line bisecting the outline of the NCC migratory stream, with the dorsoventral axis bisecting the neural tube (d) gap between nerve-associated SCP and the nearest free NCC, (e) mediolateral thickness of the NCC/SCP streams (distance between most medial and lateral SOX10+ cells just ventrolateral to the neural tube), perpendicular to (c). The red, blue, and green colors in (c-e) represent measurements from individual E10.5 embryos (n=3). Linear regression assessed correlation coefficients and p-values."

3. Fig 3 b: Should caudal be lumbar to be consistent?

Yes, we changed the terminology in the figure 3 for consistency.

4. Are the sox10-positive cells in the left lower corner in fig S1 part of the condensing SG or is it just the smaller clump closer to the motor nerves? Please clarify in the figure legend.

Only the smaller clump is condensing SG, the rest is either ectomesenchyme from cardiac or cranial neural crest streams. We added an arrowhead to be more specific and clarified in legend.

5. Figure 4h: for clarity, change title to "sensory nerve associated"

This quantification was performed by measuring the distance between PHOX2B⁺ cells located outside of the main ganglia in both controls and DTA, using the TUJ1 antibody in addition to TrkA to visualize the nerves. While saying "sensory nerve associated" is appropriate in DTA, where we detected numerous ectopic PHOX2B⁺ cells along sensory fibers, it would not correctly describe control samples. In fact, the number of PHOX2B⁺ cells outside the ganglia is not zero in controls, but there are some few primed SCPs exclusively along the white ramus (as we showed in Figure 2), which is mostly composed by motor fibers (see Figure 1i). We change the title to "peripheral nerve-associated" for more specificity than "nerve-associated".

6. Fig 4j: maybe say "ANS-primed" instead of just primed; the SCPs on the sensory nerves are likely to also be primed (but only for other purposes).

The label has been modified to include the prefix "autonomic-primed" as requested.